

# GrimoireLab: A toolset for software development analytics

Santiago Dueñas[1], Valerio Cosentino[1], Jesus M. Gonzalez-Barahona[2], Alvaro del Castillo San Felix[1], Daniel Izquierdo-Cortazar[1], Luis Cañas-Díaz[1] and Alberto Pérez García-Plaza[1]

[1] Bitergia, Leganes, Madrid, Spain
[2] Escuela Superior de Ingeniería de Telecomunicación, Universidad Rey Juan Carlos, Fuenlabrada, Madrid, Spain

## ABSTRACT

**Background:** After many years of research on software repositories, the knowledge for building mature, reusable tools that perform data retrieval, storage and basic analytics is readily available. However, there is still room to improvement in the area of reusable tools implementing this knowledge.

**Goal:** To produce a reusable toolset supporting the most common tasks when retrieving, curating and visualizing data from software repositories, allowing for the easy reproduction of data sets ready for more complex analytics, and sparing the researcher or the analyst of most of the tasks that can be automated.

**Method:** Use our experience in building tools in this domain to identify a collection of scenarios where a reusable toolset would be convenient, and the main components of such a toolset. Then build those components, and refine them incrementally using the feedback from their use in both commercial, community-based, and academic environments.

**Results:** GrimoireLab, an efficient toolset composed of five main components, supporting about 30 different kinds of data sources related to software development. It has been tested in many environments, for performing different kinds of studies, and providing different kinds of services. It features a common API for accessing the retrieved data, facilities for relating items from different data sources, semi-structured storage for easing later analysis and reproduction, and basic facilities for visualization, preliminary analysis and drill-down in the data. It is also modular, making it easy to support new kinds of data sources and analysis.

**Conclusions:** We present a mature toolset, widely tested in the field, that can help to improve the situation in the area of reusable tools for mining software repositories. We show some scenarios where it has already been used. We expect it will help to reduce the effort for doing studies or providing services in this area, leading to advances in reproducibility and comparison of results.

# INTRODUCTION

Software development, and in particular open source software development, relies on an increasing number of support tools (*Dabbish et al., 2012*; *Storey et al., 2010*; *Lanubile et al., 2010*). Each of them maintain data about the software development process, the

Corresponding author
Jesus M. Gonzalez-Barahona, jesus.gonzalez.barahona@urjc.es

developed artifacts, and how developers are working. The analysis of these data sources (usually referred as software repositories) has favored the creation of an active community of miners, both from academia and industry, interested in the empirical study of how software artifacts are created and maintained, and the related processes, activities and persons (*Cosentino, Izquierdo & Cabot, 2017*).

## Motivation

As the mining software repositories community has matured (*Hemmati et al., 2013*), tools have been built to retrieve and curate large datasets. Already before 2010, many tools had been built to deal with a wide spectrum of repositories: *CVSAnalY* (*Robles, Koch & Gonzalez-Barahona, 2004*), *FLOSSMole* (*Howison, Conklin & Crowston, 2006*), *FOSSology* (*Gobeille, 2008*), *SQO-OSS* (*Gousios, Kalliamvakou & Spinellis, 2008*), to name just a few of them. These tools showed how data retrieval, storage, and at least a part of the analysis could be automated and made generic enough to support different kinds of studies; were used to explore the limits to scalability, and the benefits of developing reusable tools; and served to demonstrate different approaches to avoid harming the project hosting systems, while at the same time being efficient in retrieving data (for example, by retrieving data once, storing it in a database, and later analyzing that data as many times as needed). After this "first wave" many other tools, developed during the last decade, were built on these lessons, offering more sophisticated functionality, better performance and scalability, and in some cases, more variety of data sources. Examples of this second-generation tools are *MetricsGrimoire* (*Gonzalez-Barahona, Robles & Izquierdo-Cortazar, 2015*), *Kibble* (*Apache, 2022*), or *Gitana* (*Cosentino, Izquierdo & Cabot, 2018*).

In the specific case of GitHub, which currently hosts a vast majority of FOSS software projects, and most of the public code available today, several tools are retrieving data and source code from it. Some of them provide means to query that data, or to produce some analysis and visualizations of it: *GHTorrent* (*Gousios & Spinellis, 2012*) and *BOA* (*Dyer et al., 2013a*), *GHArchive* (*Grigorik, 2022*), and *OpenHub* (*Farah, Tejada & Correal, 2014*), to mention just some of the better known. Some tools have also been deployed specifically to retrieve source code or data related to software development, and store it for preservation, such as *Software Heritage* (*Di Cosmo & Zacchiroli, 2017*) and *SARA* (*SARA, 2022*).

Despite the many benefits that all of these tools provide when a researcher or a practitioner needs to deal with software development data retrieval and analysis, there is still room for improvement in many areas. Most of the tools in these two generations are focused on one, or in some cases, a small subset of kinds of data sources; use disparate data formats and integration APIs, making it difficult to combine results for different kinds of repositories; and in many cases are not easy to deploy and operate, or difficult to use for large-scale, continuous data retrieval. Not all of these tools provide support for retrieval, storage and analysis of the data, and when they do, usually the opportunities for analysis are very limited. Only a few of them are extensible, and just a few are mature enough for large-scale, industrial endeavors. During the last years, some new tools or toolsets are emerging that try to address some of these issues, such as *PyDriller*

(*Spadini, Aniche & Bacchelli, 2018*) and *SmartSHARK* (*Trautsch et al., 2017*). *GrimoireLab*, which we started to design and implement in 2016, is one of them.

## Overview

*GrimoireLab* is a free, open source set of Python tools to retrieve, organize, analyze and visualize software development data. It automatically collects, processes and stores data from more than 30 kinds of repositories used in software development (source code management, issue tracking, code review, messaging, continuous integration, etc). *GrimoireLab* builds on previous experiences, paying special attention to recurrent issues that miners face in their activities such as data loss or corruption due to connection problems, data freshness and incremental retrieval, identities management, and heterogeneous formats that come from different data sources. It has been designed and built as a modular toolset suitable for its use by third parties, with the aim of satisfying the needs of researchers, but also of commercial exploitation.

Miners can use the functionality provided by *GrimoireLab* as a black box, to efficiently retrieve, analyze, store, and visualize data for a collection of projects. Or they can use specific tools, maybe integrating them with their own mining applications. For example, it provides modules for retrieving data from many kinds of data sources, with a common API, and for integrating third party tools for code analysis that can be used standalone from Python scripts. *GrimoireLab* also includes a module for identity management that can be used in combination with custom code to merge or tag identities, something that is fundamental to analyze activity of persons using several identities, to merge activity from different data sources, and to annotate identities with affiliation information, for example. There are also scheduling and orchestration modules that can be used or not, depending on the complexity of the scenario. *GrimoireLab* also defines some data formats for several steps in the usual analysis pipelines (raw retrieval formats, enriched formats) that can be used for integration with other tools or for replication.

## Contributions

The main contributions of the toolset presented in this paper are:

- Breadth. Support for activities in many areas related to the mining of software repositories: data retrieval, storage, analysis, identity management, scheduling, visualization, reporting, etc.
- Modularity. A modular and extensible design, including the identification of the modules useful in common tasks in this domain, and common APIs for similar functions.
- Data formats. Definition of data formats for the main stages of software development analytics.
- Readiness. Implementation as a collection of easy-to-install, easy-to-use Python packages, also available as Docker images.
- Maturity. Extensive testing and regular usage, both in academic and industrial environments.

- Extra functionality. Built-in functionality for addressing common problems in real-world data retrieval, storage and analysis: fault-tolerance, incremental retrieval, extensibility, facilities for data curation, identity management (including tracking of identities in different data sources), data persistence, traceability and uniform access to the data.

## Structure of the paper and definitions

This paper presents *GrimoireLab*, the main result of a "solution-seeking" research line (*Stol & Fitzgerald, 2018*), aiming to improve the situation in solving the practical problem of retrieving data from software development repositories, preparing it for further analysis, and providing basic analysis and visualization tools that help in exploratory studies. The approach used has been holistic, trying to first understand (by experience and by study) the problems, and then providing a toolset that addresses many of them in combination. We also show how *GrimoireLab* can be used in some research scenarios, and how it was used in some real use cases, and discuss its main characteristics both in research and industrial environments.

The rest of this paper is organized as follows: "The components" section describes the different components of *GrimoireLab*; the "Combining the modules" section illustrates how those components can be combined in several exemplary research scenarios, and in some real use cases (presented with their main magnitudes and performance metrics); the "Discussion" section summarizes and discusses the main features of the toolset, how its use in research studies may affect researchers, some lessons learned from its use in industry, and presents *GrimoireLab* in the context of other related work. Finally, "Availability and usage" summarizes availability and usage of the toolset, and "Conclusion" highlights some conclusions and future work.

Some definitions of terms that we will use through this paper are:

- *data source*: any system providing retrieval mechanisms (usually, an API) to access data related to software development: source code management, issue tracking, code review, synchronous or asynchronous communication, testing, collaborative documentation writing, Q/A forums, etc. Examples of a data source are a Git server, a Bugzilla instance, a Mailman archive, or some Slack instance.
- *kind of data source*: all data sources with the same retrieval API. Examples of kinds of data sources are "Git", "Bugzilla", "Mailman", or "Slack".
- *repository*: a part of a data source, usually corresponding to the data managed for a certain project. Examples of repositories are a Git repository, a Bugzilla issue tracker, a mailing list archive in a Mailman instance, or a Slack channel.
- *index*: all data corresponding to a certain kind of data source, as it is stored in the *GrimoireLab* database. Indexes may be raw, with data as similar as possible to the one provided by the data sources, or enriched, which are tailored to easy visualization and reporting.

- *item*: unity of data stored in an index, usually corresponding to what developers consider as a unity of action a kind of data source. Examples of items are "commit" for Git, "issue" for Bugzilla, "message" for Mailman, or "message" for Slack.
- *GrimoireLab component*: software module, maintained in a separate repository, and as a separate Python package, which is a part of *GrimoireLab*.

## THE COMPONENTS

*GrimoireLab* is structured in several components, which are outlined in this section. Components can be composed in different ways, to support different use cases. Each component can be installed as a Python package, which may need some other components to work: in that case they are installed as dependencies. Most components are Python modules that can be imported as libraries, but many of them also provide driver scripts to provide a certain CLI (command line interface).

The overall structure of *GrimoireLab* is sketched in Fig. 1. In it, components are grouped in four areas: Data Retrieval, Analytics (including permanent storage), Identities Management, and Visualization and Reporting. Additionally, there is also a module for orchestration. This separation in areas is introduced to help in the process to understand *GrimoireLab* components, and their role in the functionalities provided. At the same time, it allows for the introduction of the main interfaces that allow for the relatively independent development of the components presented in the rest of this section. These interfaces are:

- Retrieval components always provide JSON documents: data retrieved for each item is encoded as a JSON document resembling as much as possible the data structure provided by the corresponding data source. However, it also includes some metadata common for all kinds of data sources, which allows for a uniform data processing when peculiarities of a data source are not relevant (for example, when storing the data in permanent storage, or for temporal ordering of the items). When convenient, these JSON documents are mapped to Python dictionaries to provide a Python API based on Python generators.
- Identities Management components are accessed through a Python API, mapped to a CLI (command line interface) when convenient. This API allows for the registration of new identities found in data sources, for mapping them to unique identities identifying persons, and for the retrieval of tags (such as affiliation) associated to identities.
- Analytics modules produce results that are stored in permanent storage (enriched indexes in a database). Other components (Visualization and Reporting) using these results access them via the database interface to these indexes. Enriched indexes are composed by a flat JSON document per item. These documents are suitable to plug into visualization tools, or to perform further processing (for example, mapping collections of JSON documents to Python Pandas dataframes) towards specific reports. The document for each item includes the most usual fields for the analysis of the

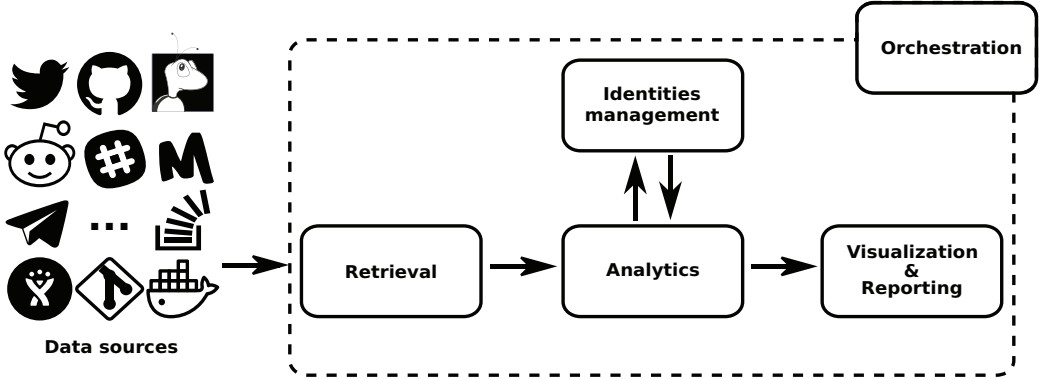

**Figure 1 Main structure of GrimoireLab.** The four main areas that structure the toolset are presented, showing the data flow between them and with data sources. Orchestration is also represented as managingall components. •Twitter: https://www.iconfinder.com/icons/211920/twitter_83905social_icon -> MIT; •GitHub: https://www.iconfinder.com/icons/298822/github_mark_icon -> MIT; •bugzilla: recreated from https://commons.wikimedia.org/wiki/File:Buggie.svg -> MPL 1.1; Reddit: https://www.iconfinder.com/icons/211911/reddit_social_icon -> MIT; •Slack: https://www.iconfinder.com/icons/710268/slack_social_icon -> CC by 2.5; •Meetup: https://www.iconfinder.com/icons/306191/meetup_icon -> CC by 2.5; •Telegram: https://www.iconfinder.com/icons/2644993/media_messenger_social_telegram_icon -> -> CC by 3.0; •Stackoverflow: https://www.iconfinder.com/icons/394194/overflow_stack_stackoverflow_icon -> Free commercial use. Use icon for commercial purpose, edit, share, etc.; •Jira: from https://iconscout.com/icon/jira-1-> MIT.     

corresponding data source, and some others common to all data sources, thus enabling cross-data source analysis (for example, "creation date" or "author" of the item).

The details of each of these areas, and their underlying components, are described in the next sections. For help going through them, check Fig. 2.

## Retrieval

*GrimoireLab* pipelines usually start by retrieving data from some software development repository. This involves accessing the APIs of the services managing those repositories (such as GitHub, Stack Overflow or Slack, San Francisco, California, USA), or using external tools or libraries to directly access the artifacts (such as Git repositories or mailing list archives). In the specific case of source code management repositories, some tools may also be run to obtain metrics about the source code. For large-scale retrieval, work is organized in jobs that have to be scheduled to minimize impact on the target platform, and to maximize performance. *GrimoireLab* provides three components for dealing with these issues:

- *Perceval* fetches data from the original data sources. Usually, it works as a library, providing a uniform Python API to access software development repositories. Relevant data in these repositories are produced as "items", that can be managed as Python dictionaries or JSON documents. *Perceval* provides access to the following data sources (although for some of them, not all APIs are always supported):

  -Version control systems: Git.
  -Source code review systems: Gerrit, GitHub, GitLab.

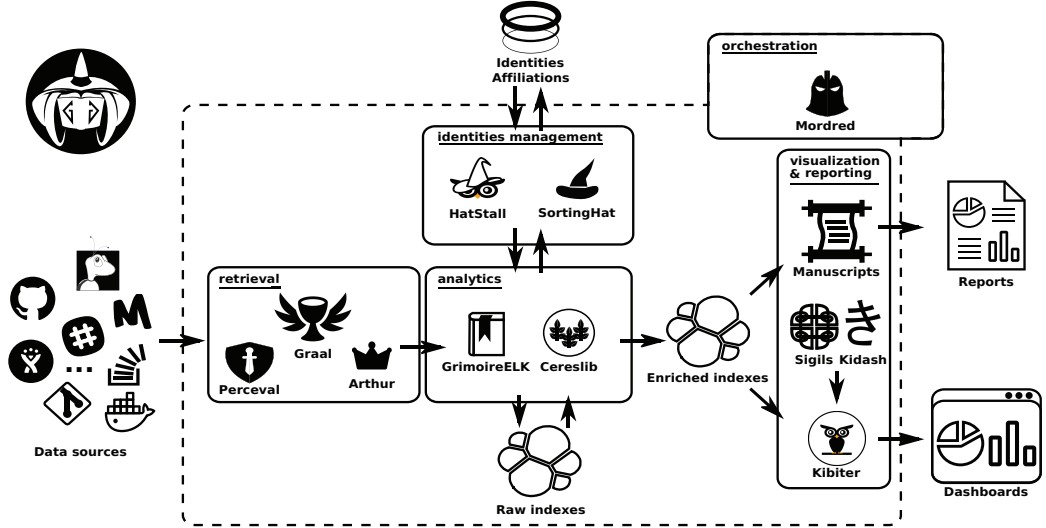

**Figure 2 Components of the GrimoireLab toolset, grouped in the same areas shown in Fig. 1.**
Permanent storage used (an SQL database for identity data, on top, and Elasticsearch for raw and en-
riched indexes) is also shown. Arrows represent the flow of data from data sources to the components in
the different areas, between them, and with the databases. •Kidash: https://es.wikipedia.org/wiki/Archivo:
KI-hiragana.gif -> CC by 3.0; •Graal: https://icon-icons.com/es/icono/Santo-Grial-holy/39098-> CC
Attribution.

- Bugs/Ticketing tools: Bugzilla, GitHub, JIRA, Launchpad, Phabricator, Redmine,
  GitLab.
- Asynchronous. communication: Hyperkitty, MBox archives, NNTP, Pipermail,
  Groups.io
- Forums: RSS, NNTP
- Continuous integration: Jenkins
- Instant messaging: Slack, Mattermost, Gitter, RocketChat, Supybot archives (IRC),
  Telegram
- Q/A: Askbot, Discourse, Stack Exchange
- Documentation: Confluence, Mediawiki
- Other: DockerHub, Meetup, Twitter

- *Graal* runs third party tools on Git repositories, to obtain source code analysis data, at
  the file level, for each commit found. It uses *Perceval* to get the list of commits, and then
  runs the tools selected on checkouts of those commits. *Graal* can run tools for
  computing metrics in the areas of code complexity, code size, code quality, potential
  vulnerabilities, and licensing. *Graal* captures the output of these tools, encoding the data
  they produce in JSON items similar to those produced by *Perceval*.
- *Arthur* schedules and run *Perceval* and *Graal* jobs at scale through distributed queues[1].

Therefore, *Perceval* and *Graal* are the two only components in *GrimoireLab* directly
performing data retrieval. *Perceval* has backends for dealing with the peculiarities of data
sources APIs, and *Graal* is specialized in the analysis of snapshots of code retrieved from a
source code management system (using *Perceval* just for getting the list of commit hashes,

[1] At the moment of writing this paper, support for *Graal* in *Arthur* is still not completely integrated.

```
{
    "backend_name": "GitHub",
    "backend_version": "0.2.2",
    "data": {
        ...
    },
    "origin": "https://github.com/grimoirelab/perceval",
    "perceval_version": "0.1.0",
    "timestamp": 1476139775.852378,
    "updated_on": 1451929343.0,
    "uuid": "c403532b196ed4020cc86d001feb091c009d3d26"
}
```

**Figure 3 Top-level fields for a certain item produced by Perceval (the example is for a GitHub pullrequest).** Origin refers to the repository of origin for the item. Timestamp refers to the moment theitem was retrieved, updated on to the moment the item was last updated in the data source. uuid is a unique identifier for the item.

in the case of Git). *Arthur*'s concern is to organize the work of *Perceval* and *Graal* when retrieving large quantities of data, by providing a system supporting the scheduling of parallel asynchronous jobs, in several nodes, and making all the details transparent to the next component in the pipeline (usually, *GrimoireELK*, see below).

A common *Perceval* job consists of fetching a collection of homogeneous items from a given data source: tickets extracted from Bugzilla or GitHub issue trackers, commits from a Git repository, or code reviews from a Gerrit instance. Each item is extended with related information (e.g., comments and authors of a GitHub issue) obtained from the data source, and metadata useful for debugging and tracing (e.g., backend version and timestamp of the execution). When a data source provides several types of items, *Perceval* usually labels the resulting items in a way that can be identified by other components processing them later. For example, the GitHub Issues API provides both issues and pull requests for a repository: *Perceval* uses the field `pull_request` to let other components know if the item is an issue or a pull request.

The output of the execution of *Perceval* is a list of Python dictionaries (or JSON documents), one per item. All these dictionaries, for all data sources, follow the same top-level schema: some fields with metainformation that can be used for traceability, for incremental retrieval, and to simplify tasks by other components. Figure 3 shows an example of the top level fields for an item corresponding to a GitHub pull request. The field `data` is a dictionary with all the data produced by the data source API, with a structure as similar as possible to the one produced by that API.

*Perceval*'s design is shown in Figs. 4 and 5. For each data source, it includes a *Client*, a *Backend*, and a *CommandLine* class. *Backend* organizes the gathering process for a specific data source sharing common features, such as incrementally and caching, and defines those specific to the data source. For instance, the GitHub backend requires an API token and the names of the repository and owner, while the Stack Exchange backend needs an API token plus the tag to filter questions. The complexities for querying the data source are encapsulated in *Client*. Most of the code for each *Client* is specific for the kind of data source it is dealing with. However, some code is shared, such as token management (for those HTTP APIs that implement it), handling of interrupted connections (for APIs

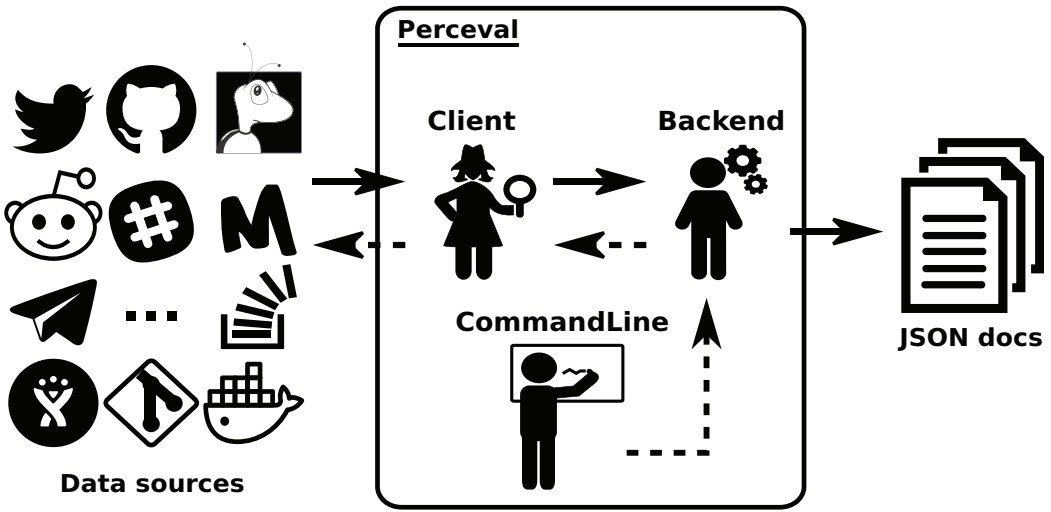

**Figure 4 Overview of the structure of Perceval.** Solid arrows show the flow of data from data sources tothe JSON documents produced (one for each item in the data source). Dashed lines show the flow ofinvocations, from the command line module to the data source API.

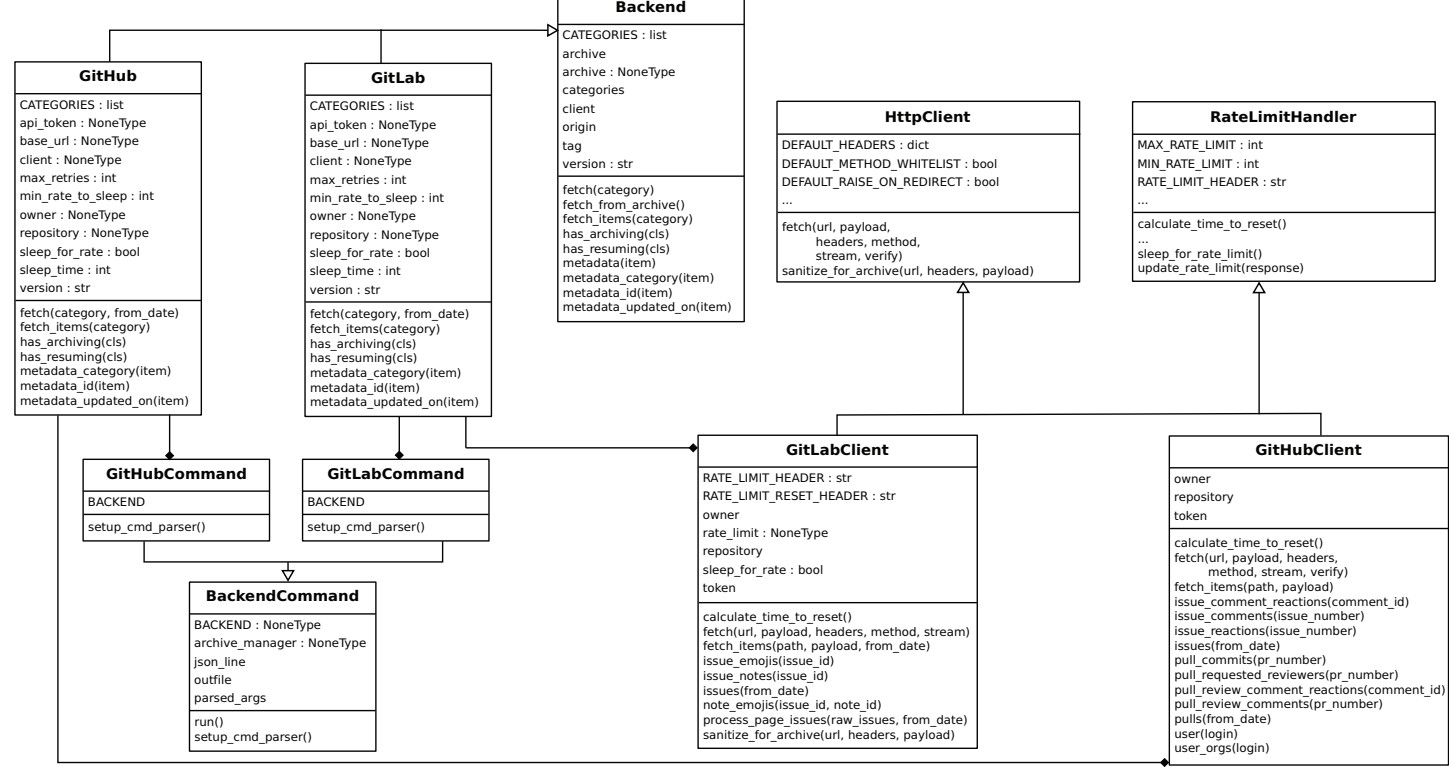

**Figure 5 Simplified structure of Perceval, in UML.** Only the main hierarchies of classes (Backend, BackendCommand, HttpClient and RateLimitHandler are shown, and only for two backends (GitHub and GitLab)).

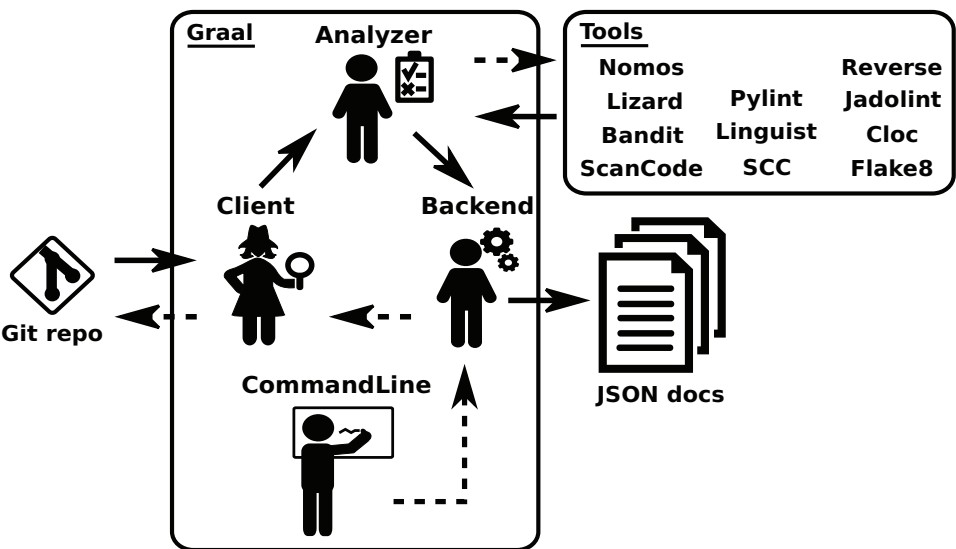

**Figure 6 Overview of the structure of Graal. Solid arrows show the data flow from the Git repository tothe final JSON documents produced by Graal, dashed arrows show the flow of invocation data.** The Analyzer module runs third party tools on the checkouts of the repository, produced by the Client module, and passes the results, properly formatted, to the Backend module. The "tools" box shows third party tools.                                 

accessed via a TCP connection), and management of the retrieval cycle (provision of a Python generator to consumers of data retrieved by *Perceval*). *CommandLine* is provided to make parameters for each data source available via the command line. More details about *Perceval* are described in (*Dueñas et al., 2018*).

*Graal* provides a mechanism to plug third party tools and libraries performing source code analysis. It produces analysis in the areas of code complexity, quality, dependencies, vulnerability and licensing. See an overview of the structure of *Graal* in Fig. 6. *Graal* uses *Perceval* to clone the Git repository to analyze, and to get its list of commit hashes, via the *Graal* Client module. Then, the *Graal* Analyzer module runs some of the third party tools on each specified snapshots (by checking out the corresponding commits), and transforms the data produced by the tools in a Python dictionary. This dictionary is fed to the Backend component, which complements it with some data (such as *Graal* version, date of the analysis, etc.), and produces the resulting JSON document. The structure and functioning of *Graal* is described in more detail in (*Cosentino et al., 2018*).

*Arthur* provides an HTTP API (via its *Server* class), which allows for the management (submit, delete, list) of *Perceval* jobs, defined as JSON documents specifying the details of the job. These details include the category of the job, parameters to run *Perceval*, or parameters to the scheduler, such as the maximum number of retries upon failures. Jobs are sent to the *Scheduler* class, which maintains queues for first-time and incremental retrievals, rescheduling in case of failures. These queues submit jobs to *Workers* (which can run in different machines), which are the key scalability element of *Arthur*. When jobs are done, workers notify the scheduler, and in case of success, they send the JSON documents, resulting from *Perceval* data retrieval, to a storage queue, where they are consumed by

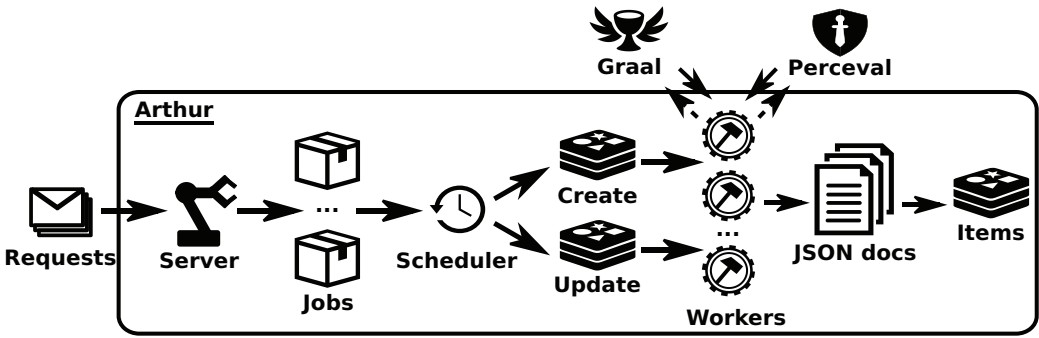

**Figure 7 Overview of the structure of Arthur.** Solid arrows show the jobs flow, since they arrive as jobrequests (usually produced by GrimoireELK), to the moment they run as Perceval or Graal invocations inworkers. Dashed arrows show the data flow from Graal and Perceval (which access the data sources) until it produces items ready to be uploaded to the database. •Redis: https://www.iconfinder.com/icons/4691219/redis_icon -> CC by 3.0; •workers: https://icon-icons.com/icon/gear-hammer/38299 -> CC Attribution.            

writers, making it possible to live-stream data or serialize it to database management systems. See an overview of the structure of *Arthur* in Fig. 7.

*Perceval* and *Graal* can be used on their own, usually from a Python script. *Arthur* provides a HTTP API, to control its operation.

## Analytics (and permanent storage)

The main aim of Analytics components is to process retrieved data to produce items more suitable for visualization and reporting, in a process called enrichment. This allows for separation of retrieval; preparation for the final visualization and reporting; and the actual visualization and reporting. Components in this area also store both the retrieved and the enriched data. This strategy of storing the data at two points is convenient for two reasons: allowing the easy reproduction of the pipelines if needed, without the need to retrieve the data once again from the original data sources, and the production of visualization and analysis at any time, sparing the need to re-enrich raw data. Of course, this is possible thanks to the identification of several actions needed for most visualizations and reporting: flattening of the data, normalization of dates, identities management, etc.

The Analytics area is covered by two components, *GrimoireELK* and *Cereslib*. The first one implements the core *GrimoireLab* pipeline: obtaining JSON items from the Data Retrieval components, storing them with persistence in "raw indexes", enriching those indexes by producing items more suitable for visualization and reporting, and storing them in persistent "enriched indexes". In the process, *GrimoireELK* also uses *SortingHat*, in the Identities Management area, for identifying new identities, and finding the corresponding unique (merged) identities. Since both raw and enriched indexes are Elasticsearch indexes, they are basically collections of JSON items (named "documents" in Elasticsearch). All usual operations on noSQL databases are possible on those indexes: retrieving one or more items given some constraints, aggregating values for certain fields for a certain selection of items, updating items matching certain values, etc.

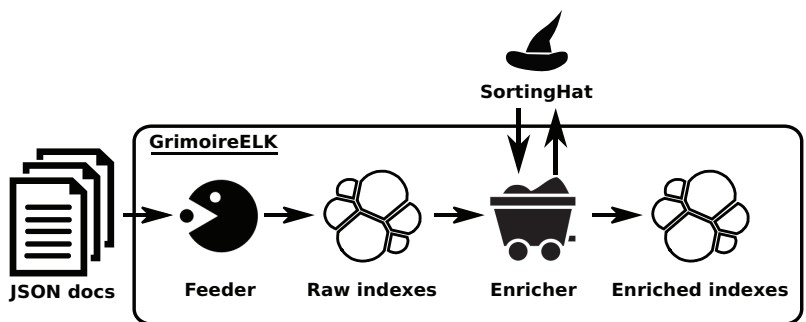

**Figure 8 Overview of the structure of GrimoireELK. Arrows show the data flow from JSON documents fed from Retrieval components to the enriched indexes.** Retrieved data is received by the Feeder, whichstores them in raw indexes. Then, the Enricher module produces enriched items, submits new identities toSortingHat, and adds unique (merged) identities and related data to those items before storing them. •feeder: https://thenounproject.com/term/pac-man/461024/ -> CC Attribution; •enricher: https://icon-icons.com/icon/mine-wagon/39492 -> CC Attribution; •redis: https://www.iconfinder.com/icons/4691219/redis_icon -> CC by 3.0.

The other component, *Cereslib*, is a library providing an API with useful functionality for certain kinds of specialized functionality. The *Cereslib* API is invoked by *GrimoireELK* to run "studies", which produces some specific enriched indexes. Studies are specialized preanalysis, producing items with a specific aim in mind. For example, one of them, "Areas of code", produces commit data at the file level (each item consists of commit metadata for each revision of each file), which is useful to analyze how different areas of code evolve.

*GrimoireELK* is the main actor of this area, interacting with the database. Its design is shown in Fig. 8. A feeder collects JSON documents produced by the data retrieval, storing them as the raw database (in an Elasticsearch index). Dumps of this raw data can be easily created to make any analysis reproducible, or to analyze directly with third party tools.

Raw data is then enriched, summarizing the information usually needed for analysis and visualization, in some cases computing new fields. For example, pair programming information is added to Git data, when it can be extracted from commit messages; or time to solve an issue is added to GitHub data. The enriched data is stored in Elasticsearch as an index with flat JSON documents, embedding references to the raw documents for traceability.

Each of the items in enriched indexes stores data about a single commit, issue report, code review, message, etc. For example, for a commit, 54 different fields are stored (see Fig. 9 for a more complete description of some of them[2]), including, among others: `author_uuid` (unique author identified, provided by SortingHat), `author_date` (author date in the commit record), `files` (number of files touched by this commit), `lines_added` (number of lines added), `lines_removed` (number of lines removed), `message` (commit message), `project` (project to which the repository is assigned), `branches` (list of branches in which the commit appears). Enriched items are not normalized due to limitations of Elasticsearch, which does not support table (index) join. This has some impact on the size of the indexes (some fields are repeated once and again,

---

[2] Full list of fields per item in the enriched Git index: https://github.com/chaoss/grimoirelab-elk/blob/master/schema/git.csv

| | name | type | aggregatable | description |
|---|---|---|---|---|
| 1 | | | | |
| 2 | author_bot | boolean | true | True if the given author is identified as a bot. |
| 3 | author_date | date | true | Author date (when the original author made the commit). |
| 4 | author_date_weekday | long | true | Day of the week when the original author made the commit. |
| 5 | author_date_hour | long | true | Hour of day when the original author made the commit. |
| 6 | author_domain | keyword | true | Domain associated to the author in SortingHat profile. |
| 7 | author_id | keyword | true | Author Id from SortingHat. |
| 8 | author_max_date | date | true | Date of most recent commit made by this author. |
| 9 | author_min_date | date | true | Date of the first commit made by this author. |
| 10 | author_name | keyword | true | Author name. |
| 11 | author_org_name | keyword | true | Author organization name from SortingHat profile. |
| 12 | author_multi_org_names | keyword | true | List of the author organizations from SortingHat profile. |
| 13 | author_uuid | keyword | true | Author UUID from SortingHat. |
| 14 | commit_date | date | true | Date when committer made this commit. |
| 15 | commit_date_weekday | long | true | Day of the week when the committer made the commit. |
| 16 | commit_date_hour | long | true | Hour of the day when the committer made the commit. |

**Figure 9 Description of some fields of the Git enriched indexes.**

when they could be in a separate table, with cross-references). However, the impact is not large, since those fields tend to be relatively small compared with the whole size of the item. The main impact of this lack of normalization is observed when one of those fields changes, and all items with the old value have to be modified. For example, if the name of an author was wrong, and is fixed, all the items authored for that person need to be fixed.

For each of the data sources supported, one or more enriched indexes are produced, aimed to have useful data to produce the metrics that are finally visualized, or used to produce reports. Therefore, aggregated metrics are not a part of the indexes stored in Elasticsearch: they are computed either by the visualizations, or by the tools producing the reports, by aggregating and filtering data present as fields in each of the items. A list of all the fields of all the indexes is also available[3].

## Identities Management

Modules in the Identities Management area manage data about personal identities. This allows analysis in which contributor identities and related information (tags), such as team/organization affiliations, are needed. *SortingHat* and *HatStall* are the components in this area. The first one deals with identities management itself, receiving new identities found, grouping them in unique (merged) identities, etc. *HatStall* provides a web-based interface so that users can manually mage identities when needed, thus complementing the algorithmic procedures that *SortingHat* follows. *HatStall* does no management on its own: for any operation on identities, it uses *SortingHat* services.

[3] Full list of fields for all enriched indexes produced by *GrimoireLab*: https://github.com/chaoss/grimoirelab-elk/tree/master/schema

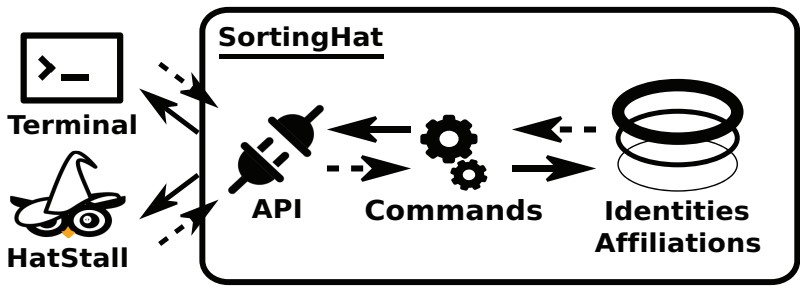

**Figure 10 Overview of the structure of SortingHat. Dashed arrows show the flow of new identities, andsolid arrows, the flow of unique (merged) identities.** The API can be accessed via a CLI (command lineinterface), or HatStall. Usually, GrimoireLab plugs to it via the CLI. The API invokes commands thatquery the database with identities, affiliation and other tags.

For understanding why identities management is convenient in *GrimoireLab*, it is important to notice how personal identities are found in data sources. Depending on the data source, identities come in different formats: commit signatures (e.g., full names and email addresses) in Git repositories, email addresses, GitHub or Slack usernames, etc. Any person may use several identities even in the same repository, and certainly in different data sources. In some cases, an identity can be shared by several contributors (e.g., support email addresses in forums). Finally, identities may need to be linked to other information, in a process we call "tagging", for certain analysis. For example, affiliation data can be extracted from domains in email addresses, or from other sources, and used to tag unique (merged) identities, so that affiliation information becomes available for actions for the corresponding person even in data sources where the data was not originally available.

In the usual pipeline, *GrimoireELK* feeds *SortingHat* with identities found in raw data, which deals with merging and tagging according to its configuration, and sends them back to be added to the enriched data. For doing its job, *SortingHat* maintains a relational database with identities and related data, including the origin of each identity, for traceability. *SortingHat* may also automatically read identities-related data in some formats: *Gitdm*, *MailMap*, *Stackalytics*, and the formats used by *Eclipse* and *Mozilla* projects. The overall design of *SortingHat* is summarized in Fig. 10. The conceptual schema of the *SortingHat* database is shown in Fig. 11. More details are described in (*Moreno et al., 2019*).

*SortingHat* uses a very conservative approach to merging identities: it uses algorithms that are quite likely to only merge identities that really correspond with the same person. This approach is used because in production environments, experience has shown how erroneously merging identities causes much more problems than failing to merge some identities, and because it can more easily be complemented with manual curation of the data. For example, the naive algorithm of "merge two identities if the email address is present in both, and it is exactly equal", fails in large datasets for common cases such as "`root@localhost`", merging for example "`John Smith <root@localhost>`" with "`Mary Williams <root@localhost>`". *SortingHat* provides this algorithm, which can be

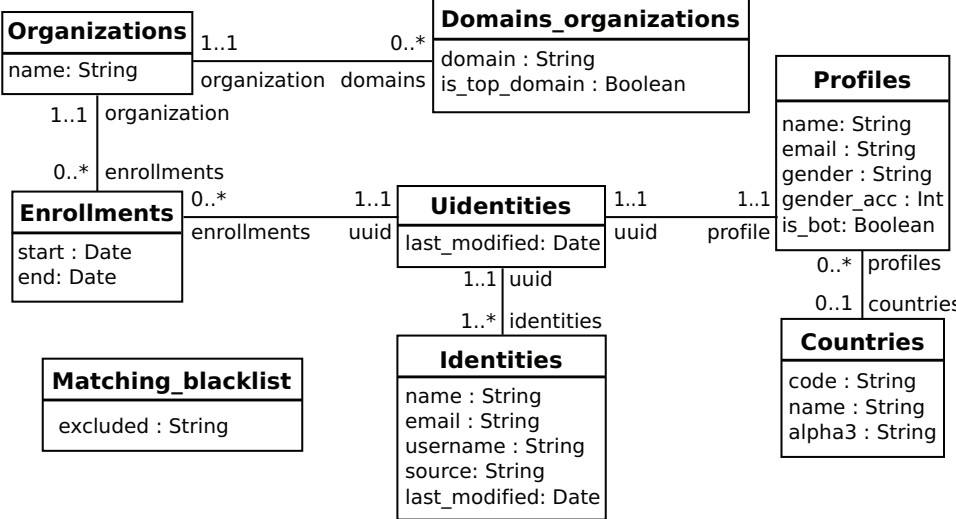

**Figure 11** Overview of the SortingHat schema, modeling identities and related information.

activated, but we had to include a deny list with common addresses such as this "`root@localhost`" to make it useful.

*SortingHat* periodically merges identities using these conservative algorithms, that can also be activated (or not) in its configuration. If more detail is needed, ingestion of identities data from reliable sources (such as company records, or FOSS Foundation data about its developers), or manual curation (usually via *HatStall*) can be used. However, since *SortingHat* offers an API to manage the identities it stores, more aggressive automatic algorithms for merging them could be easily implemented.

*HatStall* complements the automatic processes followed by *SortingHat*, by providing a web interface that can be used to manually manage identities. That interface permits, for example, manually merging, or adding affiliation data to identities. *HatStall* has proven to be very useful to fix by hand some errors that automatic procedures may cause in complex situations, or to manually complement data related to identities when there are informal sources of knowledge.

Most identities found in software repositories can be considered as personal information, therefore subject to laws protecting privacy, and to ethical guidelines on the matter. Due to this circumstance, in some cases identity management can deemed unethical, or unlawful (for example, under GDPR (*European Parliament & Council of the European Union, 2016*), if there is no clear legitimate interest for the processing of personal information, and it is considered that there is no informed consent from identity holders). To have this situation into account, *GrimoireLab* allows for the pseudoanonymization of identities as they are retrieved, via configuration switches in *Perceval* and *GrimoireELK*. If those switches are activated, *Perceval* hashes identities found in retrieved data, and *GrimoireELK* does not use *SortingHat*, producing raw and enriched indexes with pseudoanonymized identities. When orchestration is used, switches are activated with an option in the *Mordred* configuration file.

## Visualization and reporting

Visualization and reporting are usually the latest stages of any study performed with *GrimoireLab*. They are usually performed by querying data in enriched indexes, and then further processing it until the expected results are produced, or visualizing it. Although any custom program can do this, *GrimoireLab* provides some components that may help in this area:

- *Manuscripts* is a tool that queries enriched indexes, providing analytics results such as summary tables, built from templates. Tables are produced in CSV format, thus they can be imported into spreadsheets or other programs. It can also produce reports as PDF documents, including a part of the information in those tables, with some textual explanations. *Manuscripts* therefore produces a certain kind of report for a set of repositories, but can also be used as a template to produce customized reports.

- For assisting in the creation and presentation of interactive visualizations, *GrimoireLab* provides three components: *Sigils*, a set of predefined widgets (visualizations and charts); *Kidash*, which loads *Sigils* widgets to *Kibiter*, and *Kibiter*, a soft-fork of Kibana[4]) which provides web-based actionable dashboards (users can interact with the data shown, by filtering, bucketing, drilling down, etc.).

The predefined widgets provided by *Sigils* are organized as a collection of Kibana panels[5], usually grouping several metrics in an interactive dashboard that can be used not only to track their evolution over time, but also to drill down in case more details are needed. Some example of those panels and the metrics that they provide are:

- Contributors growth (shown as illustration in Fig. 12[6]): total number of contributors, active contributors over time, contributors growth by repository, difference with average of active contributors over time. This panel is offered for most of the data sources (Git, GitHub issues, GitHub pull requests, Gerrit, Bugzilla, Jira, mailing lists, etc), and in each case "contributor" is defined accordingly to the actions in that kind of repository (for Git, it is commit authors, for Bugzilla it is issue reporters, for GitHub pull requests is the pull requester, etc.).

- Bugzilla timing[7]: median and 80% percentile of open time, evolution of the status of issues over time, issues by resolution and issues by severity, evolution of the number of issue submitters over time, table with main submitters, table with latest issues, etc. Similar panels are provided for other issue tracking systems and code review systems.

- Gerrit efficiency[8]: review efficiency index (number of closed divided by open changesets), average and median time to merge, over time. Similar panels are provided for other code review systems.

- Jenkins jobs[9]: total number of builds, jobs, active nodes; proportion of build results; evolution of jobs over time: table with builds, durations, success status per job.

Almost all of the panels are actionable, in the sense that can be filtered by arbitrary periods of time (including selecting time periods in the charts), by specific repositories, by organizations (when this makes sense, such as commits performed by authors of a given

[4] Kibana: https://www.elastic.co/products/kibana

[5] Full list of the descriptions of panels provided by *Sigils* in https://chaoss.github.io/grimoirelab-sigils/

[6] Full description of the Contributors growth panel: https://chaoss.github.io/grimoirelab-sigils/panels/contributors-growth/

[7] Full description of the Bugzilla timing panel: https://chaoss.github.io/grimoirelab-sigils/panels/bugzilla-timing/

[8] Full description of the Gerrit efficiency panel: https://chaoss.github.io/grimoirelab-sigils/panels/gerrit-efficiency/

[9] Full description of the Jenkins jobs panel: https://chaoss.github.io/grimoirelab-sigils/panels/jenkins-jobs/

## Contributors Growth

This panel aims at providing a view of the contributors, their evolution and their growth over time. This information is provided for all of the data sources and gives a glimpse about the differences between the periods of analysis. The information can be filtered by data source, by organization and by project.

### Metrics

From left to right and top to bottom, the metrics provided are:

- **Total Contributors**: total number of unique contributors in any data source.
- **Active Contributors over time and Growth Analysis**: evolutionary chart displaying the number of active contributors in all of the data sources and the differences between each timeslot of analysis. The line represents the net number of contributors. The bar charts displays the difference between that timeslot and the previous one. If the difference is positive, so there's a growth in the number of contributors, the bar is green, otherwise the bar is yellow.
- **Contributors Growth by Data Source**: this table displays the several data sources of analysis and the number of contributors in each of them.
- **Active Contributors over time and Difference with the Average**: evolutionary chart displaying the number of active contributors in all of the data sources and the difference between the average of contributors for the analyzed period and how far a timeslot is from that average. The green line represents the net number of contributors. The bar charts displays how far each of the slots of time in number of contributors are from the average. The average is represented as a horizontal line in red color. This displays the average number of contributors for the period of analysis.

In addition to Kibana filters and search box ont top, filtering by `Data Source`, `Organization` and/or `Project` is allowed by using the top left corner widget.

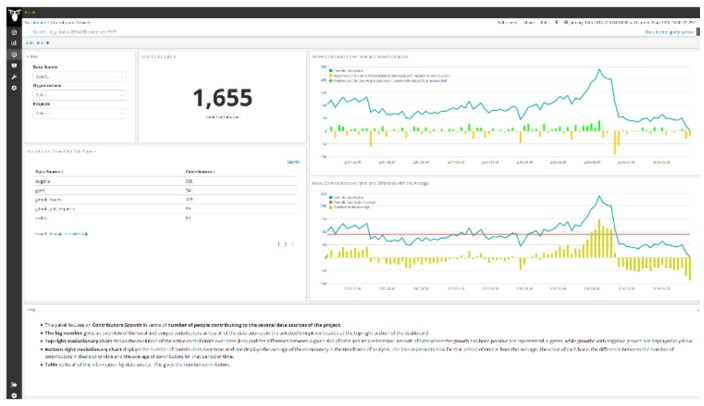

**Figure 12 Example of information about a Sigils panel: contributors growth.**

organization), etc. For these features, *Sigils* panels use the facilities provided by Kibana, which is querying in real time the enriched indexes stored in Elasticsearch. This setup allows for a lot of flexibility. In addition, users can produce their own visualizations in Kibana, if they have the right permissions.

## Orchestration

*Mordred* can be used to orchestrate all the other components to retrieve data from a set of repositories, produce raw and enriched data, load predefined widgets and generate documents and web-based dashboards. It uses some configuration files, designed to keep sensitive data separated from the one that can be publicly shared. These files include the details for accessing all repositories, including addresses and credentials, and all the servers (e.g., the *SortingHat* database manager). Repositories can be arranged hierarchically in several levels (projects, sub-projects).

*Mordred* also takes care of continuous incremental retrieval. In general, *GrimoireLab* does not use event streams and similar synchronous APIs, because they usually do not allow for the retrieval of past items, which are already not available from them. Instead, it uses timestamps and batch retrieval from APIs that provide all the items in the history of a repository. For allowing this incremental retrieval, *GrimoireELK* includes some metadata in raw and enriched indexes, based on the date when retrieved items were last updated. This metadata can be used to query data sources for all items since last update, and when processing the raw index, all items since the last processed. Even when these techniques in some cases are more complex than those based on event streams, they ensure complete retrieval of all items in the data source at the price of polling it frequently to check if new items are available. Fortunately, most of the use cases allow for some minutes

of delay in data processing, which means data sources are not polled too much. *Mordred* instructs *GrimoireELK* about when to poll based on its configuration, and *GrimoireELK* constructs the corresponding queries (to data sources, via *Perceval*, or to raw indexes) using metadata (in raw indexes or in enriched indexes, respectively). When *Arthur* is used, *Mordred* instructs it directly about polling frequencies.

## COMBINING THE MODULES

Due to its structure as a toolset, rather than a monolithic application, *GrimoireLab* modules can be used in many different combinations. In this section we describe some of them, focusing on those that may be more relevant for researchers. First, we illustrate how *GrimoireLab* can be used in some scenarios common in research: data retrieval for a custom analysis; retrieval and storage of data for an exploratory study; and large-scale continuously updated dataset suitable for different studies. Then, we describe in detail three systems that were deployed to fulfill the requirements of specific use cases: a one-time analysis of a large set of repositories, a deployment for continuously analyzing a large software project, and a system providing software development metrics a service. These cases do not intend to show insights on the analyzed data sources, but to show how *GrimoireLab* can be used to collect and process data which could later be used for different purposes, sparing the researcher or the practitioner of the burden and complexities involved.

### Research scenario: data retrieval for custom analysis

**Description:** Analysis of a relatively small number of repositories, retrieving all the data available from the API they provide, using some scripts to answer the research questions.
   **Examples:**

- Changes to the source code: activity and length of comments. **Research objective:** to explore the relationship between activity of developers in modifying the source code, and the details of their comments in those modifications. **Example of RQ:** Are more active developers writing less detailed commits? **Method:** Extraction of all commit records from a small set of Git repositories. For each of them, identification of author, computation of some metrics which could be a proxy for detail (length, number of distinct words, etc.) and estimation of correlations between aggregations of them (mean, median) for each author, and their number of commits.
- Complexity of code and change requests. **Research objective:** to explore the relationship between characteristics of code review and the complexity of the code change being reviewed. **Example of RQ:** Are those changes with complexity to the code ore prone to have longer code review processes? **Method:** Retrieval of accepted change requests from a code review system (GitHub pull requests, Gerrit, etc.), and of complexity metrics for the corresponding snapshots in the source code. For each change request, identify the starting and end time of the code review process, and its duration. For each snapshot, identify the added or removed complexity. Then, compute the correlation between duration and added complexity to answer the RQ.

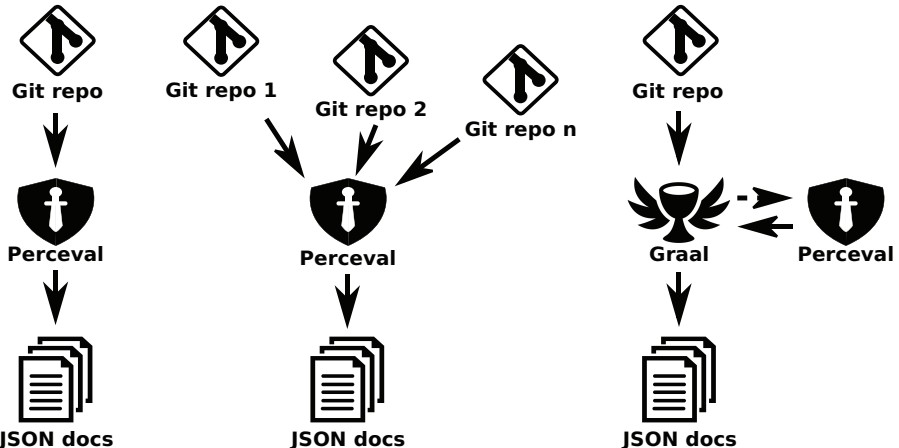

**Figure 13 Using Perceval to retrieve data from a single repository (left), or from several (center), and Graal to analyze a git repository (right).** Arrows represent the data flow from data sources to the JSON documents produced.

*GrimoireLab* **components and procedure:** The main component in this scenario is *Perceval*, which will retrieve data from the data sources API. If source code metrics are to be obtained, *Graal* will also be involved. All of them will produce collections of JSON documents that will be stored for further processing. Those JSON documents will be stored and published for reproducibility of the study, and used as the data set for the analysis.

To illustrate how *GrimoireLab* can be used in this kind of scenario, let's start with the most most simple case: the retrieval of data from a single repository (in this example, a GitHub repository), as of a file with one JSON document per line, that will be processed later. Each document, in this case, will correspond to an issue or a pull request.

```
$ perceval github [owner] [repo] --json-line > file.json
```

Since the generated JSON documents include fields to identify the mined repository, this operation can be repeated for as many repositories as needed, just adding items to the file, so that all data for a multi-repository analysis can be contained in that file.

```
$ perceval github [owner2] [repo2] --json-line >> file.json
$ perceval github [owner3] [repo3] --json-line >> file.json
```

A similar approach can be used to obtain metrics about files in any checkout of a git repository. In this case, we will use a single command to run *Graal*, which will use *Perceval* (as library) in the background to clone the repository and get the list of commits. Then, *Graal* will run third party tools to obtain complexity metrics for each source code file in each commit, producing a single JSON document:

```
$ graal cocom [repo_url] --git-path [dir_for_clone] > file.json
```

Fig. 13 shows schemes for these three cases (retrieving metadata from a single git repository, from several git repositories, and analyzing some source code metrics for all files in a git repository).

Instead of the command line version of *Perceval* and *Graal* we can also use them as modules, from a Python script. The script can perform any analysis needed, benefiting from the uniform structure of the dictionaries returned by *Perceval* and *Graal* generators, that can be consumed in loops. Of course, these scripts can be written as Python notebooks, and integrated with the usual Python data analytics tools. The general code structure in this case is as follows (DataSource is a class provided by *Perceval* or *Graal*, which implements fetch as a Python generator). For each item, origin allows to identify the origin repository, and data is a dictionary with the retrieved data.

```
repos = [DataSource([repo1 args]),
        DataSource([repo2 args]), …]
for repo in repos:
  for item in repo.fetch():
    process(item['origin'],
            item['data'])
```

Following this code structure, see below an example program to obtain the number of spelling errors in git comments per year for a collection of GitLab Git repositories. This example assumes spell_errors returns the number of spelling errors for a certain string, and get_year gets the year from a Git date.

```
repos = [Git(uri='https://gitlab.com/owner/repo1', …),
        Git(uri='https://gitlab.com/owner/repo2', …),
        …
        Git(uri='https://gitlab.com/owner/repon', …)]
terrors = { }
cerrors = { }
for repo in repos:
    for item in repo.fetch( ):
        print(f"Processing {item['data']['commit']} from {item['origin']}")
        errors = spell_errors(item['data']['message'])
        if errors > 0:
            year = get_year(item['data']['AuthorDate'])
            terrors[year] = terrors.get(year, 0) + errors
            cerrors[year] = cerrors.get(year, 0) + 1
for year in sorted(terrors):
    c = cerrors[year]
    t = terrors[year]
    print(f"{year}: {c} commits with errors, {t} total errors")
```

## Research scenario: retrieval and storage for exploratory study

**Description:** Retrieval of data from a large collection of repositories to store it in a database, so that it can be later analyzed as a part of an exploratory study.

**Examples:**

- Relationship between how bug reports are closed and developer retention. **Research objective:** To explore how the timing, or other features, related to how bug reports are closed, could influence core developer retention in a FOSS (free, open source software) project. **Example of RQ:** Does a longer time-to-close for bug reports cause developers to stop earlier contributing to the source code of a project? **Method:** Retrieve data about the issues (including bug reports) for a large and diverse set of FOSS projects, if possible with different issue tracking systems, so that specific features of it don't affect the results. Retrieve data from the source code management system of the same projects. Once all the data retrieved is stored in a database, use it to explore different proxies for time-to-close bug reports and for estimating periods of continuous contribution. For estimating time-to-close, explore different strategies for telling bug reports apart from other issues (machine learning on title and description, tags, etc). For estimating periods of contribution explore different approaches (maximum period without contributions, number of contributions over a certain period, etc.) to tell apart frequent (likely core) contributors from casual contributors. Then, explore how to estimate the period until stopping contributions (considering extending temporary periods, such as vacation). Once the most reliable method is exactly defined, conduct the study in as many repositories as possible.

- Personal trajectories in software development. **Research objective:** Explore ways to track trajectories of developers, by analyzing their footprints in different kinds of software development repositories. **Example of RQ:** Do core contributors usually follow a path from messages in communication channels to issue submitters, to code review submitters? **Method:** Retrieve data from mailing lists, GitHub issues and pull requests, and GitHub Git repositories, for a large collection of projects. Merge identities using email addresses for linking identities in email messages to identities in Git commits, and the GitHub commit API to link email addresses to GitHub user IDs. If possible, improve identities data by manually merging and de-merging identities using other data sources (for example, public Internet profiles). Once the identities data is curated, use it to identify contributions by persons in all data sources, and explore the different tracks followed.

*GrimoireLab* **components and procedure:** For these kind of studies, *GrimoireLab* enriched indexes would be convenient, and could be complemented, if needed, with *GrimoireLab* raw indexes that will be produced anyway. Using *GrimoireELK* for the data collection and enrichment ensures that the indexes will be properly stored in Elasticsearch databases. *SortingHat* will be used when identity merging is important for the study (as in the second example above). Kibana can be used to visualize the indexes in the enriched database, which can be useful for the exploratory study. For example, Kibana can easily show the activity of a single person in all data sources over time.

In this case (see Fig. 14), *GrimoireELK* will run *Perceval* to retrieve data from repositories, and then store it in Elasticsearch raw indexes. Then, *GrimoireELK* processes

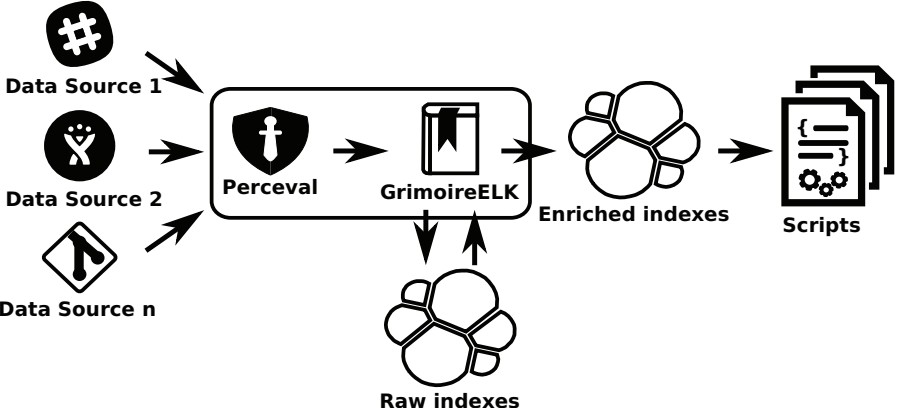

**Figure 14 Using GrimoireELK to produce Elasticsearch raw and enriched indexes.** Arrows show thedata flow from data sources to database indexes and finally to scripts that query them.

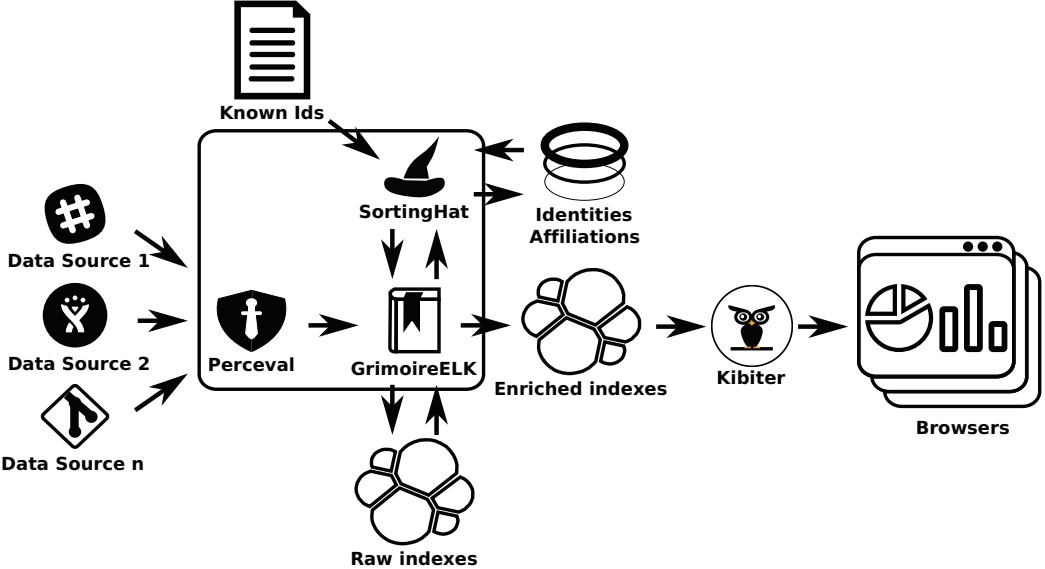

**Figure 15 Using GrimoireELK and SortingHat. Data is consumed by Kibana.**

them, creating enriched indexes. Usually, researchers will write scripts to query enriched indexes, since they are easier to query and process. But they can also query raw indexes if they need some detail that is only available in them. Since there are Elasticsearch modules for many programming languages, scripts can be written in any of them. Indexes can also be dumped as JSON files, that can be consumed directly by scripts, or uploaded to another Elasticsearch instance, where fellow researchers can work with exactly the same data.

If identity management is needed (as in the second example), *SortingHat* will be used (see Fig. 15). When processing raw indexes, *GrimoireELK* will extract identities found in raw items, providing them to *SortingHat*. For each identity in a raw item, *SortingHat*

will return its corresponding unique (merged) identity, and tagging information for it, that *GrimoireELK* will use when producing the enriched database. *GrimoireELK* can also access the GitHub commit API to obtain relationships between email addresses and GitHub user ids, and inject that data to *SortingHat*. *SortingHat* can run simple exact email-address matching algorithms to merge identities. Via *HatStall*, researchers can curate the resulting merged identities manually.

The introduction of the database allows for the massive collection of data, just by running (sequentially or in parallel) *GrimoireELK* for the different repositories to be mined. The database can easily include data for hundreds or even thousands of repositories of different kinds of data sources. The availability of enriched indexes, which are summarized, flat versions of the data obtained from the data source APIs, also allows for easy import in data structures such as Python/Pandas or R data frames, and visualizations using tools, such as Kibana or Graphana that can connect to Elasticsearch. Files produced when dumping the data in the database are also the core of good reproduction packages, and a simple way to exchange and archive data for other researchers.

## Research scenario: large-scale, continuously updated dataset

**Description:** Production of a large-scale, continuously updated dataset, with data for projects of interest using different kinds of data sources

**Example:**

- Dataset about all the projects hosted by the Apache Foundation **Research objective:** To produce a dataset that may help to better understand software development processes used in Apache projects. **Example of RQ:** Which ones are the different patterns of joining and leaving Apache projects? **Method:** Obtain the description of all the Apache projects, maintained by the Apache Foundation. Since this description includes links to all data sources (and repositories) used by those projects, produce a comprehensive list of all repositories that should be visited to maintain the dataset. Then, do a first retrieval of data from all of them, update it by frequent periodic visits, and dump it in a file that can be easily shared with researchers. Apache projects use, in different projects, Git repositories, GitHub projects for issues and pull requests, Bugzilla for issues, and change requests, mailing lists, and some other kinds of data sources, thus all of them need to be mined.

*GrimoireLab* **components and procedure:** In this scenario, involving thousands, maybe tens of thousands of repositories, from several different data sources, new problems arise. It is no longer possible to just use a single script to call *GrimoireELK*. Configuration and organization of the retrieval process becomes an issue, and for the continuous update it is important to keep raw and enriched indexes in sync with updates in the repositories, in presence of network or other infrastructure temporary failures. In these cases, *Mordred* can be used to orchestrate the setting (see Fig. 16).

To configure this setting, once this system is deployed, the list of repositories to analyze (their URLs) is written in a JSON file. Then, another file is used to configure *Mordred*

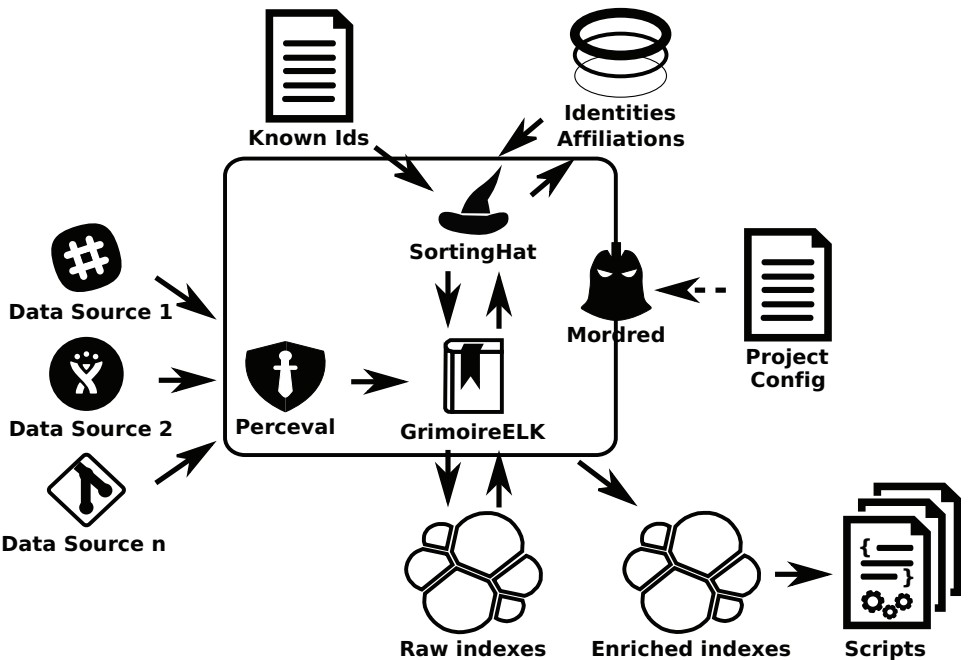

**Figure 16 Mordred driving GrimoireELK, SortingHat, and other components.**

specifying the details of the deployment (such as polling periods, kind of identity unification to perform, or specific processing to the data). For example in this file it is specified how often data sources are visited for incremental retrieval.

In some cases it is convenient to schedule the retrieval as a collection of tasks that can run in parallel. This happens for example when we can benefit from several nodes analyzing different Git repositories in parallel, or when several nodes can consume a certain API quicker than a single one. In these cases we can add *Arthur*, which will schedule *Perceval* and *Graal* jobs taking into account aspects such as availability of tokens to access data sources, or refresh periods (how often data will be retrieved incrementally from repositories). *Arthur* uses a Redis database to manage jobs and batches of retrieved items (see Fig. 17).

In this scenario, we can review the main interactions between *GrimoireLab* components:

- *Perceval* retrieves data from repositories.
- For Git repositories, *Graal* analyzes source code, by running third party tools with the help of *Perceval*.
- *Arthur* schedules *Perceval* and *Graal* jobs in workers, to organize the retrieval.
- *GrimoireELK* receives retrieved items to produce raw indexes in Elasticsearch, to some extent replicating data sources.
- *GrimoireELK* interacts with *SortingHat* to store new identities in its database and be informed about merged identities and their tags.

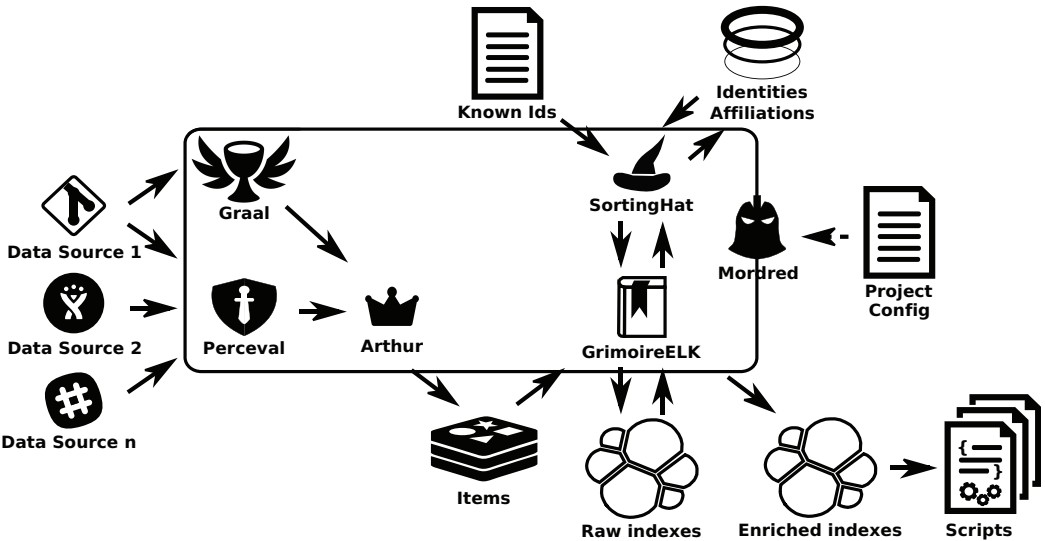

**Figure 17 A GrimoireLab system including Mordred and Arthur.**

- Using data from raw indexes and *SortingHat*, *GrimoireELK* produces enriched indexes in Elasticsearch. These indexes (and raw indexes, when convenient) can be analyzed with scripts.
- *Mordred* orchestrates all the process, according to the information in its configuration files, deciding which repositories to retrieve, how enriched indexes are produced, when data should be updated, etc.

In all the cases when Kibana is used for interactively visualizing data (see Fig. 15), *Sigils* provides a set of ready-to-use visualizations and dashboards. See examples of a summary dashboard provided by *Sigils* in Figs. 18 and 19. The use of *Arthur* is optional: users can write their own schedulers, if they prefer.

The kind of studies that can be done in this setting is similar to those done on subsets of repositories in GHTorrent, for example, but letting researchers decide both the kinds of data sources they want, and the specific projects they target (be them in GitHub or not). The drawback, of course, is that once the list of repositories is defined, researchers need to deploy the system, configure it, and wait until the data is obtained from the different data sources.

## Use case: one-time analysis of a collection of repositories

**Requirements:** One-time retrieval of all the data from two kinds of data sources (Git and GitHub) for a medium sized list of repositories, all of them related to IoT (Internet of Things).

**Magnitudes:** See Table 1.

*GrimoireLab* **setup:** The setup corresponds to the description of the research scenario "Large-scale, continuously updated dataset", described in Subsection "Research scenario: Large-scale, continuously updated dataset" (Fig. 16), although in this case the data

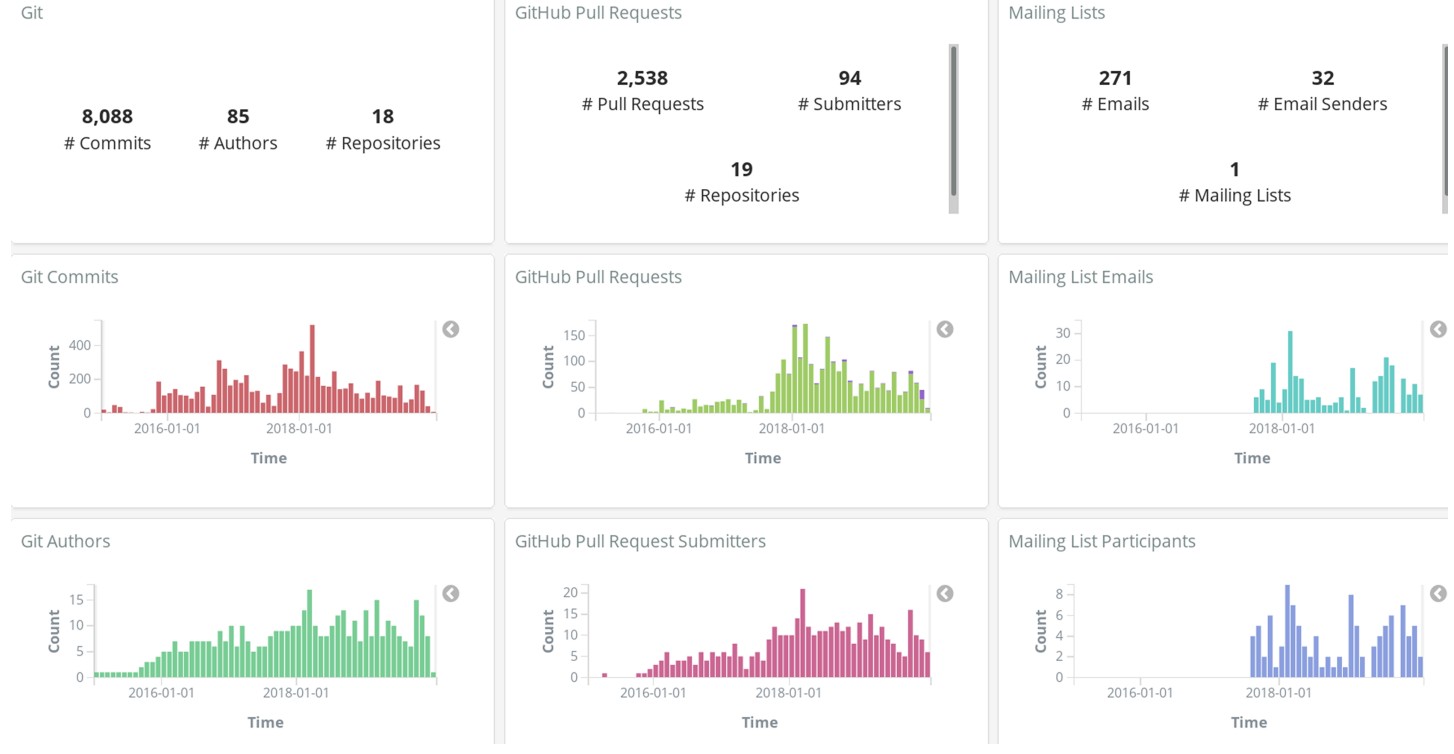

**Figure 18 Metrics summary dashboard, produced with GrimoireLab for the GrimoireLab project.**

retrieval was performed once, and not updated later. All data retrieval and analysis was done in a single thread.

Everything was run by *Mordred*, which started with data retrieval in two threads: one cloning and then extracting metadata from Git repositories (for all commits in all of them), the other one accessing the GitHub API to retrieve issues and pull requests for all repositories, using three API tokens. In each of the threads, once the retrieval for all repositories is complete, with the production of the corresponding raw index, the analysis of the retrieved data starts, until all the items (commits in the case of Git repositories, issues and pull requests in the case of the GitHub API) are analyzed.

Table 2 shows when the most relevant stages of this case started and finished, and their duration. The deployment was in a 2.5 GHz CPU with 4 cores, 8 GB of RAM, SSD storage. In both tables, "git" refers to the analysis of Git repositories, "github" to the analysis of GitHub issues and pull requests retrieved from the GitHub API. Data collection for "git" includes cloning of Git repositories for GitHub, and production of the raw index by analyzing those clones. Data collection for "github" includes waiting periods while the API tokens are exhausted, and calls to the API to resolve identities, not only retrieval of issues and pull requests. Two API tokens were used in this case.

Also in Table 2, it can be seen how the performance is close to the maximum allowed by the GitHub API token rate: 5,000 calls per hour, or about 83 calls per minute. Using two tokens, that maximum would amount to 163 calls per minute. Processing of Git commits is much faster, since it does not involve API rates, and is limited only by the Git server

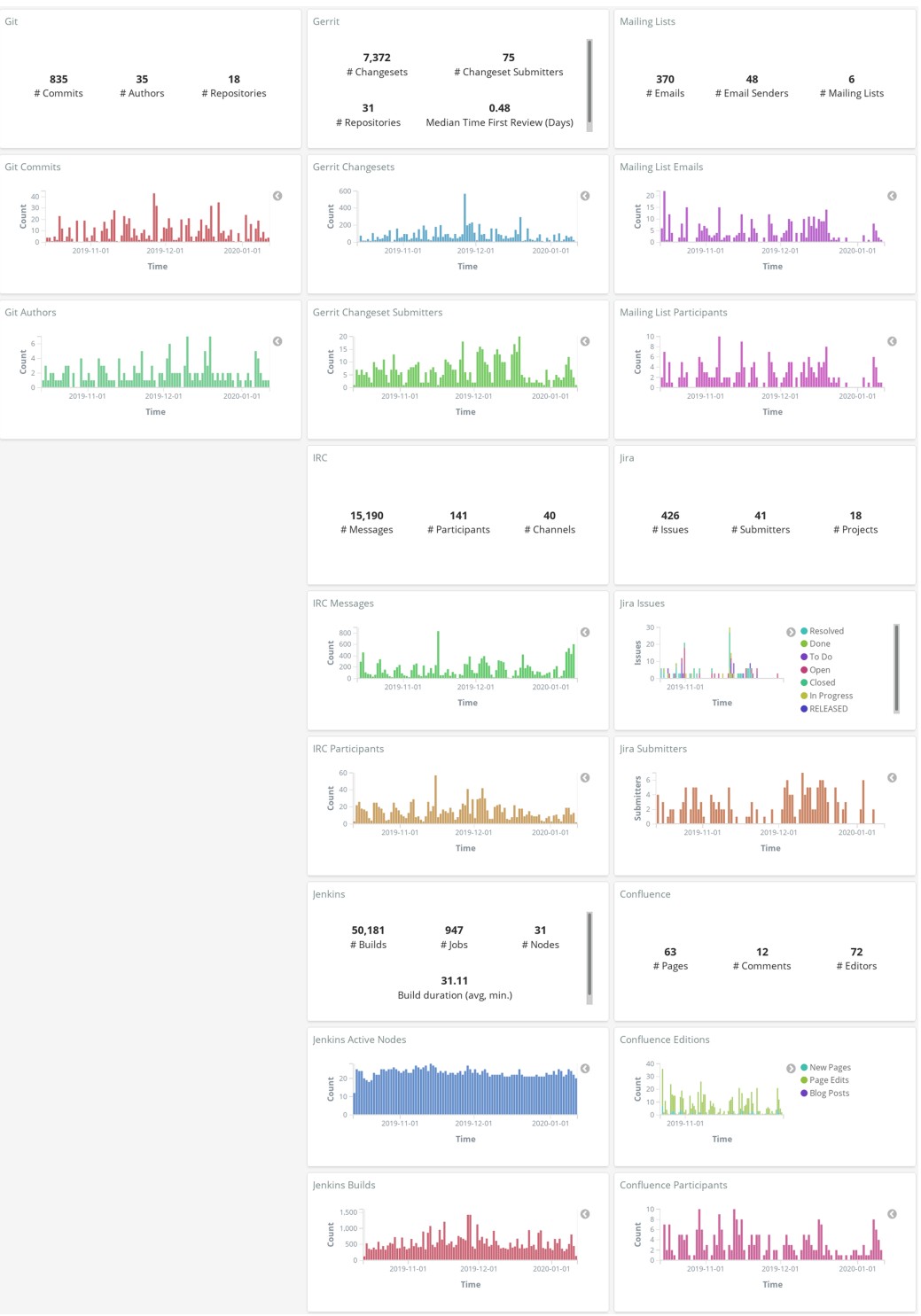

**Figure 19 Example of summary of metrics over time, produced with GrimoireLab, in this case byvisualizing enriched indexes in Kibana (data for OPNFV project as of January 2020).** Information isretrieved from Git, Jira, Gerrit, mailing lists, IRC, Confluence and Jenkins repositories.

**Table 1 Magnitudes of the IoT repositories case.**

| Git repos: | 54 |
| --- | --- |
| GitHub repos: | 48 |
| Commits (items in raw index) | 276,860 |
| Issues & pull requests (items in raw index) | 95,370 |

**Table 2 Main stages of data collection and enrichment for the IoT repositories case.**

| Elapsed time | Event |
| --- | --- |
| 00:00:00 | Starting |
| 00:00:14 | Retrieval starts (git, github) |
| 00:43:00 | Retrieval finished (git) |
| 00:43:10 | Enrichment starts (git) |
| 01:05:19 | Enrichment finished (git) |
| 10:09:18 | Retrieval finished (github) |
| 10:16:29 | Enrichment starts (github) |
| 10:51:37 | Enrichment finished (github) |

| Action | Duration | Performance |
| --- | --- | --- |
| Retrieval (git) | 00:42:46 | 108 commits/s |
| Retrieval (github) | 10:09:04 | 2.6 items/s |
| Enrichment (git) | 00:22:09 | 208 commits/s |
| Enrichment (github) | 00:35:08 | 45 items/s |

**Note:**

Top: elapsed time for the main recorded events. Bottom: duration and performance. Elapsed time and duration are in hours. Items are GitHub issues and pull requests. "Retrieval finished (git)" means all git repositories were cloned and their metadata was stored in the raw index. "Retrieval finished (github)", means all data was retrieved from the GitHub API (issues and pull requests), and it was stored in the raw index.

response time, download time, and git processing. Enrichment processes for git are much faster than for GitHub because they are also lighter: in the GitHub case they include the computing of some duration metrics, based on the data in the raw index, which is a bit longer to retrieve.

## Use case: continuous analysis of a large project

**Requirements:** Continuous analysis of all relevant repositories, from several data sources, of a large project (all software promoted by Wikimedia Foundation).

    **Magnitudes:** See Table 3.

    *GrimoireLab* **setup:** The setup corresponds to the description of the research scenario "Large-scale, continuously updated dataset", described in Subsection "Research scenario: Large-scale, continuously updated dataset" (Fig. 16), including continuous update and identity management.

    Since identity management is included in this use case, identities found during the production of the enriched indexes are by *GrimoireELK* to *SortingHat*, to get the corresponding merged identity. Therefore, enriched indexes include the identifier of the

**Table 3 Magnitudes of the GrimoireLab deployment for analyzing Wikimedia Foundation projects (enriched data), as of January 10, 2020.**

| Data source | Repos | Items | Items no. | Size (GB) |
|---|---|---|---|---|
| Git | 2,675 | Commits | 1,647,481 | 4.9 |
| Git (AOC) | | File revisions | 12,309,316 | 7.6 |
| Gerrit | 2,098 | Patchsets | 7,461,755 | 27.8 |
| Maniphest | | Issues | 231,833 | 0.44 |
| Mailing lists | 46 | Messages | 263,419 | 0.59 |
| Mediawiki | | Pages | 1,116,469 | 0.83 |

merged identity, which permits that persons with several identities are considered as a single person (merged identity). Since continuous update is configured, after enrichment threads sleep for a configurable amount of time (300 s by default), and then restart the process, retrieving incrementally new data. In a separate process, run periodically (usually between incremental retrieval phases), *SortingHat* processes its data finding new cases of identities to merge, and enriched indexes are modified adding these new merged identities to their items.

In the case of the Wikimedia Foundation Git is used for code management, Gerrit for code review, Maniphest (issue tracker in the Phabricator forge) for issue tracking, mailing lists for asynchronous communication, and Mediawiki for documentation. In Table 3 there is no repository count for Maniphest because issues in Phabricator are not organized in repositories, and in Mediawiki because in that case data is organized in pages (61,301). "Git AOC" is the enriched index for areas of code analysis, with Git data for each version of each file. The deployment has been running continuously for more than 4 years, retrieving data incrementally from repositories, except for downtime due to stopping and restarting the system due to the deployment of a version. The dashboard showing visualizations for main metrics, using standard *Sigils* visualizations, is available publicly online[10] (see screenshot of its entry page in Fig. 20).

This use case shows how *GrimoireLab* can be used in production, for continuously analyzing large-scale projects, during long periods of time. We consider this use case, which is representative of a number of others similar, as an illustration of the maturity, and adaptation to real-world constraints, of the toolset.

## Use case: metrics as a service

**Requirements:** Industrial-grade deployment, for retrieval of an arbitrary, potentially very large (tens of thousands of repositories), and visualization of some of the main metrics of arbitrary groups of repositories in it.

**Magnitudes:** See Table 4.

***GrimoireLab* setup:** The setup corresponds to the description of the research scenario "Large-scale, continuously updated dataset", described in Subsection "Research scenario: Large-scale, continuously updated dataset" (Fig. 16), not including identity management (therefore *SortingHat* and *HatStall* are not used), and with a specialized

[10] Dashboard for all Wikimedia projects: https://wikimedia.biterg.io (last visited on March 10 2021)

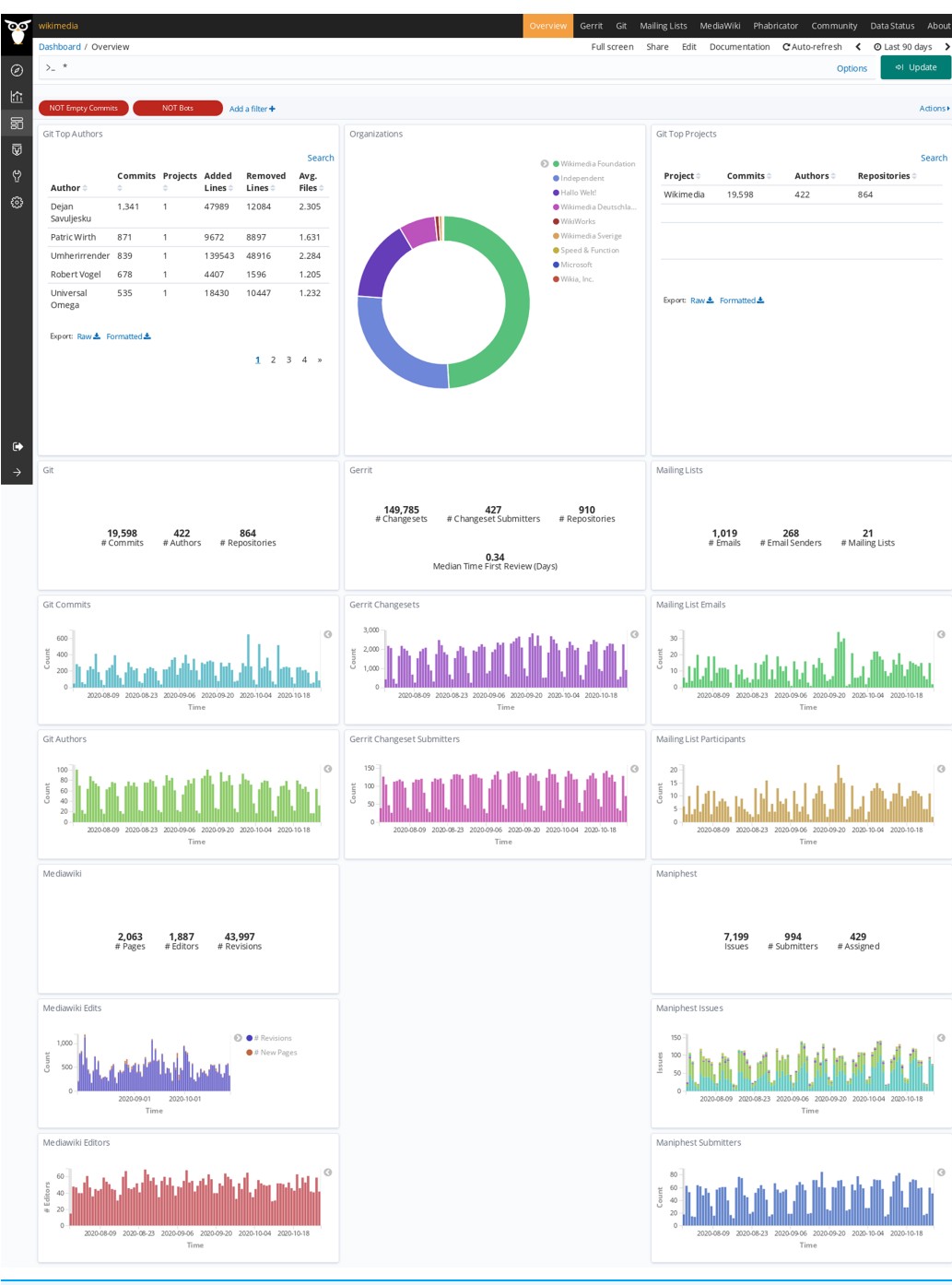

**Figure 20** Entry page to the Wikimedia dashboard, produced with GrimoireLab.

scheduling system for *Mordred* jobs, that are serviced by an arbitrary number of parallel workers.

The system described in this use case is *Cauldron*[11], which provides customized dashboards with data obtained from FOSS software development repositories. Users can order an analysis of as many repositories as they want, organized in projects (collections of

[11] *Cauldron*: https://cauldron.io

**Table 4 Some magnitudes for Cauldron, as of March 10, 202.**

| Data source | Index | Repos | Items | Size (GB) |
| --- | --- | --- | --- | --- |
| Git | Raw | 50,280 | 76,269,234 | 136.1 |
| Git | Enriched | 50,280 | 76,134,443 | 135 |
| GitHub | Raw | 40,907 | 9,089,137 | 38.1 |
| GitHub | Enriched | 40,907 | 9,016,199 | 9.6 |
| GitLab | Raw | 3,073 | 282,739 | 2.2 |
| GitLab | Enriched | 3,073 | 282,598 | 0.3 |

repositories). *Cauldron* uses *GrimoireLab* to retrieve data, analyze it, and provide visualizations. The system has been running for more than 14 months, allowing users to select GitHub, GitLab (issues and change requests in both cases), and Git repositories to analyze.

Data is retrieved by *Perceval*, by cloning repositories, and the analysis is performed by *GrimoireELK*. For GitHub and GitLab, issues and pull requests (or merge requests) are retrieved by *Perceval* from the GitHub API, using access tokens provided by users, and then enriched indexes are produced by *GrimoireELK*. Several instances of *Mordred* run in parallel (in different workers) driving the retrieval and analysis of some repositories each, according to users' demand. The system is designed to grow at least one order of magnitude larger with no change. As of January 10, 2021, it served 732 users.

In addition to some of the visualization panels provided by *Sigils*, Cauldron offers also more than 50 different Kibana visualizations, and a summary of more than 40 metrics as charts produced with JavaScript, using data provided by a Django API that queries directly the Elasticsearch enriched indexes produced by *GrimoireLab*. The main view for a project in Cauldron (see Fig. 21) includes four of these visualizations, showing the extensibility of *GrimoireLab*, in this case to interface to external visualization services.

Cauldron is one of the largest *GrimoireLab* deployments, by size of analyzed data. Therefore, the numbers shown in Table 4 can be used to estimate a lower limit of the scale (by number of repositories, by number of items in those repositories) that can be analyzed with *GrimoireLab*.

For benchmarking *Cauldron* more precisely, and with it *GrimoireLab*, we set up a specific instance of *Cauldron* running in a single machine (2.5 GHz CPU with 4 cores, 16 GB of RAM, SSD storage). We configured it to analyze the complete GNOME organization in the GitLab instance maintained by the GNOME project[12]: a total of 564 GitLab repositories, each of them with the corresponding Git repository. We started with an empty database, and we analyzed nothing else in parallel. We had 15 workers, deployed as Docker containers with *GrimoireLab* installed in them. At any given time, each worker runs at most a single job, corresponding to a certain repository, for which it was producing either raw or the enriched data. A single GitLab token was used for the experiment.

The main performance metrics of this experiment are detailed in Table 5. Several different processes were measured: production of raw or enriched indexes for each type of

[12] GNOME GitLab instance: https://gitlab.gnome.org/GNOME
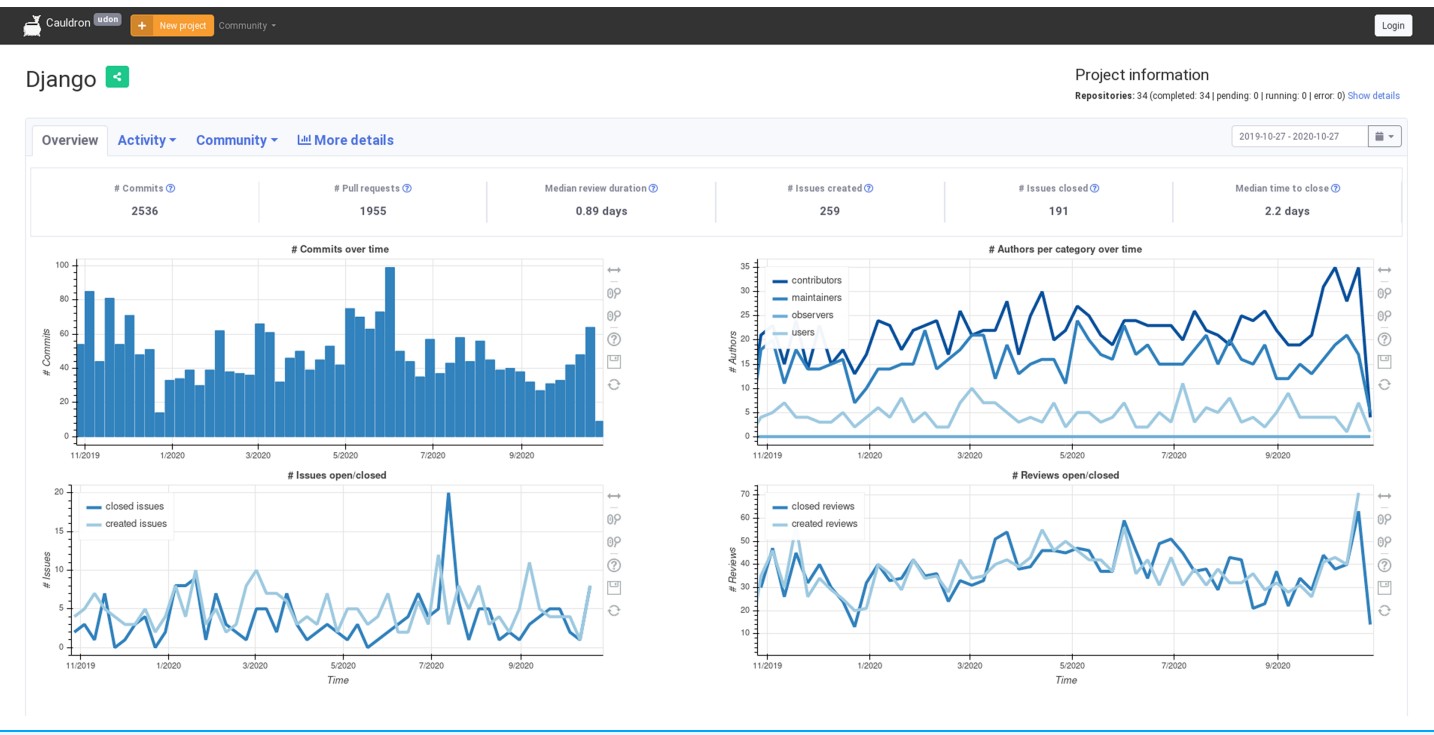

**Figure 21 Main view provided by Cauldron for an analyzed project (January 2021).**

**Table 5 Some metrics for the GNOME GitLab experiment.**

| Process | Items | Clock time | Processing time | Items/s (clock) | Items/s (proc) |
|---|---|---|---|---|---|
| All (complete) | 1,460,054 | 18:52:37.64 | 58:33:36.27 | 21.48 | 6.93 |
| Commits (complete) | 2,676,345 | 00:41:48.67 | 04:37:40.86 | 1066.84 | 160.64 |
| Commits (raw) | 1,369,386 | 00:38:13.50 | 02:46:13.62 | 597.07 | 137.30 |
| Commits (enriched) | 1,306,959 | 00:41:31.60 | 01:51:27.24 | 524.55 | 195.44 |
| Issues (complete) | 122,294 | 18:52:25.05 | 25:31:54.53 | 1.80 | 1.33 |
| Issues (raw) | 61,432 | 18:03:12.00 | 25:28:12.98 | 0.95 | 0.67 |
| Issues (enriched) | 60,862 | 18:52:11.13 | 00:03:41.56 | 0.90 | 274.70 |
| Merges (complete) | 58,311 | 18:52:27.07 | 28:24:00.87 | 0.86 | 0.57 |
| Merges (raw) | 29,236 | 18:52:00.07 | 28:20:52.28 | 0.43 | 0.29 |
| Merges (enriched) | 29,075 | 18:52:12.41 | 00:03:08.59 | 0.43 | 154.17 |

**Note:**
  Processing time is the accumulated processing time of all the jobs in all the workers, for the specified process. Items/sec is the number of items processed per second, either for clock time, or for processing time.

item (commits, issues, merge requests), production of both indexes ("complete") for each type of item, and production of all the indexes ("all"). Timing for "all (complete)" shows how long did the experiment need to complete (almost 19 h of clock time), and how much processing time it needed from workers (about 58.5 h). With 19 h of clock time and 15 workers, it is clear how most of the time workers were idle, usually waiting for the token rate to be reset. In general, throughput (in items per time) numbers are similar

to those presented in Section "Use case: One-time analysis of a collection of repositories". As we already observed in that case, processing for Git is much faster than for GitLab (or GitHub), because for them the API token rate is a strong limitation. The table also shows how processing time for enriching processes is one or two orders of magnitude shorter than clock time. The main cause of this loss of performance is the access to the database, which is done frequently to ensure enriched data is soon in stable storage. In the context of the total time elapsed for the retrieval and analysis, this decision does not cause long delays. The access to the database could be improved by using caching, Elasticsearch shards (which allow for parallel access), and by improving the hardware setup, which was not designed in this case to optimize database access.

## DISCUSSION

*GrimoireLab* is the result of many years of research and development in the area of tools for software development analytics. A part of the people involved in building it had the experience of building another, less ambitious toolset, *MetricsGrimoire*. This was fundamental to identify problems and features for both the research and industrial scenarios. In this section we discuss the main features of *GrimoireLab* along with their rationale, provide some notes on its usage for research, and present some lessons learned from its usage in industrial settings.

### Main features and their rationale

Some of the most relevant features of *GrimoireLab*, and their rationale, are:

- Minimizing interactions with data sources. Accessing data sources causes stress to them, which may lead to being banned. In addition, retrieving data from data sources is usually slow, compared to just accessing it locally. *GrimoireLab* follows the strategy of getting as much data as reasonably possible from the data source, and storing it to avoid retrieving it again in the future. Raw indexes are used with this objective, and to some extent, enriched indexes too: once an item is in the index, there is no need to retrieve it again, with no loss of functionality. A consequence is that retrieval is incremental whenever possible: when synchronizing with a data source, only changes since the last retrieval will be requested.

- Support of non-uniform data sources. A data source may be implemented by different versions of a software system. The code needs to detect the version of the data source API, and access the data in slightly different ways (for example, using different parameters in a call to the API) depending on the version detected[13]. *GrimoireLab* deals with these problems by hiding all data source API details, providing a unified API to data consumers, which can thus be written as modules independent from these details.

- Support of non-uniform data provided by data sources. Different versions of data sources, or even deployments of the same version, may provide different data details. For example, different instances of the same issue tracking system may configure states or additional fields to tickets or comments. In some cases those changes are significant for an analysis, but in some others they are not. *GrimoireLab* deals with this

[13] For example, there are many versions of Bugzilla, providing similar but not exactly equal APIs. One of these differences is the name of the parameter to order a list of issues by date: Last + Changed (prior to the 3.4 branch) or changeddate (starting with 3.4 branch). The difference is minimum, but enough to break the retrieval process if it uses the wrong name.

diversity by being agnostic with respect to the data provided by data sources: each item will be a Python dictionary with any needed structure, that can be flattened as a JSON document. If different versions of a data source produce different fields, they will just be mapped to the corresponding parts of the dictionary. This means that the code may be more generic, and more resilient to changes in data structure.

- Unified APIs and data formats. For building a toolbox, reusing code, and enabling users to reuse code, is fundamental. It is key to identify points in the usual pipelines where a single API can be provided, with independence of the data sources considered, and data formats that are common, specially when the code dealing with them can be reused. *GrimoireLab* does this in several areas. *Perceval* provides a common API to all data sources. The data produced by *Perceval* is always a dictionary, with some common fields useful for traceability (for example the data source, or when the item was retrieved), and can be dumped in a JSON document, so that it can be easily consumed. Elasticsearch provides a well known and documented API for storing and retrieving both raw and enriched indexes, which consumers can query for the JSON documents they need. *SortingHat* provides a common API for managing identities that can be used from any component. Data formats for raw and enriched indexes are defined with the same general structure, so that code for, for example, dealing with date formats or identities, can be made generic. Finally, enriched indexes are designed so they can be consumed directly by most visualization systems: we have tested them with Kibana, Grafana, and several Python-based visualization libraries.

- Separation of raw indexes and enriched indexes. This separation has proven very useful to isolate problems. Data retrieval problems are "encapsulated" in raw indexes. Once they have been solved, and a correct raw index has been produced, there is no need to come back to the original data source to compute anything: querying the raw index is enough. Querying the raw index instead of retrieving from the data source is also much more efficient, and does not cause stress in the infrastructure supporting the data source. Then, enriched indexes are simpler, and more convenient for visualization tools. Since they are flat, they can be queried more easily and efficiently than raw indexes, which need to maintain a nested structure close to the data in the data source. Enriched indexes are also enriched with summary metrics computed at the item level, but not present directly in the original data source. For example, from the number of lines added and removed by a commit to a file, the total number of added and removed lines can be pre-computed, or from the submission and closing time of an issue report, the time-open can be computed. Since these metrics are computed from data in the item, they are invariant as long as the raw item does not change.

- Aggregated metrics are, in general, not stored in the indexes, but computed by visualization or reporting tools when needed. This approach is the result of a tradeoff between the convenience of having all data (including aggregated data) readily available in the database, and the flexibility of allowing many different aggregations to be performed, and the easy addition of new data, at any point, with little overhead. On the one hand, having aggregated data readily available is convenient, because it can be

accessed and shown right away. However, we wanted a more flexible setup, where the consumer of the data may decide how to aggregate it. This is common with actionable visualization tools, where every time the user defines a different time period, or filters data in a different way, aggregations change. In addition, we wanted to add data to the database incrementally and frequently, as new data is available in the original data sources. Having pre-computed aggregations in the database forces to re-compute them every time new data is added, which means an overhead, especially large for large databases with many different aggregations. Fortunately, when the consumer wants some aggregation, it can usually be computed by adding it to the database query, which makes the operation efficient and relatively simple.

- Data merging. Even when data is retrieved from different data sources, an analysis may want to consider common aspects of all retrieved data. For example, a study on onboarding in a project may be interested in knowing about contributors to all repositories in all its data sources. Thus, including facilities for merging data is fundamental, and *GrimoireLab* recognizes it. Date fields are all converted to the same format, and have uniform names (eg, creation_date), so they are easily merged in, for example, a Pandas data frame, and in "person fields" such as the author as well. All items include fields to track the repository of origin, and fields for hierarchical organization of repositories. *SortingHat* provides a unified view of all contributors, by merging identities in different data sources, mapping all of them to the same uuid which is used in enriched indexes. When data sources are similar, common fields are identified, so that data can be merged and to some extent compared. For example, for issue trackers there are fields with opening and closing dates. Finally, the fact that data is stored in Elasticsearch may allow for merging directly when querying, such as "all data for all projects from date A to date B, authored by person P".

- Consumer-agnostic. It is very difficult to know how some person will exploit the data in the future. Since our approach is "retrieve and store", stored data may be useful even long after it was retrieved, or by a different team. *GrimoireLab* makes as few assumptions about analysis or visualization tools as possible. Enriched indexes are usually flat because then they are easier to import in visualization and analysis tools (for example, import into a Pandas data frame), but this is most of it. Thanks to this approach, we have tested the exploitation of data with several visualization tools, and the most common toolsets used for analysis. Anything that can consume JSON documents from Elasticsearch can be easily used to visualize or further analyze data produced by *GrimoireLab*.

- Performance and efficiency. When retrieving data from many repositories, performance is an issue. From one point of view, data sources should be used efficiently to avoid unnecessary stress. From another, we want the data as soon as possible. To address the first approach, *GrimoireLab* builds on an extensive research of data source APIs and retrieval options, to avoid those that could cause trouble. This is one of the reasons for using a toolset like this if you are not familiar enough with the details of data retrieval: avoiding causing unnecessary pain to the data source. Incremental access, and

storage of data in raw indexes to avoid recurrent retrievals are a consequence of this goal. To address the second aspect, *GrimoireLab* can do data retrieval and enrichment in parallel, with components such as *Arthur* and *Mordred* in charge of scheduling and coordinating concurrent activities. For several data sources, *GrimoireLab* allows also for the use of a pool of API rate tokens, when available and allowed, which may increase data retrieval speed.

- Identity management. Dealing with personal identities is a key aspect of many studies about software development. Even for studies as simple as counting contributors, you need to find out the several identities that the same person may be using. *SortingHat* deals with these issues, plus the problem of tagging people (for example, mapping them to the company they work for). *SortingHat* is no silver bullet: it just uses some heuristics, allows for the importing of identity data when it is available, and provides means (via command line, or via the *HatStall* web application) to manually merge and tag identities. Since *SortingHat* is integrated in the toolchain, changes reflected in its identities database are later reflected in any enriched index. A related problem to identities is privacy: in many cases, identities should not be provided to consumers of the data, to respect privacy of the persons participating in projects. Currently, there is on-going work in *GrimoireLab* to improve the situation in this area.

- Long-term performance. For dealing with the retrieval and analysis of large projects, the system must run for extended periods of time. During that time, failures occur: token rate exhaustion, network glitches, server reboots, even host shutdowns. *GrimoireLab* recovers as nicely as possible by retrying, and by continuing after failure. *Perceval*, *Arthur* and *Mordred* are designed to retry when the API rate provided by a token is exhausted, or when some network failures happen. For dealing with continuation, new runs of *GrimoireELK* and *Mordred* can be configured to be incremental, checking in the database the last items retrieved and enriched, and following from there.

- Data maintainability. Data retrieved and enriched should be easily inspected, so that people using it can detect errors. Errors can happen, for example, when the wrong repository is configured, or when timestamps are shifted due to misconfigured default timezones, or when bugs in the code produce some field with errors. If detected, errors should be fixed with minimum impact on the original data sources. *GrimoireLab* components store data in databases, so that errors can be fixed directly on them. For example, a bug due to incorrect parsing of a date format can be fixed by substituting wrong dates in enriched indexes from data in raw indexes. Some fields are also included in all items to assist traceability.

- Modularity. Since the kinds of consumption of data will be diverse, a toolset provides more flexibility, with many of its components being able to work on their own, but also in different combinations. Raw and enriched indexes provide good synchronization points for components in pipelines, and information hiding (such as configuration data in *Mordred* or identities data in *SortingHat*) helps to keep each problem domain within its own component, exposing only an API hiding unneeded details.

- Extensibility. In several aspects, *GrimoireLab* provides a vanilla system that can be easily extended to fit specific needs. The most clear cases of extensibility are:

  -New visualizations. *Kibiter* (or Kibana) allows for the creation of new visualizations, and arrangement of them in dashboards. All the process can be done via the graphical user interface, and only requires some knowledge about the data in Elasticsearch indexes, and some training on the Kibana user interface. New visualizations can be created from scratch, or by modifying those provided by *Sigils*. Both new visualization and *Sigils* visualizations can be mixed in dashboards. Once these visualizations and dashboards are created, they benefit from the data produced by the rest of *GrimoireLab*.

  -New indexes. *GrimoireELK* provides a simple mechanism for creating new enriched indexes: studies. A *GrimoireELK* study is a Python script, with a certain structure, that basically is fed with raw or enriched indexes, and produces a new index tailored for some specific analysis. *GrimoireLab* provides some of these studies, for example for analyzing the joining and leaving processes of a project (enrollment, abandonment, experience, etc.). These studies can be used as templates for producing new ones. Studies are run by *Mordred*, so that they are easily integrated in the data retrieval and analysis pipelines, and new visualizations can be produced for the indexes they produce.

  -New data sources. Supporting a new data source with *GrimoireLab* amounts to building some modules which integrate with the rest of the system. The process starts by building a new *Perceval* client, which will implement a Python generator that will retrieve data from the intended source, and produce dictionaries with a common structure. This client usually will automatically plug into the *Perceval* backend, producing JSON documents that will be stored in Elasticsearch by *Arthur* and *GrimoireELK*. Then, enrichment code has to be inserted in *GrimoireELK* so that enriched indexes can be produced, usually by selecting which fields from raw items should be copied, or transformed, into fields in the enriched index. If identities are to be managed, the appropriate calls to *SortingHat* will be included in this code too. Finally, new visualizations have to be produced in *Kibiter* to show the data in these new enriched indexes.

*GrimoireLab* has been evolving for several years, improving incrementally in dealing with all these issues. Unfortunately, many of them appeared while the system was already in use and evolving, which means that they needed to be addressed on the go, leading in some cases to not so clean solutions.

### *GrimoireLab* for researchers

In the specific case of researchers, *GrimoireLab* can contribute to solve some usual problems:

- Data retrieval software. For some kinds of data sources, writing some script to retrieve data is not difficult, if only some casual data is wanted. But retrieving large datasets in an

efficient way, with minimal stress on the mined infrastructure, it is not that easy. *GrimoireLab* provides a simple way of getting the data needed for a study in reasonable time, just by deploying it with the right configuration.

- Reproducibility. *GrimoireLab* helps reproducibility in two different ways, depending on the starting point of the reproduction study. To assist in full reproduction, starting by retrieving the data from data sources (maybe with new data in them), researchers can declare the version of *GrimoireLab* used, and its configuration files, in addition to the processing scripts. That would be enough for anyone to get the same data again, provided API of the data sources didn't change. If they did, it is likely that new versions of *GrimoireLab* adapted, so the retrieval can still be tried with those newer versions. For improved reproducibility, the complete list of *GrimoireLab* versions and dependencies can be provided, using common Python tools (for example, versions freeze of a virtual environment), or versions of Docker images. To assist on reproducibility from the retrieved dataset, raw and enriched indexes can be dumped, so that other researchers use them as their starting point, or compare them with those they get. This can be done at the *Perceval* level, producing mirrors of the relevant data sources, or by dumping data from Elasticsearch with customary tools like `elasticdump`.

- Collaboration in producing datasets. The use of *GrimoireLab* by different research groups allows the production of collaborative collections of data, resulting from merging the datasets produced by each group. The groups producing the collection should just use a common codification for repository identifiers, and *GrimoireLab* raw indexes can just be put together to produce a raw index for the collection. A "common codification" means using the same identifier for the same repository: for example, if they are accessible via different URLs, ensuring a common one is decided. Since all items in raw indexes have origin fields, with the identifier for the repository, if the same repository is more than once in the datasets, its items will clash. A simple policy, such as "if two items clash, use the most recent one" will allow for the production of a collection with no duplicated items. Metastudies can then easily run on the aggregated dataset.

 However, at least in some cases, new problems may appear:

- Bugs. Researchers using *GrimoireLab* become dependent on it producing data correctly. This makes their studies subject to bugs or errors in *GrimoireLab* or its configuration. Experience shows that it is more likely that a single researcher writing code makes mistakes causing bugs, than a software used in many different scenarios by many different people. *GrimoireLab* uses unit testing to prevent new bugs and regressions, with relatively high test coverage (*Graal*: 99%, *Perceval*: 98%, *SortingHat*: 93%, *GrimoireELK*: 82%, *Mordred*: 63%). But still, if there are bugs in some component, those could cause wrong data to entire datasets. Therefore, data checking should still be an important part of any research using *GrimoireLab*.

- Adaptability. If a study is designed as a function of what can be done with *GrimoireLab*, or what can be done easily with it, there is a risk of focusing on what the toolset can

do, and being limited by it. Researchers should confront the *GrimoireLab* model, designing new components when needed, or realizing when it is not well suited for a certain kind of study.

- Evolution and poor documentation. Even when *GrimoireLab* features extensive documentation, including a tutorial, and specific documentation for modules, it is not always easy to know what it can do, or how to tailor it to specific needs. This, together with the fact that *GrimoireLab* is for now a moving target, continuously evolving, may make it complex to use in some cases. Its community is already working on improving documentation, and keeping it updated, but still this may be an issue.

## Industrial use

*GrimoireLab* has been used for more than 5 years by a company to provide metrics as a service (with deployments tailored to the needs of customers) and consulting services based on software development metrics. To know about the main lessons learned after this experience, we conducted some conversations with key persons in the company (some of them co-authors of this paper). These are the main takeaways obtained from those conversations (some are specifically related to the tools, some other are more general of using metrics to track software development):

- Importance of automation and configuration. Since the deployment of *GrimoireLab* requires a relatively small effort, and is fully automated once configuration files are ready, most of the effort for running systems is in maintenance. And most of the maintenance tasks can be traced to external events (such as recovery from failures) or configuration changes (due to changes in the data sources to track, for example). When both cases are covered, maintenance of continuously working instances of GrimoireLab are negligible. To cover those cases, the company built scripts to react to common events and changes in configuration (most of them already a part of *Mordred* and other *GrimoireLab* components).
- Traceability of data items. In some cases, when specific items are checked, some error in them can be found. This could be tracked to errors in data collection, or more usually, in data enrichment. In any case, in those cases it is fundamental to be able to trace as much as possible how the item was produced, so that it can be linked to specific versions of the software, and to the corresponding log files. This is one of the reasons to keep detailed information in every item about the version of *GrimoireLab* used to produce it, and about the exact date when it was produced.
- Importance of details. Some details that in other environments could be considered as minor are not minor when people invested in the analyzed projects have access to the data. For example, the name of a person should be correct, even if during some time it was wrong in some repository. For example, all contributions of a person, or of all persons working of a company, in different data sources, have to be found in the final reports. Even if the impact in the final numbers is minor, errors in these cases diminishes the trust on the number provided in the reports, in general.

- Capturing complexity with metrics. Real software development is complex, and that complexity has to be captured when producing useful metrics about it. For example, even apparently simple metrics, such as number of bug reports still open are difficult to present at a given time spot, as a metric suitable for comparison with previous time spots, to learn if the situation is improving or not. First of all, bug reports have to be identified among all issues, which usually requires careful work with developers who label them. Then, important details need to be taken into account: impact of duplicates, impact of bots closing old bug reports, policies of closing bug reports without actually fixing them (for example, for difficult to reproduce cases), etc. If a certain project has less bug reports today than 1 month ago, it is important to know if the reason is a bot, running yesterday, and closing all issues older than six months. This kind of complexity happens everywhere: pair-programming makes many metrics based on code changes and code review difficult to apply; which Git branches to take into account may cause big differences, etc. Many of these cases can be dealt with options on how retrieval or enrichment works (for example, selection of branches, detection of bots closing issues, different metrics if pair-programming is used). Therefore, letting users filter the data the way they need, or configure the retrieval and analysis processes, has to be supported by the tools.

- No single metric characterizes software development. There is not a single metric, not even a small set of metrics, that can characterize software development in a given project. Depending on the goals, relevant metrics are different, depending on the project, how to compute those metrics is different. Therefore, every stakeholder in every project needs the flexibility to select what metrics to track, and to tune them to their needs. Tools, again, need to provide as much flexibility as possible for this.

- Explanation of metrics matter. In some cases, a specific set of metrics can capture the fundamental aspects of some process. But still, it is important to explain well why and how those metrics are a good characterization, and what happens when they change. For example, time-to-close, measured over the issues closed during the last month tells a different story than time-to-close, measured for the issues opened during the last month. Both are important, but both are radically different. Quite usually, when one of them goes up, the other may go down, and both things could be positive or negative depending on the goals of the project. If people are to engage with the metrics they need some training, or at least some explanation, of how they can use and understand the metrics in a way that helps them to understand the underlying trends and processes.

- The problem of gaming. If some metrics are used as a proxy for individual or collective performance, it is important to find incentives to avoid that gaming. And one of the best incentives comes from transparency: when anyone can see how your metrics are composed, and how are your individual (or corporate) contributions, it is much easier to have collective control over them. For this, the tool needs to be able to show not only aggregated metrics, but also to allow for drilling down, showing how aggregated metrics are composed.

**Table 6 Feature analysis of some related work.**

| | GrimoireLab | GHTorrent | GitHub Archive | Boa | Gitana | PyDriller | Metrics Grimoire | Kibble | SmartSHARK |
|---|---|---|---|---|---|---|---|---|---|
| Kind | Toolset | Dataset | Dataset | Dataset | Toolset | Toolset | Toolset | Toolset | Toolset |
| Data sources | 34 | 2 | 1 | 1 | 8 | 1 | 15 | 13 | 8 |
| Storage | Elastic, MySQL | MongoDB, MySQL | BigQuery | Hadoop | MySQL | None | MySQL | Elastic | MongoDB |
| Scope | Selected repos | GitHub complete | Complete gitHub | GitHub subset | Selected repos | Selected repos | Selected repos | Selected repos | Selected repos |
| Visualization | Yes | No | No | No | No | No | Yes | Yes | Yes |
| Reporting | Yes | No | No | No | No | No | Yes | Yes | No |
| Continuous | Yes | Yes | Yes | No | No | No | Yes | Yes | Yes |
| Raw | Yes | Yes | Yes | No | No | Yes | No | No | No |
| Processed | Yes | Yes | No | Yes | Yes | No | Yes | Yes | Yes |
| Identities | Manual heurist | GitHub | GitHub | No | Manual | No | Manual heurist, | No | Heurist |

Note:
The analysis is based on published literature, and in some cases, direct experience with the system. *Kind* shows if the system is specifically targeted at producing a dataset ("Dataset"), or intended as a generic toolset ("Toolset"). In *Data sources* we have considered different GitHub or GitLab APIs as different data sources, and code analysis as a single data source. In *Storage* we have considered the main storage systems, "Elastic" stands for Elasticsearch. In *Scope* "Selected repos" means that the user can select any collection of repositories for retrieval and storage (if applicable) from the set of supported data sources, "GitHub Complete" means that all of GitHub is retrieved and stored, "GitHub Subset" means that a pre-defined subset of GitHub repositories is retrieved and stored. *Visualization* and *Reporting* show if there are visualization and reporting components in the system. "Continuous" shows if the system can perform continuous retrieval. *Raw* and *Processed* shows if raw data (as obtained from the data source) and processed data (data with some processing more suitable for analysis) are available, stored, for further analysis. *Identities* show if identity management is included: "Manual" stands for "support for manual management", "Heurist." stands for "components performing heuristics for identity management", "GitHub" stands for "uses GitHub users API only".

# RELATED WORK

The idea of providing analytics for software development, which would help to track performance, and improve processes, is not new (see summaries of past experiences in (*Buse & Zimmermann, 2010*; *Menzies & Zimmermann, 2013*; *Zhang et al., 2013*)). Many companies also realized early the benefits of building their internal systems for performing software development analytics on the software they produced, as is shown for example in (*Czerwonka et al., 2013*). During the last decade, these ideas have been explored mainly in two areas: the creation (and in some cases, maintenance) of large datasets for researchers and practitioners, and the tools designed to assist in the creation and analysis of datasets about specific software repositories. In this section we discuss related work in both areas. As a brief summary of this discussion Table 6 provides a briefing about some features of some of the systems most comparable to *GrimoireLab*.

## Large datasets about software development

Among the large datasets about software development with their own software used to create and maintain them, we can differentiate two kinds: those specialized in source code, and those focused on other data related to software development (including metadata about source code changes, but usually not source code itself). The keystone systems of the second kind are:

- The *SourceForge Research Data Archive (SRDA)* (*Van Antwerp & Madey, 2008*) was the first dataset organized and maintained to allow researchers to have metadata about a large number of software development repositories. It provided reduced dumps of SourceForge, which was during most of the 2000s decade the preferred hosting site (software forge) for FOSS projects. At about the same time, FLOSSmole (*Howison, Conklin & Crowston, 2006*) retrieved data from SourceForge, and later other software development hosting sites, via their APIs, offering a wealth of metadata for researchers. FLOSSmole was to our knowledge the first system to perform a method, large-scale data retrieval via their API from large forges, making its data available to third parties. The kind of metadata they provide for each project include project description, project status, programming language, developers, license, programming language, and some general statistics about it.

- *FLOSSMetrics* (*Herraiz et al., 2009*) and *SQO-OSS* ("Software quality observatory for open source software" (*Gousios et al., 2007*)) had similar aims: to collect not only metadata about the projects, but as much data as possible about software development (the complete list of commits or issues, for example), via the APIs provided by software forges. Both produced their own software to that aim (*MetricsGrimoire* and Alitheia Core, mentioned below), which they used to collect data from hundreds of software repositories (commit records, issue reports, code review discussions, asynchronous communication via mailing lists, etc.), being some of the first demonstrators of how massive retrieval of data from software forges could be performed. Both worked with a diverse collection of data sources. FLOSSMetrics and SQO-OSS started the path towards automated collection of many different data kinds about software development, which is also the goal of *GrimoireLab*. Many of the features provided by *GrimoireLab* were first demonstrated, or at least set as a long-term goal, by those systems: retrieval of data from many different data sources; automated retrieval, storage, and analysis; fault-tolerance; massive collection and analysis; etc. The tools provided by *GrimoireLab* could be composed to produce systems similar to FLOSSMetrics and SQO-OSS, and *GrimoireLab* owes these systems the architecture based on storing data for further analysis (instead of having to retrieve it once and again from the original data sources), and the idea of incremental retrieval, fundamental for efficient retrieval of data from repositories already visited. In all of them, the basic idea was to interfere as little as possible with the infrastructure provided by the original data sources.

- *GHTorrent* (*Gousios & Spinellis, 2012*) and *GHArchive* (*Grigorik, 2022*) were developed later, focusing on the GitHub platform. Both work by querying the GitHub API to produce a complete dataset including most of the events noticeable in it (code commit records, issue reports, pull requests, changes in repository metadata). Both developed their own software for the retrieval and curation of the data. In the case of GHTorrent, curation includes linking actors of events to GitHub users, and adding metainformation provided by the GitHub repositories API (such as number of stars or programming languages used), that make the dataset more valuable. Given that both are focused on

GitHub, both have specialized components tailored to optimize the retrieval of data from the GitHub events channel.

In this respect, the GitHub backend of *Perceval* in *GrimoireLab* offers a similar functionality, but using the API for projects. In the end, both APIs (the projects API and the events channel) offer similar data, but with different perspectives. The former provides a log with all the data "as it is today": all issues and pull requests, with details about all comments, timing, etc, for example. The latter provides similar data "as it happens": as an issue or pull request is modified, a new event is received through the channel. The events API is therefore designed to be consumed as time passes (although it has some memory, of 300 events per end-point and not more than 3 months in the past). This means that the events channel needs to be parsed for all the projects of interest, and the information you cannot consume at a certain point, cannot be retrieved again. It also does not permit to retrieve data since the beginning of the activity for each project. The projects API, on the contrary, can be queried at any time, allowing to get most of the history pertaining to that repository at any point in time. *GrimoireLab* uses the projects API approach to be able to selectively retrieve data from any repository, at any point in time, since the project started. In addition, the *Perceval* backend for Git directly clones Git repositories, instead of relying on commit events produced by GitHub. Again, this allows it to get at any point in time most of the history of any given repository.

With respect to systems massively retrieving and archiving source code, and in some cases doing some kind of analysis on it, we can mention:

- *Sourcerer* (*Bajracharya et al., 2006*) was designed to be a search engine for source code. It retrieved code from source code repositories, analyzed it, and produced a database designed to be queried. In some sense, it can be considered the ancestor of systems massively archiving source code and allowing queries on it.

- *Boa* is a system to massively collect source code, analyze it, and store some of its characteristics in a database, allowing researchers to query on them. Boa also provides a programming language (also named Boa), supported by an infrastructure, to ease queries and studies about large quantities of source code repositories. In its first incarnation (*Dyer et al., 2013b*, *Dyer et al., 2015*) it extracted metadata and source code from SourceForge (a popular software development forge at that time), storing it in mirrored Subversion repositories and a Hadoop cluster, which is where the queries were actually run. It was later complemented with a massive number of repositories from GitHub and other sources. Boa was designed to let users create simple programs that allowed for a quick and comprehensive exploration of all the data.

- *Debsources* (*Caneill, Germán & Zacchiroli, 2017*), *Software Heritage* (*Pietri, Spinellis & Zacchiroli, 2019*) and *World of Code* (*Ma et al., 2019*) are aimed to retrieve and archive source code, for different purposes. Debsources maintains a dataset with all the source code from Debian packages, including some metadata about it. Software Heritage retrieves data from source code management systems, with the goal of preserving it and making it available in the long term. Their goal is to store all publicly available

source code. World of code is specifically targeted at maintaining a database with all FOSS code, which researchers can use to investigate the global properties of FOSS. All of them are designed to be updated with new data, incrementally and automatically.

*GrimoireLab* allows source code retrieval and analysis, via *Graal*. But instead of being designed for archiving all the source code, its approach is to analyze it, computing the desired metrics of each version of each file found, and storing them in a database, which can later be queried. In this respect, this is in part similar to what Sourcerer or Boa do, but the focus of *Graal* is not to produce a specific kind of analysis, but to allow any third party tool, run via a simple plugin module, to produce data that will be stored in the database. Both Software Heritage and World of Code can also be queried via a database, but in the case of Software Heritage this is only for finding specific source code, while in the case of World of Code, the approach is to provide data about the interdependence of projects. Both are very different from the modular approach used in *GrimoireLab*, which allows for studies at many scales (from the single repository to the many tens of thousands of them), and for many different purposes.

*MetricMiner* (*Sokol, Aniche & Gerosa, 2013*), *SeCold* (*Keivanloo et al., 2012*), and *OpenHub* (*Farah, Tejada & Correal, 2014*) are systems that offered a mix of source code analysis and data about software development. In that respect, their goals are more similar to those of *GrimoireLab*, although their structure and implementation are aimed to produce specific outcomes, very different from the flexible toolbox approach of *GrimoireLab*.

## Tools for analyzing software repositories

With respect to tools capable of retrieving and analyzing software repositories, there are many of them:

- *SoftChange* (*German & Mockus, 2003*) and *GlueTheos* (*Robles, González-Barahona & Ghosh, 2004*) were some of the first tools that were designed to retrieve data from source code repositories (CVS, and later Subversion), with the aim of being generic enough to be useful in more than a single study.
- *gitdm* (*Corbet, 2008*) and *PyDriller* (*Spadini, Aniche & Bacchelli, 2018*) are good examples of more modern reimplementations of these tools. Both are focused on the analysis of Git repositories. gitdm was developed to analyze the Linux kernel Git repository, and produce some simple stats for it. It has been later adapted to produce other metrics of interest to other projects. PyDriller is a recent framework for retrieving and analyzing Git repositories. It allows for sophisticated options to decide which parts of a repository to analyze, and is easily extensible to produce different kinds of metrics.
- *MetricsGrimoire* (*Gonzalez-Barahona, Robles & Izquierdo-Cortazar, 2015*) was to our knowledge the first toolset for retrieval of data from several kinds of software development repositories (source code management, issue tracking, code review, mailing lists discussions, etc.) conceived to be reusable outside the team producing them. *MetricsGrimoire* in fact included in its maintenance team some volunteers external to

the original research team that produced them, and were maintained over a period of about a decade, being used by many different research teams. *GrimoireLab* owes to it the design as a toolset capable of retrieving data from several kinds of repositories, with tools that can work alone or in combination. However, the coordination of tools in *MetricsGrimoire* was very shallow, not even using compatible formats for the same information. *GrimoireLab* went further in this approach, allowing for a much complete integration of data coming from different data sources, with for example same fields for actors in the database, and a single system (*SortingHat*, which was originally built as a part of *MetricsGrimoire*) for managing identities in exactly the same way. *GrimoireLab* also learned from these systems the problems of trying to define a specific schema for all incoming data, avoiding these problems by using flexible, nested JSON-based formats in the data-retrieval stage, while enforcing flat, uniform formats for enriched data.

- *Alitheia Core* (*Gousios & Spinellis, 2009*) was built to support SQO-OSS, and was one of the first systems designed to continuously retrieve data from different kinds of software repositories, via their APIs. It was also a system composed as a toolset, although tools were specifically designed to work together, not in isolation. One of its features was its continuous operation, well suited for automatic retrieval and update, which was also one of the design goals of *GrimoireLab*, although in this case via a specific component, *Mordred*, that orchestrates the rest of the tools.

- Some other tools that allowed for the retrieval and relatively simple analysis of software development repositories are: *CODEMINE* (*Czerwonka et al., 2013*), *BuCo Reporter* (*Ligu, Chaikalis & Chatzigeorgiou, 2013*), Gitana (*Cosentino, Izquierdo & Cabot, 2018*), and *Candoia* (*Tiwari, Upadhyaya & Rajan, 2016*).

- *PROM* (*Rubin et al., 2007*), *FOSSology* (*Gobeille, 2008*), *Qualipso* (*Del Bianco et al., 2009*), *FRASR* (*Poncin, Serebrenik & Van Den Brand, 2011*), and *Q-Rapids* (*Martinez-Fernández et al., 2018*) are systems or tools designed to produce some specific higher level metrics (e.g., processes and quality metrics), which require more data processing. For this, they had their own components that allowed for the retrieval and analysis of data (*MetricsGrimoire* in the case of Qualipso), and in general, stored the resulting metrics in a queryable database.

- There are many other tools providing different kinds of comprehensive views of software development. Some of them are: *Complicity* (*Neu et al., 2011*), *RepoGrams* (*Rozenberg et al., 2016*), *CROSSMINER* (*Bagnato et al., 2017*), *Kibble* (*Apache, 2022*), *SmartSHARK* (*Trautsch et al., 2017*), *Augur* (*Goggins, 2022*), etc.

When compared to these tools, *GrimoireLab* is in general more diverse in terms of data sources supported with a common interface: all of them can be retrieved using the *Perceval* API. For all of them raw and enriched indexes are produced with a similar structure, all of them can be retrieved and analyzed automatically with the same *Mordred* configuration. Most of the tools mentioned above support one, or a small number, of different data sources, and in general are not designed to produce uniform data that can be

later queried for further analysis in a uniform way. *GrimoireLab* also provides identity management for all of these data sources, and a common way of visualizing and reporting data. However, the main difference is probably the fact that *GrimoireLab* tools can be used, if needed, in isolation, and that the data is stored in a way that allows for many different kinds of further analysis.

## AVAILABILITY AND USAGE

*GrimoireLab* is a free, open source software toolset, hosted in GitHub, as a part of the *CHAOSS* project. Each tool is maintained in a separate repository, using Git for source code management. This guarantees future availability, via the Software Heritage repository[14], or GitHub itself. The version described in this paper corresponds to the current latest commits for each repository, as of March 1st 2021 (see a detailed list in the companion package).

The system can be installed as a collection of coordinated Python packages, from Pypi, by running a single command which pulls as dependencies all *GrimoireLab* modules:

```
pip install grimoirelab
```

Modules which can be used on their own (*Perceval, Graal, SortingHat, GrimoireELK, Mordred, Kidash*) provide a driver program that can be run directly. Some of them may need some services to work (Elasticsearch, Kibana, MariaDB, Redis), which can be deployed locally or somewhere on the Internet.

Several Docker container images are also provided to run pre-configured versions of *GrimoireLab*, with all services already pre-installed. They can produce complete dashboards, with raw and enriched data for all repositories, just by running the container with the appropriate configuration data. They can also be used with official container images for services, via `docker-compose` (see the companion dataset, described in Section "A companion package and other information", for an example of running the toolset this way, including configuration files). Docker images for *GrimoireLab* are stored in DockerHub, so that they can be recovered later (for any *GrimoireLab* release). They are also produced from Dockerfile configuration files, publicly available from *GrimoireLab* repositories.

Extensive documentation is provided with each of the components, a tutorial is also available with step by step details for running the system as a whole, and with many specific scenarios.

An active community of developers is collaborating in the maintenance and extension of the system, led by a company which is using it for its core services. The toolset is being used by several groups, FOSS foundations, companies and individuals not related to their authors. Some of these uses have lead to research publications, most of them by other groups (*Zhao et al., 2017*; *Claes et al., 2017*; *Mens, Adams & Marsan, 2017*; *Robles et al., 2017*; *Devanbu et al., 2017*; *Claes et al., 2018a*; *Claes et al., 2018b*; *Kuutila et al., 2018*; *Claes, Mantyla & Farooq, 2018*; *Izquierdo et al., 2019a*; *Izquierdo et al., 2019b*; *Robles, Gamalielsson & Lundell, 2019*; *Sulun, Tuzun & Dogrusoz, 2019*; *Itkin, Novikov & Yavorskiy, 2019*; *Orviz Fernandez et al., 2020*; *Butler et al., 2020*; *Ashraf et al., 2020*; *Claes &*

[14] Software Heritage: https://softwareheritage.org

*Mantyla, 2020*; *Kuutila, Mantyla & Claes, 2020*; *Sulun, Tuzun & Dogrusoz, 2021*).
*GrimoireLab* deployments include *Cauldron*, a SaaS platform which allows users to select repositories to be analyzed via a web interface, producing data in Elasticsearch which is shown in custom Kibana dashboards.

## CONCLUSIONS

In this paper we have presented *GrimoireLab*, an extensible and modular open source toolset which offers (i) automatic and incremental data gathering from a large set of tools used in software development, (ii) storage and enrichment of the retrieved data, (iii) identities management, and (iv) data visualization and reporting to allow inspecting specific aspects of software development. *GrimoireLab* relies on different components that can be used together or standalone. It also may help researchers to enhance reproducibility of their studies, and traceability of their data. *GrimoireLab* reimplements and extends previous approaches to create a mature platform, currently used in commercial and academic settings.

The main characteristics of *GrimoireLab* which make it unique when compared to other tools to analyze software development repositories are:

- Support of many different data sources (close to 30).
- Flexibility and configurability of the tool, even for large-scale analysis (10,000 s of repositories).
- The storage model, with raw data mimicking the original API, kept for further analysis, and enriched, identity-merged data for visualization.
- Identity merging is unique (or at least at the level of the best of other tools).
- Easy deployment as a complete system using Docker or docker-compose, with a single command, but at the same time can be custom-installed using pypi packages.
- It can be used as a complete toolset, but most of its tools can be also used by themselves, as modules, integrated with user-implemented software.
- Tested in real-world (both industrial and research) cases, including systems running continuously for months, retrieving data from thousands of repositories.

*GrimoireLab* is managed as an open free, open source software project, with a public roadmap, and all contributions managed through pull requests in GitHub. The project documents how to contribute to it, and in fact some important contributions (such as partial support for some data sources) have been received by the core team of developers. Researchers and developers of any kind are welcome to propose their patches fixing errors or providing new features.

To improve the usability in different scenarios, we are working in community repositories to share configuration files and widgets tailored to specific analysis. We also intend to extend *GrimoireLab* in different directions. First, we plan to support graph data storage to allow the user to answer questions that cannot be easily addressed with a document-based database (*Kaur & Rani, 2013*). Second, we will speed up the data enrichment using in-memory data processing libraries (*McKinney, 2011*). Finally, we

would like to improve the retrieval of data buried in source code (*Cosentino et al., 2018*), helping researchers to perform cross-cutting analysis over a wider spectrum of software development data.

## COMPANION PACKAGE AND OTHER INFORMATION

A companion package for this paper is available[15]. It includes data, logs, and configuration files for the IoT and the GNOME GitLab cases, and a list of Software Heritage identifiers for the software components presented in this paper. The full source code of *GrimoireLab*, tutorials about it, and other information is also available[16].

### Funding

This work is supported by Ministerio de Ciencia y Tecnología of Spain under Project BugBirth, RTI2018-101963-B-I00 (Retos) and Grimoire as a Service, RTC-2017-6554-7 (Retos Colaboracion), and by Ministerio de Economia y Competitividad of Spain under Grant PTQ-15-07709 (Torres Quevedo). There was no additional external funding received for this study. The funders had no role in study design, data collection and analysis, decision to publish, or preparation of the manuscript.

### Grant Disclosures

The following grant information was disclosed by the authors:
Ministerio de Ciencia y Tecnología of Spain: RTI2018-101963-B-I00, RTC-2017-6554-7.
Ministerio de Economia y Competitividad of Spain: PTQ-15-07709.

### Competing Interests

Santiago Dueñas, Daniel Izquierdo-Cortazar, Luis Cañas and Alberto Pérez García-Plaza are employees of Bitergia, the main contributor to GrimoireLab.

### Author Contributions

- Santiago Dueñas performed the computation work, authored or reviewed drafts of the paper, and approved the final draft.
- Valerio Cosentino conceived and designed the experiments, performed the experiments, analyzed the data, performed the computation work, prepared figures and/or tables, authored or reviewed drafts of the paper, and approved the final draft.
- Jesus M. Gonzalez-Barahona conceived and designed the experiments, performed the experiments, analyzed the data, performed the computation work, prepared figures and/or tables, authored or reviewed drafts of the paper, and approved the final draft.
- Alvaro del Castillo San Felix conceived and designed the experiments, performed the experiments, analyzed the data, performed the computation work, authored or reviewed drafts of the paper, and approved the final draft.
- Daniel Izquierdo-Cortazar performed the computation work, authored or reviewed drafts of the paper, and approved the final draft.

[15] Companion package for this paper: https://doi.org/10.5281/zenodo.4656469

[16] *GrimoireLab* main website: https://grimoirelab.github.io

- Luis Cañas-Díaz performed the computation work, authored or reviewed drafts of the paper, and approved the final draft.
- Alberto Pérez García-Plaza performed the computation work, authored or reviewed drafts of the paper, and approved the final draft.

### Data Availability

The companion package is available at Zenodo: "Companion package for GrimorieLab: A toolset for software development analytics" https://doi.org/10.5281/zenodo.4656469

The source code for GrimoireLab is available at GitHub, in several repositories, linked from https://chaoss.github.io/grimoirelab/#components.

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
