# Peer review of "GrimoireLab: A toolset for software development analytics"

_PeerJ Computer Science, doi:10.7717/peerj-cs.601_

## Round 0.1 · original submission · Major Revisions

Overall the reviewers are quite positive about the paper and the work. All reviewers as well appreciate the contributions of the authors to the community!

Nevertheless, the reviewers raise a good number of concerns that are mostly related to the presentation, writeup and flow of the paper content. Also, reviewer 3 pointed out concerns about the installation not working and the need for a replication package. Please address these concerns and prepare a detailed rebuttal that explains how you went about addressing these concerns.

·

Basic reporting

Here are my concerns:

- Introduction. First Paragraph. "They showed how data retrieval..."; "They explored the
48 limits to scalability"; etc. -> Who is the subject that the authors want to refer to when using "They"? "The papers cited? Previous Work? Make clear this aspect in the paragraph. Maybe the entire paragraph can be rephrased.

- Introduction. It would be nice to have the aim of the paper described more clear. From my perspective, the goal is to present the infrastructure and tools, to provide architectural details about it, and also some "usage scenario - proof of concept" to validate the architecture. Making the aim clear, the authors can better "set" the manuscript expectations for the reader since during the manuscript there are a lot of technical terms and examples. The goal can be better defined in the abstract as well.

- GHArchive Grigorik (NA) - ???, SARA (NA) - ???? - It seems that some references are missing. Please, revise all the references in the manuscript.

- Figures need to be improved. The first eight figures are large with large icons. Figures 9 to 14 have some colors that cannot be visualized by colorblind persons. The different types of icons used are not explained in the text. I also recommend improving the image quality (some parts are blurred).

- 1.3 Contributions. -> The fact of having an open-source community around the Grimoire Labs, would be stated as a contribution?

- Section 2.1 and 2.4 -> Must have a general explanation about the data retrieval component as we have for sections 2.2 and 2.3.

- Lines 133 to 143 must be described together with the Perceval explanation (line 125).

- Section 4 - Table 4. "Retrieval (github) 10:02:12 hours 160 items/min" -> Itens means, pull request/issues and their messages? Commit Metadata? - Please clarify in the text.

- EXAMPLES OF METRICS AND VISUALIZATIONS Section -> The authors provided examples and explanations about metrics and visualizations available. My concern here is: Is it possible to create new metrics and visualizations? How can the users extend it? It would be worth having some instructions and explanations.

- Ethical issues. Recently, GHTorrent lead with some ethical issues (The #issue32 incident) - https://ghtorrent.org/faq.html. I think that this topic deserves some special discussion in this paper. Is there any concern/issue/implication for researchers/practitioners when using GrimoireLab?

- The paper description has too many bullets (e.g Related Work, Components description, Discussion, conclusions, etc). I suggest reviewing the manuscript to avoid this practice that breaks the text fluency.

Minor
- Line 343 - GrimoireLabis - GrimoireLab is
- Line 378 - Bugzilla timing9): -> Exclude )
- The term "experiment" is used to describe the case studies/scenarios

Experimental design

No comments. The manuscript describes the Metrics Grimoire infrastructure. The infrastructure is validated with a ``Proof of concept'' presented by describing some usage scenarios.

Validity of the findings

No comments. The validation was described by presenting some usage scenarios, by providing the replication package and access to the tools and source code.

Additional comments

The paper presents GrimoireLabs, a toolset composed of five main components, supporting about 30 different kinds of data sources related to software development. The toolset has been tested in many environments, for performing different kinds of studies, and providing different kinds of services. The components and infrastructure are presented. The authors used some usage scenarios to validate the presented toolset.

In general, I liked the paper idea. As a Researcher, I also would like to thank the authors for their effort to built this set of tools and the infrastructure. A scenario replication is available, which is really good for practitioners and researchers. However, the paper needs to address presentation issues (image quality, too many bullets, etc). It is also worth to include discussions about some potential Ethical issues and Metrics/Visualization extension.

Check my comments to improve the paper manuscript.

Reviewer 2 ·

Basic reporting

Comments about basic reporting are done in the "General comments for the author" field.

Experimental design

Comments about experimental design are done in the "General comments for the author" field.

Validity of the findings

Comments about the validity of the findings are done in the "General comments for the author" field.

Additional comments

# General comments

This paper presents GrimoireLab, a set of components to data retrieval and analysis of software repositories. The paper describes the structure and functionality of the main components of GrimoireLab (data retrieval, data storage, identities management, analytics, and orchestration). The paper presents real-world use cases of the proposed components, as well as examples of metrics that can be calculated and visualizations that can be generated by GrimoireLab. The paper also provides a discussion of challenges that need to be addressed to collect and analyze software engineering data in both the industrial and academic settings. The strongest point and greatest contribution of the paper is a set of components that were developed during many years of research effort and have matured as open source projects. As for the weakness associated to this paper, it is possible to highlight the lack of validation of the claims performed as discussion points. Below I provide specific comments for the overall paper presentation of for each section of the paper.

# Overall presentation

In general, the paper is adequately written but a few clarifications are needed, which I specifically point in the detailed comments for each section. Regarding presentation, there are two major issues.

- The first issue is the excessive usage of pronouns with an unclear reference (i.e., antecedent noun), which yields too many ambiguous sentences. For example, in the sentence “They showed how data retrieval …” [P2,L46], the antecedent noun is not clear (does “they” refer to repositories or tools?). Occasionally, the antecedent noun can be inferred from the context, however this is not always the case. Other examples include: “They explored the limits of scalability …” [P2,L47], “They demonstrate …” [P2,L48], “Some of them …” [P2,L56], “Most of them …”[P2,L63], “They use …”[P2,L64], “… of how it …” [P3,L105], “… that is can provide.” [P3,L106], “… contained in it.” [P8,L239], “… from them …” [P8,L243], “… these cases …” [P8,L245], “… visualize it.“ [P9,L277], “… all of these scenarios …” [P11,L304], “.. some of them,” [P11,L308], “… its analysis starts,” [P12,L318], “… all its items …” [P12,L318], “… depending on it.” [P18,L405], “… deals with this …” [P18,L411]. Many other similar cases appear throughout the paper.

- The second issue is the occurrence of vague descriptions of rather important information. For example, in the sentence “When the data to be collect is really large …”, it is hard to judge what “really large” is. For such technical descriptions, the text should be as precise as possible. Other examples include: “… some idea …” [P11,L312], “… for a little while …” [P12,L319], “… every few minutes.” [P14,L336], “… some data …” [P14,L338], “… some custom Kibana visualizations …” [P14,L351], “… key data …” [P14,L357], “… some exceptions …” [P16,L365], “… some fields …” [caption of Figure 19], “… in some cases …” [P20,L504], [P20,L507], “… some other goodies …” [P21,L561], “… some if its properties.” [P21,L577].

- Typos: “… we will us …” instead of “… we will use …” [L111], “… HatStallThen,” instead of “… HatStall. Then,” [P9,L277], “Data collection for “git” Includes …” instead of “Data collection for git includes …” [P12,L324], “GrimoireLabis …” instead of “GrimoireLab is …” [P14,L343], “… more indexes is produced …” instead of “… more indexes are produced …” [P16,L364], “… the should fixed …” instead of “they should be fixed …” [P19,L26], “… kinds consumption…” [P19,L480] instead of “… kinds of consumption …”, “This make their …” [P20,L508]

- [P20,L494-502] The usage of the word “you” in that context is discouraged and inconsistent with the rest of the paper.

- Unclear sentences: “… consult analysis and visualizations of it” [P2,L56], “… specifically to retrieve source code or data” [P2,L59], “Interlaced with this process, SortingHat process identities found.” [P12,L320], “For many data sources, this is software not difficult to write …” [P19,L26], “… plugged to the data.” [P19,L447]

- Citation format: all citations are in the format “Author et al. (Year)”, while in many cases the correct format seems to be “(Author et al., Year)”

# Introduction

Main point: The motivation for another system to collect and analyze data from software repositories is not well described. Although the Introduction reviews relevant literature, there is no clear description of the motivation for a study on another related system. For example, while the Introduction says “In many cases they are not easy to reply and operate, and in others they are difficult to use for large-scale, continuous data retrieval. Not all of them provide support for retrieval, storage and analysis of the data …”. It is not clear which of these limitations GrimoireLab addresses. Also, there is no description of how and why the approach taken by GrimoireLab to tackle those limitations is better than the approach taken by other systems that also tackle these limitations. Most importantly, the Introduction describes no empirical evidence that GrimoireLab can overcome previous’ systems limitations. I would suggest having a table that compares the features and functionalities of the related systems compared with the features and functionalities of GrimoireLab.

[P1,L42] The definitions of “… tools and complete systems …” it is not clear what the difference is. Is GrimoireLab being proposed as a tool or complete system?

[P2,L49] The sentence “… different approaches to avoid harming the project hosting systems …” needs clarification. What are those approaches and what type of harm are they trying to avoid?

[P2,L75] Although the types of repositories whose data collection is supported by GrimoireLab are defined in later sections, it would be interesting to define (or perhaps give examples of) different types of repositories in the Introduction.

[P2,L80-82] Is the claim of efficiency validated in the paper? It would be nice to describe how efficient is GrimoireLab compared with other solutions that are mentioned in the Introduction.

[P2,L84] It would be nice to motivate the identity management module upfront in the Introduction. When “module for identity management (…) in combination with custom code to merge or tag identities” is mentioned for the first time, it is hard to understand what this module does. Also, if this feature is unique for GrimoireLab, this should be highlighted.

[Section 1.3] The list of contributions reads more like a list of features of GrimoireLab. Beyond GrimoireLab itself (which is a great contribution), I would expect to see other contributions in terms of 1) novel techniques to investigate the efficiency and efficacy of GrimoireLab in coping with the described challenges, or 2) novel findings or results of applying GrimoireLab in the field, 3) or data that allows the comparison between GrimoireLab and related systems. Moreover, Section 1.3 seems like a right place to thoroughly describe the differences and similarities between GrimoireLab and the related systems.

[Section 1.4] Some definitions appear after their first usage (e.g., data source, kind of data source). I would suggest having a structured glossary in this section, with a formalization of all definitions used in the paper. For example, the definition of “kind of data source” is imprecise, as the data sources given as examples have a notable different API. Also, readers that are less familiar with mining of software repositories would benefit from having a definition of related terms.

# Section 2

Main points: 1) There is no discussion of the rationale behind the existence of each component. The paper describes the functionality of each component. To some degree, the relationship between these components is also described. However, except for the identity management component, it is unclear what challenges each component solves, both in terms of structure (e.g., why is this specific architectural design chosen?) and functionality (e.g., how a component overcome the current limitations of data collection and analysis?). It is also unclear how GrimoireLab addresses the flaws (in terms of design, efficiency, or efficacy) of related systems. 2) Given the paper's descriptive characteristics, I would recommend having a formal notation (e.g., UML) to describe the components and their relationships (Figures 1-7 and Figures 10-14). For example, the meaning of the relationship shown between components (Figures 1-7) is not defined (does the arrow represent data flow? functional dependency? or something else?). Also, although some colour and shape code is used in Figures 9-14, the meaning of such codes is unclear. 3) There is no discussion regarding the separation of concerns among components.

[P3,L104] Could one interpret the sentence "... describes the structure of GrimoireLab and its module ..." as "... describes GrimoireLab's architecture ..."? If so, which architectural level the paper describes? Also, the text interchangeably mixes the words "structure" and "design". Please, clarify if they have the same intended meaning.

[P3,L128-131] "Graal runs third party tools on git repositories.", "... runs a collection of tools on checkouts ...", "... produced by those tools ...". The mentioned tools need to be clearly defined, at least what are their outputs.

[P5,L150-159] Please, clarify that are the "common features" that are reused to collect data from different data sources and which components implement those features.

[P5,L164] "... controlling the details of the job ...". Which are the controlled details and why?

[Perceval and Graal] Perceval and Graal components seem to have a strong coupling. Please, clarify how they are typically integrated.

[Section 2.3] The motivation for the identity management component is well described. However, I would recommend addressing the following points: 1) There are many academic papers describing different heuristics for identity disambiguation (e.g., resolving users that adopt different e-mail addresses to commit to the codebase). Different heuristics are associated with different performance (e.g., accuracy). How does GrimoireLab's approach to identity disambiguation compare with related work in this area? What is the performance of GrimoireLab's approach? 2) "... sends them back to be added to the enriched data." [P6,L202] is not clear in Figure 7. Also, by Figure 7 it is unclear how "enriched data" relates to other sub-components of SortingHat. In general, the textual description of the figures needs to be improved.

[Section 2.4] Although the section states Visualization and Analytics, what is actually described relates to visualization and data exporting. Also, please clarify which "documents" are referenced in [P7,L213] and what "actionable inspection" [P7,L216] means in this context.

[Enriched indexes] The concept of "enriched index" needs to be clarified and better formalized.

[Figures] There is excessive usage of Figures that hinders readability. While I appreciate the effort to describe the relationship between GrimoireLab's components incrementally, many of the figures are redundant and can merge into a single one that conveys the same message. Also, the textual description of many figures (e.g., Figures 4 and 8 ) is lacking.

# Section 3

Main point: this section lacks a detailed description of the techniques used by each component. Also, it seems that a large portion of the section's contents overlaps with the previous section's contents (in particular, most of the section describes the relation between components and how they individually work). Please, consider improving the separation between the sections.

[Section 3.1] What are the advantages of using Perceval compared to having a script that uses CURL with pagination, for example? Also, does Perceval collects only issues and pull-requests from GitHub? If that is the case, the paper needs to discuss this issue as a limitation of GrimoireLab. More generally, although the system can collect data from more than 30 data sources, not all data from each source can be collected.

[P8,L242] Section 3.1 describes a data retrieval scenario and claims that "... which will use Perceval (as library) in the background, to clone the repository and get the list of commits, and then run third party tools that analyze each relevant file in each commit to get complexity metrics from them in a single JSON ". It seems that Perceval is doing both data retrieval (e.g., cloning repositories and getting commits) and data analysis (e.g., calculating complexity metrics). I wonder how much this characteristic of GrimoireLab (specifically, Perceval) is based on best practices. For example, why does the same component perform retrieval and analysis? As these are different concerns, it seems that different components should address them. Another question raised by the quoted sentence is how "relevant files" are set in Perceval.

[P9,L276] In the sentence, "SortingHat will work based on heuristics ...", what are those heuristics and how well do they perform?

[P9,L282-284] The technique adopted by GrimoireLab for incremental data retrieval needs to be way better described. For example, it is unclear from the sentence "... which can be used to select which items from the database need to be retrieved." [P10,L283] how such items are selected. Also, the sentences "... how often will data sources be visited for incremental retrieval." [P10,L287] and "... refresh periods (how often data will be retrieved incrementally from repositories)" [P10,L290] gives the impression that the incremental data retrieval is not real-time but instead performed in batches. If that is the case, how this feature compares with real-time techniques such as processing an event stream?

[P10,L294] The paper says that "... Graal analyzes source code, by running third party tools with the help of Perceval.". From Figure 2, it is possible to verify that Graal is within the "data retrieval" component. This observation evidences that the paper lacks a discussion on the separation of concerns of the components. Also, there is no illustration of the relationship between Graal, third-party tools, and Perceval in any of the figures.

# Section 4

Main points: I am surprised by how much Section 4 overlaps with prior sections. It claims to describe “three real use cases for GrimoireLab”, while the previous section claims that “GrimoireLab can be used in many different scenarios. In this section we describe some of them …”. Please, consider improving the separation of the contents described in Sections 2, 3 and 4. It seems that Section 4 has the potential to describe insightful empirical studies regarding GrimoireLab. However, there is no definition of research questions, no description of rigorous methods to perform such empirical studies, and no discussion of the learned lessons. The lack of a clear and well-thought study design leads to severe issues to the validity of the claims.

[Section 4.1] The main takeaway from the analysis presented in this section does not stand out. For example, how does the running time change by changing the associated factors (what are the factors in the first place)? Also, what is the baseline for comparison of the presented results? How good or bad are the presented results compared with other similar systems? What are the strengths and limitations?

[P11,L312] How is GrimoireLab’s performance evaluated?

[P11,L314] The sentence “The main figures of that experiment are included in Table 1”. I do not understand what “figures” means, neither which “experiment” is described. In addition, what is Mordred in the sentence "The experiment was run by Mordred"?

[P12,L319] "... sleep for a little while ...". This sentence is imprecise. Also, how much of the "little while" is reflected in the results shown in Table 2.

[P12,L320] "The data shown in this paper ...". Please, clarify which data.

[P12,P324] "Data collection for git includes cloning of git repositories for GitHub, and production of the raw index by analyzing those clones". How each of these steps is reflected in the results shown in Table 2?

[Table 3] There is no reference of Table 3 in the text

[Section 4.2] What is the main takeaway of Section 4.2?

[P14,L329] “Table 4 shows data about an example of large deployment of GrimoireLab”. What does “large deployment” mean?

[P14,L335-336] “The deployment has been running continuously for more than four years, retrieving data incrementally from repositories every few minutes”. How many minutes?

# Section 5

[P16,L365-367] “… metrics are not (except for some exceptions) a part of the indexes stored in Elasticsearch: they are computed either by the visualizations, or by the tools producing the reports.”. What is the rationale behind this design decision? Why is it better recalculating the metrics instead of storing them?

# Section 6

Main point: this section needs to improve the discussion of the novel learned lessons after years of research and development of GrimoireLab. For example, prior studies discuss the same pointed issues: “minimizing interactions with data sources”, “data source APIs are not uniform”, and “data provided are not uniform”. How good or bad are the solutions proposed by GrimoireLab? More importantly, which findings and evidence support the claims made in this section? Without clearly providing evidence to support the discussion, most of the claims are understood as speculation. Another important point is that, in general, there is a good amount of discussion about what the system does, but barely any discussion on how and why.

Some excerpts in this section deserve clarifications. Below I point to specific excerpts:

[P18,L405] “… do some different tricks depending on it”. Please, clarify what those “tricks” are and provide the motivation for using them. Preferably, use more specific and technical wording.

[P18,L414] “… the code may be more generic …”. It is unclear which “code” is being referenced.

[P19,L450] “… to address the first approach …”. Please, clarify which “first approach”, as the previous sentence was talking performance issues of data collection.

[P19,L450] “… to avoid those that could cause trouble.” Please, clarify what type of trouble. Also, please, prefer a more technical and specific wording.

[P19,L470-471] “GrimoireLab recovers as nicely as possible by retrying, and by continuing after failure”. Please, elaborate on how this feature is implemented. Is there any increase in the waiting time after retrying? How each type of failure is handled? Is the failure handling approach dependent on the data source?

[P19,L475-476] How can consumers detect errors in the retrieved data? What are those errors? What are the most common ones?

[P19,L477-478] What is a “bug in time interpretation”?

[P20,L494-502] Reproducibility: why providing a reproducible build of GrimoireLab instead of providing a dump of previously collected data? The argument that “that would be enough for anyone to get the same data again, provided data sources didn’t change” does not hold, as rarely a data source will not change.

# Section 7

[P20,L503-506] I do not understand what this paragraph conveys. For example, what does “production of merged collections” mean? Please, clarify the sentence “The availability of origin fields simplify the merge, allowing for complex combinations, such as using raw indexes for different data sources to compose a meta study”.

# Section 8

[P20,L530-533] How the two proposed categories of related work arose? Is this classification by the own authors? What process was used to achieve this classification?

[P20,L541-542] “… not only metadata about the projects, but as much data as possible about software development …”. Please, clarify what are the differences between “metadata about the projects” and “data about software development”.

[P21,L563-567] I appreciate the comparison between GrimoireLab and GHTorrent (same for the paragraph of [P21,L591-599]). While I agree that using the projects API will require a reduced number of queries to the endpoint, the events API (as used by GHTorrent) provides a richer source of data about the activities performed in a repository. Please, clarify which type of data can be obtained by using the projects API in comparison with the events API and vice-versa.

[P21,L576-577] “Boa is a system …” [L576] and “Boa is a programming language …” [L577]. What is Boa anyways?

[Lack of subsections] To improve readability, I would recommend separating Section 8 into sub-sections (per related work’s theme).

[P22,L628] Please, define the acronym “SQO-OSS”.

# Section 9

I appreciate seeing a mature and well-supported set of tools being provided to the academic and industrial communities of mining software repositories. Many of the provided tools are tested in-field and used to support different types of research, which clearly shows the value of those tools. As a side note, it would be a great addition to this paper if any lessons learned from the usage of GrimoireLab by the core services of a company could be discussed, even if the discussion stems from public information.

# Section 10

[P23,L685] “… an industry-strength platform …”. What evidence is given for this claim?

[P32,L687-699] This excerpt claims unique characteristics of GrimoireLab that differentiates these tools from others. However, in the list of characteristics, we can read “support of different data sources”, “flexibility and configurability”, “identity merging”, “tested in real-world”, etc. Unfortunately, none of these characteristics are “unique” to GrimoireLab and can be observed in other systems or academic papers. Perhaps, the unique feature of GrimoireLab is how all these characteristics are put together, which reinforces the need to present a strong comparison between GrimoireLab and other existing related systems.

·

Basic reporting

Overall the manuscript is well structured. The writing is clear with only few syntax, grammar, or spelling mistakes. I recommend carefully re-reading the references to fix some capitalization errors.

The numerous figures provided are helpful. I suggest using a standardized notation (such as UML) or explaining the meaning of the various used shapes (e.g. rectangle, vs ellipse, vs, rounded rectangle. Also, I recommend using a lighter color for the purpse shapes, which are difficult to read.

Extend Table 2 to include the time it took for each phase to run.

Suffixing the duration with hours/min in Table 3 is confusing. Provide all durations simply in HH:MM:SS units.

Table 3: Clarify what "items" refers to.

I recommend rephrasing the abstract's "Conclusions" section to state what GrimoireLab has achieved, rather than summarize the paper.

Experimental design

According to the framework provided by Stol and Fitzgerald (2018), this is solution-seeking rather than a knowledge-seeking study. Solution seeking studies aim to solve practical problems for which solutions to practical problems can be engineered. In such studies, researchers design, create, or develop solutions for a software engineering challenge; in this case by building the GrimoireLab tools. As such, I do not expect a research question filling an identified knowledge gap. Instead, I expect to see description of a tool advancing the state of the art.

The described system (GrimoireLab) advances the state of the art by supporting more data sources and analyses as well as enhanced interoperability. The study can benefit from a rigorous comparison between GrimoireLab and other existing systems in terms of performance, supported data sources, and analyses.

Two case studies, one on IoT repositories and one on Wikimedia Foundation repositories illustrate the power of GrimoireLab. These can be profitably extended to showcase what research questions can be answered through the provided quantitative data.

Validity of the findings

The provided conclusions mostly follow the conducted study. They can be extended through more rigorous benchmarking and more ambitious example studies.

The authors provide links to a companion package on Zenodo, which contains data, logs, and configuration files for the IoT case. I recommend that this should be expanded to include the Wikimedia foundation case, and also additional data regarding the recommended additions. Uploading a source code snapshot on Zenodo can ensure the source's long-term survival.

The provided online documentation is quite detailed and well-written.

I was able to install and run Percical using the provided instructions. On the other hand, the installation of Grimoirelab appeared to get stuck in a loop with lines such as the following.

Requirement already satisfied: setuptools in ./venv/lib/python3.7/site-packages (from importlib-metadata->flake8>=3.7.7->graal==0.2.3->grimoirelab) (50.3.2)
Requirement already satisfied: wheel in ./venv/lib/python3.7/site-packages (from importlib-metadata->flake8>=3.7.7->graal==0.2.3->grimoirelab) (0.36.1)
Requirement already satisfied: setuptools in ./venv/lib/python3.7/site-packages (from importlib-metadata->flake8>=3.7.7->graal==0.2.3->grimoirelab) (50.3.2)
Requirement already satisfied: wheel in ./venv/lib/python3.7/site-packages (from importlib-metadata->flake8>=3.7.7->graal==0.2.3->grimoirelab) (0.36.1)

The denormalized output of the grimoirelab tools seems to result in considerable waste. For example, each commit contains, again and again, the full details of its author and committer: about 49 elements. What is the rationale of this decision and what is its effect in terms of waster storage and processing cost?

Additional comments

L. 36: Your phrasing focuses on the process. Consider stating that tools provide data about the software development process and the developed artifacts.

L. 91: The described attributes refer to coverage breadth, rather than exensibility.

L. 129: Generally, running tools on a commit's checkout is needlessly expensive. Accessing directly Git's file-system is more efficient, because it avoids the cost of copying all files to the OS's file-system.

L. 244: I find it a pity that Graal uses Perceval internally, rather than allowing the user to compose various tools through a common protocol and data format.

L. 254: Consider providing an actual example, rather than an abstract description. How could one e.g. measure the evolution of comment spelling mistakes over time?

L. 490: I recommend backing up this claim with benchmark results.

L. 855: One citation to the smartshark ecosystem should be enough.

I recommend clarifying in the manuscript following questions

- How does GrimoireLab handle GitHub pull requests?
- What is the schema of the data provided by Perceval?
- Do SortingHat and HatStall offer an help to address GDPR requirements? How do you propose that the tools should be employed for handling them?
- L. 234 Perceval appears to retrieve the repository's metadata and also a clone of the actual repository. How does it handle clashes of repository names between repository names from different sources?
- How can GrimoireLab be extended? What does it take to create a new data source or analysis metric? How can such additions be contributed back to the community?
- What does an index hold? What operations does it facilitate?
- What are the practical limits of GrimoireLab in the number of repositories or data volume it can process? On what volumes has it been tested? What support is offered for running GrimoireLab on multiple hosts?
- How is GrimoireLab tested (c.f L. 508)? Are there some metrics (e.g. code coverage) that can help track the quality of its testing? Where do these stand today?
- How are the provided GrimoireLab Docker images supported and updated? Given GrimoireLab 's evolution how can GrimoireLab users ensure the long-term replicability of their studies?

---

## Round 0.2 · Minor Revisions

Thanks for the revision! The reviewers are quite happy with the revision and appreciate your work to address their concerns. However, they had some minor comments. Please address them in an updated minor revision.

Ahmed

·

Basic reporting

I suggest proofreading the article. Some typos need to be fixed before publication. It is probably because of the "diff" document, but please, revise accordingly.

Just some examples:
Line 390 -> GrimoireLaballows - GrimoireLab allows
Line 301 -> Percevaland - Perceval and
Line 426 and 428 -> Gerrit efficiency) - remove a bracket
In this section we discuss - ","
Table 6 need to be formatted as the other tables

Experimental design

Section 2 was improved to better describe the proposed software architecture.

Validity of the findings

The authors properly improved the examples and usage scenario. Also, the replication package was updated.

Additional comments

I am grateful for the authors 'efforts to reorganize the article and address it according to the reviewers' suggestions. I have no more suggestions for improving the article.

Reviewer 2 ·

Basic reporting

I thank the authors for their effort in addressing all comments of the major revision. The two main problems related to the presentation were thoroughly addressed, the description and motivation of the Grimoirelab’s components were improved, and a new case study to evaluate Grimoirelab’s performance was presented. Some minor specific points are still unclear. Below I describe each of these specific points:

1. The authors improved the motivation for Grimoirelab, in specific by adding Table 6 and a sentence to the introduction. The sentence in [L114-119] still lacks a clarification of how Grimoirelab “improve the situation in solving the practical problem of retrieving data from software development repositories, preparing it for further analysis, and providing basic analysis and visualization tools help in exploratory studies”. Is this due to the holistic approach taken by the tool? In the description of the features shown in Table 6, please clarify the meaning of the “selected projects” scope. Does the “collection of projects” need to be in git repository? What is the difference with respect to “GitHub subset”?
2. If possible, would be preferable to adopt a consistent notation for the arrows throughout the figures that explain Grimoirelabs’ architecture. For example, the dashed arrow is used to represent invocations in Figure 6, workflow in Figure 7, and “entities flow” in Figure 10.
3. I believe that a brief description of the “conservative approach” to identity management, as well as the examples given in the response letter, would be of great aid to readers.
4. Typos:
a. [L230] missing closing parenthesis.
b. [L390] “GrimoireLaballows”, “Percevaland”
c. Caption of Figure 10: “sold arrows”
d. [L731] “the the”
5. Sentences that could have grammar revision: [L156-157] “… and may need some other components to work, which in that cases are installed as dependencies of it.”

Experimental design

Please, see the comment in "basic report". The new version of the manuscript improves the experimental design by adding a new case study to evaluate the performance of the tool.

Validity of the findings

Please, see the comment in "basic report".

·

Basic reporting

See previous review.

Experimental design

See previous review.

Validity of the findings

See previous review.

Additional comments

The authors have diligently addressed almost all the review comments. As a result, the paper now clarifies many important issues. I do not think that another review round is required. However, the authors might want to take the following comments into account when revising their manuscript.

The figures are now more readable. The Perceval class diagram is useful,
but I don't think that UML class diagrams are the most appropriate for
illustrating the other tools. I would still have preferred to see
them follow a UML notation (e.g. that of collaboration diagrams) rather than
the invented ad hoc ones, but I think that this choice can be left to the
authors.

The system's performance is now nicely illustrated in Table 5. The numbers
for enriched issues and merges may indicate a pathology regarding I/O
operations; you might want to investigate why clock time is two orders of
magnitude higher than processor time only for these specific two use cases. Could caching or better indexes help?

Table 6 now provides a very useful summary of related systems. Consider
adding the corresponding references below the headings or in a separate row at the end.

Line 898,899: Consider rephrasing as "GrimoireLab provides a simple way
of getting the data needed for *a* study in reasonable time"

---

## Author Rebuttal · Round 0.2

# Response to the review of the submitted paper
## *GrimoireLab: A Toolset for Software Development Analytics*

We thank the reviewers for their detailed comments. We think we have addressed all of them, modifying the paper accordingly. In this document we explain how we have addressed each comment.

Please note that we have restructured the paper, so the sections in the previous version don't necessarily correlate with the new ones. In particular, we have put together in Section 3 (Combining the modules) all three exemplary research scenarios (subsections 3.1-3.3) and all three real use cases (3.4-3.6), including a completely new use case (the GNOME GitLab case, in 3.6), to provide more detailed data about performance. In Section 4 (Discussion) we are including the description of the main features of GrimoireLab, some notes for researchers using it, and some lessons learned from industrial usage. In Section 5 we present related work, including a new table for comparison with some other systems. The rest is as it was.

This document is structured as follows: first, we address comments by the editor. Then, we address comments by each of the reviewers, in the same order that they were mentioned in the review we received. For each comment, we include the comment itself, and then, under "Answer", how we have addressed it.

We have also produced a diff document, using latexdiff, as requested in the re-submission instructions. However, for making it work, we had to delete some tables (signaled as such in the diff), and change the bib format of all citations to the right one (thus, this difference is not highlighted in the document). Due to reorganization of sections and subsections, not all text highlighted as a change is really a change, in some cases it corresponds to text moved.

# Answers to Editor

## Comment E.1

Overall the reviewers are quite positive about the paper and the work. All reviewers as well appreciate the contributions of the authors to the community!

Nevertheless, the reviewers raise a good number of concerns that are mostly related to the presentation, writeup and flow of the paper content. Also, reviewer 3 pointed out concerns about the installation not working and the need for a replication package. Please address these concerns and prepare a detailed rebuttal that explains how you went about addressing these concerns.

**Answer to E.1**

He have tried to improve presentation, writeup and flow of the content. We have also tried to be more precise in our claims, or to provide evidence when it was missing. We are including a detailed rebuttal in the rest of this document.

In the specific case of the installation not working, we have included an answer stating that we cannot reproduce the problem. We would be more than happy to interact with the reviewer so that we can reproduce, and hopefully fix, the problem.

We have complete the companion package with detailed data about the new use case, and with detailed data for reproduction of the specific version of GrimoireLab components that we're using.

# Answers to Reviewer 1

## Comment R1.1 *Introduction. First Paragraph*

"They showed how data retrieval..."; "They explored the limits to scalability"; etc. → Who is the subject that the authors want to refer to when using "They"? The papers cited? Previous Work? Make clear this aspect in the paragraph. Maybe the entire paragraph can be rephrased.

### Answer to R1.1

The subject is "the tools and systems". We've tried to clarify that, rewriting (in part) the paragraph:

> *"These tools showed how data retrieval, storage, and at least a part of the analysis could be automated and made generic enough to support different kinds of studies; were used to explore the limits to scalability, and the benefits of developing reusable tools; and served to demonstrate different approaches to avoid harming the project hosting systems"*

## Comment R1.2 *Introduction*

It would be nice to have the aim of the paper described more clear. From my perspective, the goal is to present the infrastructure and tools, to provide architectural details about it, and also some "usage scenario - proof of concept" to validate the architecture. Making the aim clear, the authors can better "set" the manuscript expectations for the reader since during the manuscript there are a lot of technical terms and examples. The goal can be better defined in the abstract as well.

### Answer to R1.2

We completely agree with this suggestion. We have included a new paragraph, in subsection "Structure of the paper and definitions":

> *"This paper presents the results of a "solution-seeking" research line, aiming to improve the situation in solving the practical problem of retrieving data from software development repositories, preparing it for further analysis, and providing basic analysis and visualization tools that help in exploratory studies. The main outcome of this line is the GrimoireLab toolset. Therefore, this paper aims to present GrimoireLab, providing some details about it; to show how it could be used in some research scenarios, and how it was used in some real use cases; and to discuss its main characteristics both in research and industrial environments."*

## Comment R1.3 *References missing*

GHArchive Grigorik (NA) - ???, SARA (NA) - ???? - It seems that some references are missing. Please, revise all the references in the manuscript.

**Answer to R1.3**

This is due to those references corresponding to web pages. Since they don't really have a publication date, we used "NA" (not applicable) for the year. The references are in the list of references, and are linked from the text.

We would be happy to change the format of those references to reflect the policies of the journal for citation of these artifacts.

We revised all references, hopefully not having errors in this version.

## Comment R1.4 *Figures need to be improved*

The first eight figures are large with large icons. Figures 9 to 14 have some colors that cannot be visualized by colorblind persons. The different types of icons used are not explained in the text. I also recommend improving the image quality (some parts are blurred).

**Answer to R1.4**

We made smaller figures 3-8, since we agree there is nothing lost with making them smaller. For figures 1-2, we're keeping its size, since we're a bit worried that reducing their size would mean worsening readability. We have also reduced the size for most figures 9-14, for the same reasons. We hope the size is more balanced now. However, we would happily resize any of them if that's still considered convenient

We improved image quality by using vector-based PDF files, instead of pixel-based JPEG files, which we were using, for all images for which it was possible (all of them, except for screenshots).

The different icons for figures 1-7 are only for (hopefully) helping with the readability. Same for the new 9-14 figures (see below).

We have changed figures 9-14 to avoid using colors, thus hopefully increasing readability for colorblind persons.

## Comment R1.5 *1.3 Contributions*

The fact of having an open-source community around the Grimoire Labs, would be stated as a contribution?

**Answer to R1.5**

When writing the paper, we decided that the fact that there is an open source community around GrimoireLab is not a contribution by itself, and we mentioned that in the section Availability and Usage, as "An active community of developers is collaborating in the maintenance and extension of the system, leaded by a company which is using it for its core services." However, we would be happy to mention that as a contribution if reviewers advise us to do it.

## Comment R1.6 *Section 2.1 and 2.4*

Must have a general explanation about the data retrieval component as we have for sections 2.2 and 2.3.

**Answer to R1.6**

Thanks for pointing this out. We have added this text at the beginning of 2.1:

*"GrimoireLab pipelines usually start by retrieving data from some software development repository. This involves accessing the APIs of the services managing those repositories (such as GitHub, Stack Overflow or Slack), or using external tools or libraries to directly access the artifacts (such as git repositories or mailing list archives). In the specific case of source code management repositories, some tools may also be run to obtain metrics about the source code. For large-scale retrieval, work is organized in jobs that have to be scheduled to minimize impact on the target platform, and to maximize performance. GrimoireLab provides three components for dealing with these issues:"*

## Comment R1.7 *Lines 133 to 143*

Lines 133 to 143 must be described together with the Perceval explanation (line 125).

### Answer to R1.7

Done, by moving those lines to the end of the Perceval explanation.

## Comment R1.8 *Section 4 - Table 4*

"Retrieval (github) 10:02:12 hours 160 items/min" $\rightarrow$ Items means, pull request/issues and their messages? Commit Metadata? - Please clarify in the text.

### Answer to R1.8

Clarified, by adding "Items are GitHub issues and pull requests" in the caption of the table.

## Comment R1.9 *Examples of metrics and visualizations Section*

The authors provided examples and explanations about metrics and visualizations available. My concern here is: Is it possible to create new metrics and visualizations? How can the users extend it? It would be worth having some instructions and explanations.

### Answer to R1.9

This is a good point. We have included a new item, "Extensibility", in the discussion section, explaining the more usual cases of extensibility:

*"Extensibility. In several aspects, GrimoireLab provides a vanilla system that can be easily extended to fit specific needs. The most clear cases of extensibility are:*

- *New visualizations. Kibiter (or Kibana) allows for the creation of new visualizations, and arrangement of them in dashboards. All the process can be done via the graphical user interface, and only requires some knowledge about the data in Elasticsearch indexes, and some training on the Kibana user interface. New visualizations can be created from scratch, or by modifying those provided by Sigils. Both new visualization and Sigils visualizations can be mixed in dashboards. Once these visualizations and dashboards are created, they benefit from the data produced by the rest of GrimoireLab.*

- *New indexes. GrimoireELK provides a simple mechanism for creating new enriched indexes: studies. A GrimoireELK study is a Python script, with a certain structure, that basically is fed with raw or enriched indexes, and produces a new index tailored for some specific analysis. GrimoireLab provides some of these studies, for example for analyzing the joining and leaving processes of a project (enrollment, abandonment, experience, etc.). These studies can be used as templates for producing new ones. Studies are run by Mordred, so that they are easily integrated in the data retrieval and analysis pipelines, and new visualizations can be produced for the indexes they produce.*

- *New data sources. Supporting a new data source with GrimoireLab amounts to building some modules which integrate with the rest of the system. The process starts by building a new Perceval client, which will implement a Python generator that will retrieve data from the intended source, and produce dictionaries with a common structure. This client usually will automatically plug to the Perceval backend, producing JSON documents that will be stored in Elasticsearch by Arthur and GrimoireELK. Then, enrichment code has to be inserted in GrimoireELK so that enriched indexes can be produced, usually by selecting which fields from raw items should be copied, or transformed, into fields in the enriched index. If identities are to be managed, the appropriate calls to SortingHat will be included in this code too. Finally, new visualizations have to be produced in Kibiter to show the data in these new enriched indexes."*

## Comment R1.10 *Ethical issues*

Recently, GHTorrent lead with some ethical issues (The #issue32 incident) - `https://ghtorrent.org/faq.html`. I think that this topic deserves some special discussion in this paper. Is there any concern/issue/implication for researchers/practitioners when using GrimoireLab?

### Answer to R1.10

Quite an interesting topic, indeed. We have added this text at the end of the subsection on the components for identity management:

*"Most identities found in software repositories can be considered as personal information, therefore subject to laws protecting privacy, and to ethical guidelines on the matter. Due to this circumstance, in some cases identity management can deemed unethical, or unlawful (for example, under GDPR, if there is no clear legitimate interest for the processing of personal information, and it is considered that there is no informed consent from identity holders). To have this situation into account, GrimoireLab allows for the pseudoanonymization of identities as they are retrieved, via configuration switches in Perceval and GrimoireELK. If those switches are activated, Perceval hashes identities found in retrieved data, and GrimoireELK does not use SortingHat, producing raw and enriched indexes with pseudoanonymized identities. When orchestration is used, switches are activated with an option in the Mordred configuration file."*

## Comment R1.11 *Paper description*

The paper description has too many bullets (e.g Related Work, Components description, Discussion, conclusions, etc). I suggest reviewing the manuscript to avoid this practice that breaks the text fluency.

**Answer to R1.11**

We have merged several sections, and the description of the paper now is (hopefully) more fluid.

## Comment R1.12 *Minor*

- Line 343 - GrimoireLabis - GrimoireLab is

- Line 378 - Bugzilla timing9): -¿ Exclude )

- The term "experiment" is used to describe the case studies/scenarios

**Answer to R1.12**

All of them fixed, thanks.

## Comment R1.13 *Experimental design*

No comments. The manuscript describes the Metrics Grimoire infrastructure. The infrastructure is validated with a "Proof of concept" presented by describing some usage scenarios.

**Answer to R1.13**

None needed.

## Comment R1.14 *Validity of the findings*

No comments. The validation was described by presenting some usage scenarios, by providing the replication package and access to the tools and source code.

**Answer to R1.14**

None needed.

## Comment R1.15 *Comments for the author*

The paper presents GrimoireLabs, a toolset composed of five main components, supporting about 30 different kinds of data sources related to software development. The toolset has been tested in many environments, for performing different kinds of studies, and providing different kinds of services. The components and infrastructure are presented. The authors used some usage scenarios to validate the presented toolset.

In general, I liked the paper idea. As a Researcher, I also would like to thank the authors for their effort to built this set of tools and the infrastructure. A scenario replication is available, which is really good for practitioners and researchers. However, the paper needs to address presentation

issues (image quality, too many bullets, etc). It is also worth to include discussions about some potential Ethical issues and Metrics/Visualization extension.

Check my comments to improve the paper manuscript.

**Answer to R1.15**

Thanks a lot for your comments. We hope you will find appropriate how we have addressed them.

# Answers to Reviewer 2

## Comment R2.1 *General comments*

This paper presents GrimoireLab, a set of components to data retrieval and analysis of software repositories. The paper describes the structure and functionality of the main components of GrimoireLab (data retrieval, data storage, identities management, analytics, and orchestration). The paper presents real-world use cases of the proposed components, as well as examples of metrics that can be calculated and visualizations that can be generated by GrimoireLab. The paper also provides a discussion of challenges that need to be addressed to collect and analyze software engineering data in both the industrial and academic settings. The strongest point and greatest contribution of the paper is a set of components that were developed during many years of research effort and have matured as open source projects. As for the weakness associated to this paper, it is possible to highlight the lack of validation of the claims performed as discussion points. Below I provide specific comments for the overall paper presentation of for each section of the paper.

### Answer to R2.1

Thanks for the detailed comments, which in our opinion were helpful to improve the paper. See below how we have tried to address them.

## Comment R2.2 *Overall presentation*

In general, the paper is adequately written but a few clarifications are needed, which I specifically point in the detailed comments for each section. Regarding presentation, there are two major issues.

### Answer to R2.2

Thanks for the detailed comments. All issues answered below.

## Comment R2.3 *First issue*

The first issue is the excessive usage of pronouns with an unclear reference (i.e., antecedent noun), which yields too many ambiguous sentences. For example, in the sentence "They showed how data retrieval ..." [P2,L46], the antecedent noun is not clear (does "they" refer to repositories or tools?). Occasionally, the antecedent noun can be inferred from the context, however this is not always the case. Other examples include: "They explored the limits of scalability ..." [P2,L47], "They demonstrate ..." [P2,L48], "Some of them ..." [P2,L56], "Most of them ..."[P2,L63], "They use ..."[P2,L64], "... of how it ..." [P3,L105], "... that is can provide." [P3,L106], "... contained in it." [P8,L239], "... from them ..." [P8,L243], "... these cases ..." [P8,L245], "... visualize it." [P9,L277], "... all of these scenarios ..." [P11,L304], "... some of them," [P11,L308], "... its analysis starts," [P12,L318], "... all its items ..." [P12,L318], "... depending on it." [P18,L405], "... deals with this ..." [P18,L411]. Many other similar cases appear throughout the paper.

### Answer to R2.3

We have hopefully fixed all of these cases, and some others that we detected.

## Comment R2.4 *Second issue*

The second issue is the occurrence of vague descriptions of rather important information. For example, in the sentence "When the data to be collect is really large ...", it is hard to judge what "really large" is. For such technical descriptions, the text should be as precise as possible. Other examples include: "... some idea ..." [P11,L312], "... for a little while ..." [P12,L319], "... every few minutes." [P14,L336], "... some data ..." [P14,L338], "... some custom Kibana visualizations ..." [P14,L351], "... key data ..." [P14,L357], "... some exceptions ..." [P16,L365], "... some fields ..." [caption of Figure 19], "... in some cases ..." [P20,L504], [P20,L507], "... some other goodies ..." [P21,L561], "... some if its properties." [P21,L577].

**Answer to R2.4**

We have done our best to make all of these cases more concrete, and others that we found while reviewing the text. Thanks for pointing out these important details. In some cases, we clarified the description, in some others we rewrote the text to make it more clear.

## Comment R2.5 *Typos*

"... we will us ..." instead of "... we will use ..." [L111], "... HatStallThen," instead of "... HatStall. Then," [P9,L277], "Data collection for "git" Includes ..." instead of "Data collection for git includes ..." [P12,L324], "GrimoireLabis ..." instead of "GrimoireLab is ..." [P14,L343], "... more indexes is produced ..." instead of "... more indexes are produced ..." [P16,L364], "... the should fixed ..." instead of "they should be fixed ..." [P19,L26], "... kinds consumption..." [P19,L480] instead of "... kinds of consumption ...", "This make their ..." [P20,L508]

**Answer to R2.5**

Thanks for pointing out these typos. We fixed them, and reread the paper fixing a few others.

## Comment R2.6 *[P20,L494-502]*

The usage of the word "you" in that context is discouraged and inconsistent with the rest of the paper.

**Answer to R2.6**

Thanks. We rewrote the sentence as:

> *"When information about the data sources (identifiers for repositories) was codified in a uniform way, a raw index for a collection of projects is obtained by just putting together raw indexes for all the projects in the collection."*

## Comment R2.7 *Unclear sentences*

"...consult analysis and visualizations of it" [P2,L56], "... specifically to retrieve source code or data" [P2,L59], "Interlaced with this process, SortingHat process identities found." [P12,L320], "For many data sources, this is software not difficult to write ..." [P19,L26], "... plugged to the data." [P19,L447]

**Answer to R2.7**

Thanks for pointing this out. We have tried to clarify all the sentences (in some cases, by correcting errors or modifying the sentence, in some others, by including an explanation).

## Comment R2.8 *Citation format*

All citations are in the format "Author et al. (Year)", while in many cases the correct format seems to be "(Author et al., Year)"

**Answer to R2.8**

Thanks. We were using the wrong LaTeX command. Fixed.

## Comment R2.9 *Introduction*

Main point: The motivation for another system to collect and analyze data from software repositories is not well described. Although the Introduction reviews relevant literature, there is no clear description of the motivation for a study on another related system. For example, while the Introduction says "In many cases they are not easy to reply and operate, and in others they are difficult to use for large-scale, continuous data retrieval. Not all of them provide support for retrieval, storage and analysis of the data ...". It is not clear which of these limitations GrimoireLab addresses. Also, there is no description of how and why the approach taken by GrimoireLab to tackle those limitations is better than the approach taken by other systems that also tackle these limitations. Most importantly, the Introduction describes no empirical evidence that GrimoireLab can overcome previous' systems limitations. I would suggest having a table that compares the features and functionalities of the related systems compared with the features and functionalities of GrimoireLab.

**Answer to R2.9** *Answer*

We have tried to make it more clear that we're not framing our paper as a study on a system to collect and analyze software repositories, but as the result of a "solution-seeking" research line. We have tried to make it more clear in the introduction. We also didn't intend to claim that GrimoireLab is better than other systems in terms of features or functionality, but on its holistic approach to all the problems, and in its maturity and field-test. We have tried to make that more clear in the discussion, in the description of the toolset, and in other parts of the paper.

Thanks for the suggestion to have a table comparing some features and functionalities with other systems. We have included that table (Table 6) in the related work section.

## Comment R2.10 *[P1,L42]*

The definitions of "... tools and complete systems ..." it is not clear what the difference is. Is GrimoireLab being proposed as a tool or complete system?

**Answer to R2.10**

We changed that part, to mention only "tools", since we realized there was really no difference between "tools" and "systems" in this context. We have only kept "toolset" when referring to GrimoireLab to reflect their structure as a set of components that can work together.

## Comment R2.11 *[P2,L49]*

The sentence "... different approaches to avoid harming the project hosting systems ..." needs clarification. What are those approaches and what type of harm are they trying to avoid?

### Answer to R2.11

We clarified with the following text:

> *"(for example, by retrieving data once, storing it in a database, and later analyzing that data as many times as needed)".*

However, we're commenting a bit more on this later, in the discussion.

## Comment R2.12 *[P2,L75]*

Although the types of repositories whose data collection is supported by GrimoireLab are defined in later sections, it would be interesting to define (or perhaps give examples of) different types of repositories in the Introduction.

### Answer to R2.12

Done, mentioning some of those kinds of data sources:

> *(source code management, issue tracking, code review, messaging, continuous integration, etc.)*

## Comment R2.13 *[P2,L80-82]*

Is the claim of efficiency validated in the paper? It would be nice to describe how efficient is GrimoireLab compared with other solutions that are mentioned in the Introduction.

### Answer to R2.13

We have tried to show how it is efficient in the analysis of the use cases. In particular, we have added the GNOME GitLab use case to provide a detailed analysis of how long did it take to analyze an relatively large set of repositories, and how the time it took is reasonable, given the limitations of the data sources. Unfortunately, we have not found data, in papers describing comparable systems, that allow us for a comparison.

## Comment R2.14 *[P2,L84]*

It would be nice to motivate the identity management module upfront in the Introduction. When "module for identity management (...) in combination with custom code to merge or tag identities" is mentioned for the first time, it is hard to understand what this module does. Also, if this feature is unique for GrimoireLab, this should be highlighted.

### Answer to R2.14

We agree with the convenience of briefly introducing why the the identity management is useful. We have included this text in the introduction:

> *"GrimoireLab also includes a module for identity management that can be used in combination with custom code to merge or tag identities, something that is fundamental to analyze activity of persons using several identities, to merge activity from different data sources, and to annotate identities with affiliation information, for example."*

We have also included a row in Table 6 showing which other systems have, to our knowledge, some identity management.

## Comment R2.15 *[Section 1.3]*

The list of contributions reads more like a list of features of GrimoireLab. Beyond GrimoireLab itself (which is a great contribution), I would expect to see other contributions in terms of 1) novel techniques to investigate the efficiency and efficacy of GrimoireLab in coping with the described challenges, or 2) novel findings or results of applying GrimoireLab in the field, 3) or data that allows the comparison between GrimoireLab and related systems. Moreover, Section 1.3 seems like a right place to thoroughly describe the differences and similarities between GrimoireLab and the related systems.

**Answer to R2.15** Answer

We have tried to make the differences and similarities with other systems in Section 5 (Related Work), including Table 6, with a summary of them.

We don't claim to use novel techniques to investigate efficiency, although we have included a detailed use case (GNOME GitLab, in Subsection 3.6, results in Table 5, with all relevant data in the companion package) to show performance results.

We have also included a new subsection, 4.3, with lessons learned after its industrial use.

## Comment R2.16 *[Section 1.4]*

Some definitions appear after their first usage (e.g., data source, kind of data source). I would suggest having a structured glossary in this section, with a formalization of all definitions used in the paper. For example, the definition of "kind of data source" is imprecise, as the data sources given as examples have a notable different API. Also, readers that are less familiar with mining of software repositories would benefit from having a definition of related terms.

**Answer to R2.16**

We have structured the definitions as a list of terms, and (hopefully) clarified the examples, so that the terms themselves are also more clear. We would be happy to add any other term that may seem convenient to add. We have added also definitions for "index" and "item", which seem to cause some confusion.

## Comment R2.17 *Section 2. Main points (1)*

There is no discussion of the rationale behind the existence of each component. The paper describes the functionality of each component. To some degree, the relationship between these components is also described. However, except for the identity management component, it is unclear what challenges each component solves, both in terms of structure (e.g., why is this specific architectural design chosen?) and functionality (e.g., how a component overcome the current

limitations of data collection and analysis?). It is also unclear how GrimoireLab addresses the flaws (in terms of design, efficiency, or efficacy) of related systems.

**Answer to R2.17**

We have tried to ad some rationale in several parts, mainly in the description of the components, and in the discussion about the features of the system. However, it is difficult to map the general structure, or the functionality of an specific component, to a specific reason: they are usually the mixture and balance between several factors. In some other cases, they are in a certain way just because that way worked (to be honest, the system has evolved for more than five years, and some of that evolution has been not completely planned in advance). However, we did try to point out the rationale in the new version of the paper.

## Comment R2.18 *Section 2. Main points (2)*

Given the paper's descriptive characteristics, I would recommend having a formal notation (e.g., UML) to describe the components and their relationships (Figures 1-7 and Figures 10-14). For example, the meaning of the relationship shown between components (Figures 1-7) is not defined (does the arrow represent data flow? functional dependency? or something else?). Also, although some colour and shape code is used in Figures 9-14, the meaning of such codes is unclear.

**Answer to R2.18**

We have tried with a new version of all the figures representing the structure, components, and scenarios. We have not used UML because we didn't want to detail all the structure, but just provide an overview of how the system, and the modules, are structured. However, we have clarified the meaning of arrows. In most cases, they represent flow of data, and when they mean something else, we have explained that (with a different type of arrow).

## Comment R2.19 *Section 2. Main points (3)*

There is no discussion regarding the separation of concerns among components.

**Answer to R2.19**

We have addressed this concern with some text at the beginning of the section, which explains the division in areas:

> *"This separation in areas is introduced to help in the process to understand GrimoireLab components, and their role in the functionalities provided. At the same time, it allows for the introduction of the main interfaces that allow for the relatively independent development of the components presented in the rest of this section."*

We have also added a brief explanation of the main interfaces between the areas (which are provided by some of the components, as stated in the rest of the section).

We have also included discussions, in the four areas (2.1-2.4) trying to explain the rationale for the different components, and they respective areas of concern.

In the process, and also to address other comments, we have renamed the four areas to "Retrieval", "Analytics", "Identities Management", and "Visualization and Reporting", and modified the graphics accordingly.

We hope all of this helps to clarify the aim of each of the components, and their relationships.

## Comment R2.20 *[P3,L104]*

Could one interpret the sentence "... describes the structure of GrimoireLab and its module ..." as "... describes GrimoireLab's architecture ..."? If so, which architectural level the paper describes? Also, the text interchangeably mixes the words "structure" and "design". Please, clarify if they have the same intended meaning.

### Answer to R2.20

Changed to *"describes the different components of GrimoireLab"*, so that it better shows what the section is about.

## Comment R2.21 *[P3,L128-131]*

"Graal runs third party tools on git repositories.", "... runs a collection of tools on checkouts ...", "... produced by those tools ...". The mentioned tools need to be clearly defined, at least what are their outputs.

### Answer to R2.21

We've tried to clarify this by editing the paragraph:

> *Graal runs third party tools on git repositories, to obtain source code analysis data, at the file level, for each commit found. It uses Perceval to get the list of commits, and then runs the tools selected on checkouts of those commits. Graal can run tools for computing metrics in the areas of code complexity, code size, code quality, potential vulnerabilities, and licensing. Graal captures the output of these tools, encoding the data they produce in JSON items similar to those produced by Perceval.*

## Comment R2.22 *[P5,L150-159]*

Please, clarify that are the "common features" that are reused to collect data from different data sources and which components implement those features.

### Answer to R2.22

We've tried to clarify that, including more detailed examples, in a new version of that paragraph.

## Comment R2.23 *[P5,L164]*

"... controlling the details of the job ...". Which are the controlled details and why?

### Answer to R2.23

We have now used "specifying" instead of controlling, which is more appropriate. We have tried to clarify which details are specified with this text:

*"These details include the category of the job, parameters to run Perceval, or parameters to the scheduler, such as the maximum number of retries upon failures."*

## Comment R2.24 *[Perceval and Graal]*

Perceval and Graal components seem to have a strong coupling. Please, clarify how they are typically integrated.

**Answer to R2.24**

In the description of Graal, we have added some detail about that, when describing its functioning:

*"Graal uses Perceval to clone the git repository to analyze, and to get its list of commit hashes, via the Graal Client module."*

## Comment R2.25 *[Section 2.3] (1)*

The motivation for the identity management component is well described. However, I would recommend addressing the following points: 1) There are many academic papers describing different heuristics for identity disambiguation (e.g., resolving users that adopt different e-mail addresses to commit to the codebase). Different heuristics are associated with different performance (e.g., accuracy). How does GrimoireLab's approach to identity disambiguation compare with related work in this area? What is the performance of GrimoireLab's approach?

**Answer to R2.25**

In fact, SortingHat uses a very conservative approach, and therefore it is not using those algorithms in the literature. Experience has shown that having identities wrongly merged is far more a problem than having identities not merged at all. We have added this paragraph explaining this approach:

*SortingHat uses a very conservative approach to merging identities: it uses algorithms that are quite likely to only merge identities that really correspond with the same person. This approach is used because in production environments, experience has shown how erroneously merging identities causes much more problems than failing to merge some identities, and because it can more easily be complemented with manual curation of the data. Therefore, SortingHat periodically merges identities using these conservative algorithms, that can also be activated (or not) in its configuration. If more detail is needed, ingestion of identities data from reliable sources (such as company records, or FOSS Foundation data about its developers), or manual curation (usually via HatStall) can be used. However, since SortingHat offers an API to manage the identities it stores, more aggressive automatic algorithms for merging them could be easily implemented."*

If it is considered convenient, we could add some illustrative example of the problems that even simple, conservative algorithms cause. For example, the naive algorithm of "merge two identities if the email address is present in both, and it is exactly equal", fails in large datasets for common

cases such as "root@localhost", merging for example "John Smith ¡root@localhost¿" with "Mary Williams ¡root@localhost¿". SortingHat provides this algorithm, which can be activated, but we had to include a deny list with common addresses such as this "root@localhost" to make it useful.

## Comment R2.26 *[Section 2.3] (2)*

2) "... sends them back to be added to the enriched data." [P6,L202] is not clear in Figure 7. Also, by Figure 7 it is unclear how "enriched data" relates to other sub-components of SortingHat. In general, the textual description of the figures needs to be improved.

### Answer to R2.26

We have redesigned the figures for GrimoireELK and SortingHat, among others, including more descriptive captions. We hope they are now more clear and useful.

## Comment R2.27 *[Section 2.4]*

Although the section states Visualization and Analytics, what is actually described relates to visualization and data exporting. Also, please clarify which "documents" are referenced in [P7,L213] and what "actionable inspection" [P7,L216] means in this context.

### Answer to R2.27

We have changed the section to "Visualization and Reporting", which we hope captures better the components being described.

We have clarified which "documents" we're referring to:

> *" It can also produce reports as PDF documents, including a part of the information in those tables, with some textual explanations."*

We have tried to clarify "actionable inspection" by changing the text to this other:

> *"web-based actionable dashboards (users can interact with the data shown, by filtering, bucketing, drilling down, etc.)"*

## Comment R2.28 *Enriched indexes*

The concept of "enriched index" needs to be clarified and better formalized.

### Answer to R2.28

We've tried to clarify the term in the introduction to the section, when presenting the interfaces to the components:

> *"Analytics and permanent storage is always accessed through the enriched index, which is the final result of the proceeding done in these modules. The enriched index provides a database with a flat JSON document per item. These documents are suitable to plug to visualization tools, or to perform further processing (for example, mapping collections of JSON documents to Python Pandas dataframes) towards specific reports."*

## Comment R2.29 *Figures*

There is excessive usage of Figures that hinders readability. While I appreciate the effort to describe the relationship between GrimoireLab's components incrementally, many of the figures are redundant and can merge into a single one that conveys the same message. Also, the textual description of many figures (e.g., Figures 4 and 8 ) is lacking.

**Answer to R2.29**

We have redesigned most of these figures, and their relationship with the text. We have also added (hopefully) more clear textual descriptions.

## Comment R2.30 *Section 3*

Main point: this section lacks a detailed description of the techniques used by each component. Also, it seems that a large portion of the section's contents overlaps with the previous section's contents (in particular, most of the section describes the relation between components and how they individually work). Please, consider improving the separation between the sections.

**Answer to R2.30**

This section (now Subsection 1, 2, 3 of Section 3) is intended to provide examples of the different scenarios in which GrimoireLab can be used, and how the different components fir together in these scenarios. We've tried to clarify this, and separate it from the previous section by making the contents in the previous section more general (about how the components work and interact), and this more particular about the specific scenarios presented. We have also structured each subsection, so that its aim becomes more clear.

## Comment R2.31 *[Section 3.1]*

What are the advantages of using Perceval compared to having a script that uses CURL with pagination, for example? Also, does Perceval collects only issues and pull-requests from GitHub? If that is the case, the paper needs to discuss this issue as a limitation of GrimoireLab. More generally, although the system can collect data from more than 30 data sources, not all data from each source can be collected.

**Answer to R2.31**

The advantages of Perceval in this specific case are mainly not having to even know about the API: you just use Perceval as you would do for any other backend. No need to know the API options, or how to access it. Of course, since Perceval uses the API, it is always possible to directly access the API: Perceval just puts uniformity and simplicity on top of it. We have tried to explain this when commenting the advantages of GrimoireLab in this area, but we could improve that explanation if convenient.

Indeed, Perceval does not access all possible APIs for all supported backends. In the case of simple APIs, it usually supports it completely, but when there are several (as is the case of GitHub), only some of them are usually supported. We have tried to clarify that with some new text, in the description of Perceval in Section 2:

> *"although for some of them, not all APIs are always supported"*

In the case of GitHub, Perceval currently collects data from several APIs, but not all of them. In detail, it collects data from the issues API, which provides items about issues and pull requests, and from the reactions API, which provides comments and other reactions to issues and pull requests. The documentation about the detailed interface to the Perceval GitHub module is at

`https://perceval.readthedocs.io/en/latest/perceval.backends.core.html#module-perceval.backends.core.github`

In particular, the git API is not supported because GrimoireLab supports git directly.

## Comment R2.32 *[P8,L242]*

Section 3.1 describes a data retrieval scenario and claims that "... which will use Perceval (as library) in the background, to clone the repository and get the list of commits, and then run third party tools that analyze each relevant file in each commit to get complexity metrics from them in a single JSON". It seems that Perceval is doing both data retrieval (e.g., cloning repositories and getting commits) and data analysis (e.g., calculating complexity metrics). I wonder how much this characteristic of GrimoireLab (specifically, Perceval) is based on best practices. For example, why does the same component perform retrieval and analysis? As these are different concerns, it seems that different components should address them. Another question raised by the quoted sentence is how "relevant files" are set in Perceval.

### Answer to R2.32

We have modified to text to avoid the confusion you mention. Perceval and Graal have different functions: Perceval deals with data retrieval, and in the case of git repositories, it uses the approach of cloning the repository before the analysis. Graal is the tool in charge of computing source code metrics, via the execution of third party tools. To avoid repeating the cloning and commit finding functionality already in Perceval, Graal uses it instead of reimplementing it. We have tried to clarify this with this text:

> *"A similar approach can be used to analyze files in any checkout of a git repository. In this case, we will use a single command to run Graal, which will use Perceval (as library) in the background to clone the repository and get the list of commits. Then, Graal will run third party tools that analyze each source code file in each commit […]"*

We have also changed "relevant files" with "source code files", which in this case are the relevant files when computing complexity.

## Comment R2.33 *[P9,L276]*

In the sentence, "SortingHat will work based on heuristics ...", what are those heuristics and how well do they perform?

### Answer to R2.33

We tried to explain these heuristics above (and in the paper, in the previous section). In the current version of the paper, most of this has been removed, as a part of removing material which is duplicated in the previous section.

## Comment R2.34 *[P9,L282-284]*

The technique adopted by GrimoireLab for incremental data retrieval needs to be way better described. For example, it is unclear from the sentence "... which can be used to select which items from the database need to be retrieved." [P10,L283] how such items are selected. Also, the sentences "... how often will data sources be visited for incremental retrieval." [P10,L287] and "... refresh periods (how often data will be retrieved incrementally from repositories)" [P10,L290] gives the impression that the incremental data retrieval is not real-time but instead performed in batches. If that is the case, how this feature compares with real-time techniques such as processing an event stream?

**Answer to R2.34**

We have tried to clarify the strategy, and why event streams are not used, in the previous section (as a part of the recommendation to avoid duplication with this section). A part of that text (at the end of the previous section) now reads:

> *"[...] In general, GrimoireLab does not use event streams and similar synchronous APIs, because they usually do not allow for the retrieval of past items, which are already not available from them. Instead, it uses timestamps and batch retrieval from APIs that provide all the items in the history of a repository. For allowing this incremental retrieval, GrimoireElk includes some metadata in raw and enriched indexes, based on the date when retrieved items were last updated. This metadata can be used to query data sources for all items since last update, and when processing the raw index, all items since the last processed. Even when these techniques in some cases are more complex than those based on event streams, they ensure complete retrieval of all items in the data source at the price of polling it frequently to check if new items are available. Fortunately, most of the use cases allow for some minutes of delay in data processing, which means data sources are not polled too much. [...]"*

## Comment R2.35 *[P10,L294]*

The paper says that "... Graal analyzes source code, by running third party tools with the help of Perceval.". From Figure 2, it is possible to verify that Graal is within the "data retrieval" component. This observation evidences that the paper lacks a discussion on the separation of concerns of the components. Also, there is no illustration of the relationship between Graal, third-party tools, and Perceval in any of the figures.

**Answer to R2.35**

Graal is within "Data retrieval" because we consider "data retrieval" as everything producing raw indexes (that is, collecting data from data sources, and storing it in the Elasticsearch raw database). We have tried to clarify this when we present "data retrieval" early in this section. Graal is in this realm because it collects metrics (using third party tools) from the git repository (several metrics for each version of each file).

We have now included in Figure 5 (Overview of the structure of Graal) a reference to the third party tools. We have also added a text describing the relationship between Graal and Perceval, stating how Graal uses Perceval only to get the list of commits (snapshots) to analyze:

> *"Graal uses Perceval to clone the git repository to analyze, and to get its list of commit hashes, via the Graal Client module."*

In case it is considered convenient, we could include a simplified UML diagram with the modules interfacing third party tools.

## Comment R2.36 *Section 4*

Main points: I am surprised by how much Section 4 overlaps with prior sections. It claims to describe "three real use cases for GrimoireLab", while the previous section claims that "GrimoireLab can be used in many different scenarios. In this section we describe some of them ...". Please, consider improving the separation of the contents described in Sections 2, 3 and 4. It seems that Section 4 has the potential to describe insightful empirical studies regarding GrimoireLab. However, there is no definition of research questions, no description of rigorous methods to perform such empirical studies, and no discussion of the learned lessons. The lack of a clear and well-thought study design leads to severe issues to the validity of the claims.

### Answer to R2.36

We have merged section 3 and section 4, addressing the overlapping by more clearly focusing the two parts of the new section, removing some text that just repeated observations, and in general, explaining in the text both the relationship of "scenarios for research" and "use cases", which are the two issues dealt with in it. We have also revised section 2, removing some redundancies.

In this paper, we didn't tried to describe empirical studies. We cite some literature using GrimoireLab (see section "Availability and usage"), but our focus in the paper was to show how GrimoireLab is designed for being used in several setups, mostly in the data retrieval and storage areas. It provides also some specific support for studies (mainly exploratory studies), and we have tried to stress that in the current redaction of this section. We have presented some possible RQs that could be addressed with studies that would use GrimoireLab, and we have tried to be specific in describing the methods that could be used in those cases, including the specific GrimoireLab setup needed. We have also tried to revise all claims, being more specific and careful about them, trying to ensure that they are supported by facts or clear explanations. We have also included a new study (the Cauldron GNOME study), which we designed specifically to measure performance of the system in one of the described scenarios. We added detailed data about it (including all system logs, and dumps of the database produced) in the reproduction packages, and a summary of results in the text of this section.

## Comment R2.37 *[Section 4.1]*

The main takeaway from the analysis presented in this section does not stand out. For example, how does the running time change by changing the associated factors (what are the factors in the first place)? Also, what is the baseline for comparison of the presented results? How good or bad are the presented results compared with other similar systems? What are the strengths and limitations?

### Answer to R2.37

In all these use cases, we didn't try to show that GrimoireLab is better or worse than other systems, mainly for two reasons: first, there is little literature showing comparable results, to compare with; second, we think that, given a certain performance, other factors are more important for the researcher, such as convenience, availability of specific features, etc. So, we didn't try to frame use cases as research studies to conclude anything in particular, but as (real) examples

showing the general usefulness of GrimoireLab in a specific use case. We provide some numbers to provide an idea of the general size of the case, and when possible, of the suitability of it (in terms of "doing the job in a reasonable amount of time with a reasonable amount of resources"). That's why, for example in this (the WMF case), we try to show that a real set of repositories, which in many cases would be considered as "large", the analysis is possible, and in fact is working day and night for several years now. We have tried to clarify this approach in the text.

Also, trying to make the rationale for the use cases more clear, we have changed the structure of this subsection (now "Use case: One-time analysis of a collection of repositories"), and the remaining use cases, so that they can be better understood, exposing their main requirements, magnitudes, and GrimoireLab setup used.

## Comment R2.38 *[P11,L312]*

How is GrimoireLab's performance evaluated?

### Answer to R2.38

We have described this case, and the new study (GNOME GitLab) in some more detail, so that it is more clear what performance is evaluated. In summary, we are mainly concerned by the throughput (items per second) that can be processed in a relatively basic machine, which could be the baseline for the kind of machines used by researchers these days. As mentioned above, we don't intend to claim that the performance is better than that provided by other systems, only that the performance is reasonable for many kinds of use cases.

## Comment R2.39 *[P11,L314]*

The sentence "The main figures of that experiment are included in Table 1". I do not understand what "figures" means, neither which "experiment" is described. In addition, what is Mordred in the sentence "The experiment was run by Mordred"?

### Answer to R2.39

To avoid confusion, we have removed that sentence, and we refer to the table in the item "Magnitudes" of the use case. We also try to describe the use case, and we have removed the word "experiment", because we agree it is more a use case than an experiment. Mordred is described with other components of GrimoireLab (in section 2): it is a component that orchestrates the retrieval of a collection of repositories.

## Comment R2.40 *[P12,L319]*

"... sleep for a little while ...". This sentence is imprecise. Also, how much of the "little while" is reflected in the results shown in Table 2.

### Answer to R2.40

This is now in the next use case, because it was confusing in this one (it described the process to later say that it didn't use incremental retrieval, which was right. So, we moved the description to the Wikimedia Foundation use case, where incremental retrieval is done. We have also changed the wording to be more precise:

*"threads sleep for a configurable amount of time (300 seconds by default)"*

## Comment R2.41 *[P12,L320]*

"The data shown in this paper ...". Please, clarify which data.

**Answer to R2.41**

This sentence has been removed in the current version of the paper.

## Comment R2.42 *[P12,P324]*

"Data collection for git includes cloning of git repositories for GitHub, and production of the raw index by analyzing those clones". How each of these steps is reflected in the results shown in Table 2?

**Answer to R2.42**

This is now clarified in the caption for Table 2:

*""Collection finished (git)" means all git repositories were cloned and their metadata was stored in the raw index."*

## Comment R2.43 *[Table 3]*

There is no reference of Table 3 in the text.

**Answer to R2.43**

We have included it in the description of the use case.

## Comment R2.44 *[Section 4.2]*

What is the main takeaway of Section 4.2?

**Answer to R2.44**

We have tried to clarify it now, with the new structure of use cases (it is now subsection 3.5), and the last paragraph in the subsection:

*"This use case shows how GrimoireLab can be used in production, for continuously analyzing large-scale projects, during long periods of time. We consider this use case, which is representative of a number of others similar, as an illustration of the maturity, and adaption to real-world constraints, of the toolset."*

## Comment R2.45 *[P14,L329]*

"Table 4 shows data about an example of large deployment of GrimoireLab". What does "large deployment" mean?

**Answer to R2.45**

We have tried to clarify the use case with the new structure of the subsection.

## Comment R2.46 *[P14,L335-336]*

"The deployment has been running continuously for more than four years, retrieving data incrementally from repositories every few minutes". How many minutes?

**Answer to R2.46**

We have clarified that in the text:

> *" threads sleep for a configurable amount of time (300 seconds by default), and then restart the process"*

## Comment R2.47 *Section 5 [P16,L365-367]*

"... metrics are not (except for some exceptions) a part of the indexes stored in Elasticsearch: they are computed either by the visualizations, or by the tools producing the reports.". What is the rationale behind this design decision? Why is it better recalculating the metrics instead of storing them?

**Answer to R2.47**

In fact, it is aggregated metrics that are not included in the enriched indexes. We have clarified this (now section 5 is the final subsection in section 2).

We have also added a discussion about the trade-off that lead us to not having aggregated metrics in the database, in the Discussion section.

## Comment R2.48 *Section 6*

Main point: this section needs to improve the discussion of the novel learned lessons after years of research and development of GrimoireLab. For example, prior studies discuss the same pointed issues: "minimizing interactions with data sources", "data source APIs are not uniform", and "data provided are not uniform". How good or bad are the solutions proposed by GrimoireLab? More importantly, which findings and evidence support the claims made in this section? Without clearly providing evidence to support the discussion, most of the claims are understood as speculation. Another important point is that, in general, there is a good amount of discussion about what the system does, but barely any discussion on how and why.

Some excerpts in this section deserve clarifications. Below I point to specific excerpts.

**Answer to R2.48**

We have tried to restructure the paper so that it is more clear what is a comparison with other systems, what is a description of GrimoireLab, what is possible uses of the system, and what

is discussion. In particular, we have added a subsection on some takeaways from the industrial experience in the company promoting GrimoireLab (entering as much as possible into that, without touching aspects that the company prefer not to publish), and some more rationale for at least some of the specific aspects that drove the design of GrimoireLab. Please, see how we addressed the specific clarifications requested, below.

## Comment R2.49 *[P18,L405]*

"... do some different tricks depending on it". Please, clarify what those "tricks" are and provide the motivation for using them. Preferably, use more specific and technical wording.

### Answer to R2.49

We have made this more precise by changing the text to:

> *"and access it in slightly different ways (for example, using different parameters in a call to the API) depending on the version detected FOOTNOTE: For example, there are many versions of Bugzilla, providing similar but not exactly equal APIs. One of this differences is the name of the parameter to order a list issues by date: Last+Changed (prior to the 3.4 branch) or changeddate (starting with 3.4 branch). The difference is minimum, but enough to break the retrieval process if it uses the wrong name."*

## Comment R2.50 *[P18,L414]*

"... the code may be more generic ...". It is unclear which "code" is being referenced.

### Answer to R2.50

We have tried to clarify the text:

> *"This means that the retrieval and enrichment code is more generic, and more resilient to changes in data structure. The retrieval code will just get everything the data source provides, marshalling it as JSON. The enrichment code will define the mappings between the fields in raw and enriched items, with most of the code for the actual enrichment being reused. When changes in the fields provided by the data source don't affect enrichment items, enrichment code needs no change at all to accommodate these differences."*

## Comment R2.51 *[P19,L450]*

"... to address the first approach ...". Please, clarify which "first approach", as the previous sentence was talking performance issues of data collection.

### Answer to R2.51

We clarified this by changing the text to:

> *"To minimize stress on data sources"*

## Comment R2.52 *[P19,L450]*

"... to avoid those that could cause trouble." Please, clarify what type of trouble. Also, please, prefer a more technical and specific wording.

**Answer to R2.52**

We changed this text to:

> "selecting those more appropriate for each retrieval job."

## Comment R2.53 *[P19,L470-471]*

"GrimoireLab recovers as nicely as possible by retrying, and by continuing after failure". Please, elaborate on how this feature is implemented. Is there any increase in the waiting time after retrying? How each type of failure is handled? Is the failure handling approach dependent on the data source?

**Answer to R2.53**

We have tried to clarify in the text how several components of GrimoireLab behave in the presence of token rate exhaustion.

## Comment R2.54 *[P19,L475-476]*

How can consumers detect errors in the retrieved data? What are those errors? What are the most common ones?

**Answer to R2.54**

It is human consumers. We have tried to clarify in the text, providing some examples. We don't have evidence about which ones are most common, so we have not specified that. Current text is:

> "Data retrieved and enriched should be easily inspected, so that people using it can detect errors. Errors can happen, for example, when the wrong repository is configured, or when timestamps are shifted due to misconfigured default timezones, or when bugs in the code produce some field with errors."

## Comment R2.55 *[P19,L477-478]*

What is a "bug in time interpretation"?

**Answer to R2.55**

Those are bugs due to incorrect parsing of date formats. We have tried to clarify the text:

> "For example, a bug due to incorrect parsing of a date format can be fixed..."

## Comment R2.56 *[P20,L494-502]*

Reproducibility: why providing a reproducible build of GrimoireLab instead of providing a dump of previously collected data? The argument that "that would be enough for anyone to get the same data again, provided data sources didn't change" does not hold, as rarely a data source will not change.

### Answer to R2.56

In fact, with GrimoireLab both things can be done. We have rewritten the paragraph trying to make it more clear. In short, both reproducibility from the original API may be achieved (for example, to repeat an study some years later, including the data produced during that time), and reproducibility from the retrieved dataset.

## Comment R2.57 *Section 7 [P20,L503-506]*

I do not understand what this paragraph conveys. For example, what does "production of merged collections" mean? Please, clarify the sentence "The availability of origin fields simplify the merge, allowing for complex combinations, such as using raw indexes for different data sources to compose a meta study".

### Answer to R2.57

We have tried to clarify the paragraph, by rewriting it. We hope it is more clear now.

## Comment R2.58 *Section 8 [P20,L530-533]*

How the two proposed categories of related work arose? Is this classification by the own authors? What process was used to achieve this classification?

### Answer to R2.58

We propose two categories: datasets about a certain collection of repositories with tools to produce them; and tools to produce datasets about specific repositories. Although there is some overlap between them, we think these two categories help to better understand the state of the art, and the historical context. We propose these two categories with the only idea of helping the reader to understand related work. The classification is by we, the authors. The differentiation between the two categories seemed natural to us. Once defined, we classified related work according to them.

## Comment R2.59 *[P20,L541-542]*

"... not only metadata about the projects, but as much data as possible about software development ...". Please, clarify what are the differences between "metadata about the projects" and "data about software development".

### Answer to R2.59

We have tried to clarify, with the following text for "metadata":

> *"The kind of metadata they provide for each project include project description, project status, programming language, developers, license, programming language, and some general statistics about it."*

and the following for "data about software development:

> *"(the complete list of commits or issues, for example"*

## Comment R2.60 *[P21,L563-567]*

I appreciate the comparison between GrimoireLab and GHTorrent (same for the paragraph of [P21,L591-599]). While I agree that using the projects API will require a reduced number of queries to the endpoint, the events API (as used by GHTorrent) provides a richer source of data about the activities performed in a repository. Please, clarify which type of data can be obtained by using the projects API in comparison with the events API and vice-versa.

### Answer to R2.60

Bot APIs provide similar information. The main difference is about how they return data: for "recent" data the events API, for all the history of the project the projects API. We have tried to clarify this in the current text.

## Comment R2.61 *[P21,L576-577]*

"Boa is a system ..." [L576] and "Boa is a programming language ..." [L577]. What is Boa anyways?

### Answer to R2.61

It is both a system and a programming language that it provides (the same name is used for both). We have tried to clarify with:

> *"Boa is a system to massively collect source code, analyze it, and store some of its characteristics in a database, allowing researchers to query on them. Boa also provides a programming language (also named Boa) ..."*

## Comment R2.62 *[Lack of subsections]*

To improve readability, I would recommend separating Section 8 into sub-sections (per related work's theme).

### Answer to R2.62

Thanks for the suggestion. Done.

## Comment R2.63 *[P22,L628]*

Please, define the acronym "SQO-OSS".

**Answer to R2.63**

Done. It stands for "Software Quality Observatory for Open Source Software".

## Comment R2.64 *Section 9*

I appreciate seeing a mature and well-supported set of tools being provided to the academic and industrial communities of mining software repositories. Many of the provided tools are tested in-field and used to support different types of research, which clearly shows the value of those tools. As a side note, it would be a great addition to this paper if any lessons learned from the usage of GrimoireLab by the core services of a company could be discussed, even if the discussion stems from public information.

**Answer to R2.64**

Thanks for the suggestion. We have added a specific subsection about this, "Industrial use". It is written after some conversation with the persons building and using the software at the company. Unfortunately, we could not keep detailed records of those conversations, but since some of the people involve are authors, we could consider most of it as reflections by the authors, informed by their knowledge of the industrial experience, and by the opinions of other colleagues involved in it.

## Comment R2.65 *Section 10 [P23,L685]*

"... an industry-strength platform ...". What evidence is given for this claim?

**Answer to R2.65**

None specific. We have changed this to "mature platform" instead, to avoid unsupported claims.

## Comment R2.66 *[P32,L687-699]*

This excerpt claims unique characteristics of GrimoireLab that differentiates these tools from others. However, in the list of characteristics, we can read "support of different data sources", "flexibility and configurability", "identity merging", "tested in real-world", etc. Unfortunately, none of these characteristics are "unique" to GrimoireLab and can be observed in other systems or academic papers. Perhaps, the unique feature of GrimoireLab is how all these characteristics are put together, which reinforces the need to present a strong comparison between GrimoireLab and other existing related systems.

**Answer to R2.66**

We agree that one of the key values of GrimoireLab is how it includes several characteristics that can be found in other systems, but usually not all of them together, combined in a coordinated way. We have produced a comparison table, Table 6, to compare some of the characteristics of GrimoireLab and other systems mentioned in the related work, and have included some text trying to highlight this situation.

# Answers to Reviewer 3

### Comment R3.1 *Basic reporting*

Overall the manuscript is well structured. The writing is clear with only few syntax, grammar, or spelling mistakes. I recommend carefully re-reading the references to fix some capitalization errors.

### Answer to R3.1

Thanks a lot for your comments. We have tried to address them (see details below).

We have gone though the references, fixing many capitalization errors. We have also fixed some other grammar and spelling errors.

### Comment R3.2

The numerous figures provided are helpful. I suggest using a standardized notation (such as UML) or explaining the meaning of the various used shapes (e.g. rectangle, vs ellipse, vs, rounded rectangle. Also, I recommend using a lighter color for the purpse shapes, which are difficult to read.

### Answer to R3.2

We have added a simplified UML diagram for one case (Perceval code). However, for other cases it grows more and more complex, and we're not sure it is useful enough. Please, review the Perceval case, and if it is found convenient, we can add UML for other components.

We have also redone most of the figures, because of this and other comments. Please, let us know if you find the changes are an improvement.

### Comment R3.3

Extend Table 2 to include the time it took for each phase to run.

### Answer to R3.3

We have included the former Table 2 and Table 3 in a single table, together, so that starting/finishing time for each phase is next to the duration of each phase. In the process, we have uncovered a bug in timing, and fixed it (numbers for performance now should be correct).

### Comment R3.4

Suffixing the duration with hours/min in Table 3 is confusing. Provide all durations simply in HH:MM:SS units.

### Answer to R3.4

Done.

### Comment R3.5

Table 3: Clarify what "items" refers to.

**Answer to R3.5**

Done.

# Comment R3.6

I recommend rephrasing the abstract's "Conclusions" section to state what GrimoireLab has achieved, rather than summarize the paper.

**Answer to R3.6**

We have rewritten that paragraph, as follows:

> *"GrimoireLab has been used in both commercial and academic projects, showing its suitability to improve the situation in the area of reusable tools for mining software repositories. It helps to reduce the effort for doing studies or providing services in the area of software development analysis, leading to advances in reproducibility and comparison of results."*

# Comment R3.7 *Experimental design*

According to the framework provided by Stol and Fitzgerald (2018), this is solution-seeking rather than a knowledge-seeking study. Solution seeking studies aim to solve practical problems for which solutions to practical problems can be engineered. In such studies, researchers design, create, or develop solutions for a software engineering challenge; in this case by building the GrimoireLab tools. As such, I do not expect a research question filling an identified knowledge gap. Instead, I expect to see description of a tool advancing the state of the art.

**Answer to R3.7**

Thanks for the focus provided by this comment. We have included some text in the last subsection of the introduction, stating:

> *"This paper presents the results of a "solution-seeking" research line, aiming to improve the situation in solving the practical problem of retrieving data from software development repositories, preparing it for further analysis, and providing basic analysis and visualization tools that help in exploratory studies. The main outcome of this lines is the GrimoireLab toolset."*

# Comment R3.8

The described system (GrimoireLab) advances the state of the art by supporting more data sources and analyses as well as enhanced interoperability. The study can benefit from a rigorous comparison between GrimoireLab and other existing systems in terms of performance, supported data sources, and analyses.

**Answer to R3.8**

We have tried to advance in this direction by including a new table in the "Related work" section, that summarizes some features of some of the most comparable systems.

We have not included a performance comparison, because we could not find data with enough detail for most of the systems, and reproducing their installation was beyond the aims of our study (and in some cases, would be very difficult to do).

## Comment R3.9

Two case studies, one on IoT repositories and one on Wikimedia Foundation repositories illustrate the power of GrimoireLab. These can be profitably extended to showcase what research questions can be answered through the provided quantitative data.

**Answer to R3.9**

We have restructured that section (merging it with the previous section), having a more formal structure for the case studies and research scenarios. We have framed use cases as instances of some of the research scenarios, and we have also extend and structured the description of the research scenarios, including examples of studies that could be done using the obtained quantitative data, and some RQs that could be answered in those studies. We have also added a third case study, illustrating in detail (with the corresponding reproduction package) how data can be obtained from another large project (GNOME), in this case using a slightly different approach which shows how the retrieval work can be parallelized.

## Comment R3.10 *Validity of the findings*

The provided conclusions mostly follow the conducted study. They can be extended through more rigorous benchmarking and more ambitious example studies.

**Answer to R3.10**

We have added another case (the GNOME GitLab case, in Subsection 3.6), mostly to provide some more benchmarking details, and to show how results are consistent in different scenarios. We have also described in more detail the example studies.

## Comment R3.11

The authors provide links to a companion package on Zenodo, which contains data, logs, and configuration files for the IoT case. I recommend that this should be expanded to include the Wikimedia foundation case, and also additional data regarding the recommended additions. Uploading a source code snapshot on Zenodo can ensure the source's long-term survival.

**Answer to R3.11**

We are uploading a new companion package, also in Zenodo. We couldn't include more data for the Wikimedia case, because we don't have the detailed data any more. But we included all the details for a similar case, retrieving the full GNOME project from the GNOME GitLab instance. For this case, we're including some scripts to parse log files, and a Python notebook to read the resulting CSV file and compute some performance metrics, which are included in the paper.

For the source code, we have adopted a different approach: we have listed the Software Heritage ids of the commits corresponding to the release of GrimoireLab used (in the reproduction package). That way, even if packages disappear from Pypi, or from GitHub, source code would still be available. Packages can easily be produced from source code, so the software would be preserved as long as Software Heritage preserves them. If a copy in Zenodo is preferred, we can also upload source code there, but the approach of using Software Heritage seemed more convenient to us.

## Comment R3.12

The provided online documentation is quite detailed and well-written.

### Answer to R3.12

Thanks

## Comment R3.13

I was able to install and run Percical using the provided instructions. On the other hand, the installation of Grimoirelab appeared to get stuck in a loop with lines such as the following.

```
Requirement already satisfied: setuptools in ./venv/lib/python3.7/site-packages
  (from importlib-metadata->flake8>=3.7.7->graal==0.2.3->grimoirelab) (50.3.2)
Requirement already satisfied: wheel in ./venv/lib/python3.7/site-packages
  (from importlib-metadata->flake8>=3.7.7->graal==0.2.3->grimoirelab) (0.36.1)
Requirement already satisfied: setuptools in ./venv/lib/python3.7/site-packages
  (from importlib-metadata->flake8>=3.7.7->graal==0.2.3->grimoirelab) (50.3.2)
Requirement already satisfied: wheel in ./venv/lib/python3.7/site-packages
  (from importlib-metadata->flake8>=3.7.7->graal==0.2.3->grimoirelab) (0.36.1)
```

### Answer to R3.13

I'm sorry about this problem. I've tried to reproduce it, but I couldn't. Current version (as of writing this answer to your review), GrimoireLab 0.2.53, installs flawlessly for me following the instructions, using Python 3.8 and Python 3.9 in several Linux flavors. We also do some integration testing when releasing, but maybe we had some error at some point. If you want, contact me directly with your architecture, maybe it is a problem with that. Or we can open an issue on your behalf, if you give me some details about the architecture, Python version, etc, and work on it until we fix the problem.

## Comment R3.14

The denormalized output of the grimoirelab tools seems to result in considerable waste. For example, each commit contains, again and again, the full details of its author and committer: about 49 elements. What is the rationale of this decision and what is its effect in terms of waster storage and processing cost?

### Answer to R3.14

Unfortunately, this is due to how Elasticsearch work. We have added some text to explain it, including how it affects to size and performance.

*"Enriched items are not normalized due to limitations of Elasticsearch, which does not support table (index) join. This has some impact on the size of the indexes (some fields are repeated once and again, when they could be in a separate table, with cross-references). However, the impact is not large, since those fields tend to be relatively small compared with the whole size of the item. The main impact of this lack of normalization is observed when one of those fields changes, and all items with the old value have to be modified. For example, if the name of an author was wrong, and is fixed, all the items authored for that person need to be fixed."*

## Comment R3.15 *L. 36*

Your phrasing focuses on the process. Consider stating that tools provide data about the software development process and the developed artifacts.

### Answer to R3.15

We agree with this approach. We have changed the text, that now reads as:

*"...relies on an increasing number of support tools [...]. Each of them maintain data about the software development process, the developed artifacts, and how developers are working. The analysis of these data sources..."*

## Comment R3.16 *L. 91*

The described attributes refer to coverage breadth, rather than exensibility.

### Answer to R3.16

Agreed. Fixed.

## Comment R3.17 *L. 129*

Generally, running tools on a commit's checkout is needlessly expensive. Accessing directly Git's file-system is more efficient, because it avoids the cost of copying all files to the OS's file-system.

### Answer to R3.17

In this case, Graal is using third-party tools, that expect regular files in a regular file system. Although it would be possible to "fake" that using the git file system, the approach of just letting git do that work seemed reasonable to us, and of course, much easier. Since git is very efficient in constructing a checkout (specially if it is based on a previous checkout), and in any case most of those third-party tools are computationally intensive, just using git seemed as a reasonable trade-off in this specific situation.

## Comment R3.18 *L. 244*

I find it a pity that Graal uses Perceval internally, rather than allowing the user to compose various tools through a common protocol and data format.

**Answer to R3.18**

Graal uses Perceval only for cloning the repository and for getting the list of commits. If needed, this could be easily done by Graal itself, but since we already had the code in Perceval, it didn't seem like a good solution to rewrite the code (or copy in Graal, for what it's worth). We've tried to make this relationship between Perceval and Graal more clear in the current version of the text.

## Comment R3.19 *L. 254*

Consider providing an actual example, rather than an abstract description. How could one e.g. measure the evolution of comment spelling mistakes over time?

**Answer to R3.19**

Thanks for the suggestion. We have kept the general schema of the code when using Perceval as a module, but we have also added an example of how to compute spelling errors per year in commit messages of a set of repositories.

## Comment R3.20 *L. 490*

I recommend backing up this claim with benchmark results.

**Answer to R3.20**

We agree we don't have enough performance data to ensure efficiency. However, we are including some more benchmark results (the GNOME GitLab experiment), and we have watered down the claim, which now is:

> " *GrimoireLab provides a simple way of getting the data needed for an study, with some efficiency,*"

## Comment R3.21 *L. 855*

One citation to the smartshark ecosystem should be enough.

**Answer to R3.21**

Fixed, and updated the citation to the one recommended by Smartshark authors.

## Comment R3.22 *I recommend clarifying in the manuscript following questions*

How does GrimoireLab handle GitHub pull requests?

**Answer to R3.22**

We have added some text in the description of Perceval, hopefully clarifying this specific case:

*"When a data source provides several types of items, Perceval usually labels the resulting items in a way that can be identified by other components processing them later. For example, the GitHub Issues API provides both issues and pull requests for a repository: Perceval uses the field* `pull_request` *to let other components know if the item is an issue or a pull request."*

## Comment R3.23

What is the schema of the data provided by Perceval?

### Answer to R3.23

We have added some text, and a figure, to clarify this detail:

*"The output of the execution of Perceval is a list of Python dictionaries (or JSON documents), one per item. All these dictionaries, for all datasources, follow the same top-level schema: some fields with metainformation that can be used for traceability, for incremental retrieval, and to simplify tasks by other components. Figure 5 shows an example of the top level fields for an item corresponding to a GitHub pull request. The field data is a dictionary with all the data produced by the data source API, with a structure as similar as possible to the one produced by that API."*

## Comment R3.24

Do SortingHat and HatStall offer an help to address GDPR requirements? How do you propose that the tools should be employed for handling them?

### Answer to R3.24

This is a very interesting topic. We have added some text at the end of the subsection on identity management that tries to clarify the situation for the whole of GrimoireLab:

*"Most identities found in software repositories can be considered as personal informa-tion, therefore subject to laws protecting privacy, and to ethical guidelines on the matter. Due to this circumstance, in some cases identity management can deemed unethical, or unlawful (for example, under GDPR, if there is no clear legitimate interest for the pro-cessing of personal information, and it is considered that there is no informed consent from identity holders). To have this situation into account, GrimoireLab allows for the pseudoanonymization of identities as they are retrieved, via configuration switches in Perceval and GrimoireELK. If those switches are activated, Perceval hashes identi-ties found in retrieved data, and GrimoireELK does not use SortingHat, producing raw and enriched indexes with pseudoanonymized identities. When orchestration is used, switches are activated with an option in the Mordred configuration file."*

In the specific case of SortingHat, GDPR was not a thing when SortingHat was designed, so it didn't have it into account. If the above options are used, personal information is not stored or managed, which is good enough for some use cases (for example, when identies for the same person

need to be merged for getting meaningful results). There are plans to add some functionality to the system to deal with some of these cses, but for now, none are implemented. We reference this situation with the following text (in "Identity management", when describing the fatures of the system):

> *"A related problem to identities is privacy: in many cases, identities should not be provided to consumers of the data, to respect privacy of the persons participating in projects. Currently, there is on-going work in GrimoireLab to improve the situation in this area."*

## Comment R3.25 *L. 234*

Perceval appears to retrieve the repository's metadata and also a clone of the actual repository. How does it handle clashes of repository names between repository names from different sources?

### Answer to R3.25

For items, with the combination of "origin" and "backend" fields (see Figure 5).

In the specific case of git, when cloning the repository, the whole "origin" is used (after a simple transformation) as the name of the dictionary where the clone is stored.

## Comment R3.26

How can GrimoireLab be extended? What does it take to create a new data source or analysis metric? How can such additions be contributed back to the community?

### Answer to R3.26

We have added a new item, "Extensibility", in Section 4.1, providing some details in this regard. We have added also a paragraph on contributions in the Conclusion section:

> *"GrimoireLab is managed as an open free, open source software project, with a public roadmap, and all contributions managed through pull requests in GitHub. The project documents how to contribute to it, and in fact some important contributions (such as partial support for some data sources) have been received by the core team of developers. Researchers and developers of any kind are welcome to propose their patches fixing errors or providing new features."*

## Comment R3.27

What does an index hold? What operations does it facilitate?

### Answer to R3.27

We have detailed that with the following text, in Subsection 2.2 "Analytics":

> *"Since both raw and enriched indexes are Elasticsearch indexes, they are basically collections of JSON items (named "documents" in Elasticsearch). All usual operations on noSQL databases are possible on those indexes: retrieving one or more items given some constraints, aggregating values for certain fields for a certain selection of items, updating items matching certain values, etc."*

## Comment R3.28

What are the practical limits of GrimoireLab in the number of repositories or data volume it can process? On what volumes has it been tested? What support is offered for running GrimoireLab on multiple hosts?

### Answer to R3.28

Some quantification of a large GrimoireLab deployment is answered, at least in part, in Table 4, "Some magnitudes for Cauldron, as of March 10, 2021", which is the largest deployment which is not covered by NDAs.

Support for multiple hosts (using workers running in different hosts, in fact) is discussed when presenting Arthur (Section 2.1).

## Comment R3.29

How is GrimoireLab tested (c.f L. 508)? Are there some metrics (e.g. code coverage) that can help track the quality of its testing? Where do these stand today?

### Answer to R3.29

We have added some text in that item, hopefully clarifying the testing approach:

> *"GrimoireLab uses unit testing to prevent new bugs and regressions, with relatively high test coverage (Graal: 99%, Perceval: 98%, SortingHat: 93%, GrimoireELK: 82%, Mordred: 63%)"*

There is also some integration testing, but since this is not done in public, it is not mentioned in the paper.

## Comment R3.30

How are the provided GrimoireLab Docker images supported and updated? Given GrimoireLab 's evolution how can GrimoireLab users ensure the long-term replicability of their studies?

### Answer to R3.30

We have tried to clarify these details in the section "Availability and usage", with the following text:

> *"Docker images for GrimoireLab are stored in DockerHub, so that they can be recovered later (for any GrimoireLab release). They are also produced from Dockerfile configuration files, publicly available from GrimoireLab repositories."*

---

## Round 0.3 · accepted · Accept

Congratulations to you and your team on a job well done! The reviewers and I appreciate your detailed responses and your work.